# Collaborative and Efficient Fine-tuning: Leveraging Task Similarity

**Gagik Magakyan   Amirhossein Reisizadeh   Chanwoo Park   Pablo A. Parrilo   Asuman Ozdaglar**

## Abstract

*Adaptability* has been regarded as a central feature in the foundation models, enabling them to effectively acclimate to unseen downstream tasks. Parameter-efficient fine-tuning methods such as celebrated LoRA facilitate efficient adaptation of large foundation models using labeled, high-quality and generally scarce task data. To mitigate data scarcity in fine-tuning of foundation models, we propose to leverage *task similarity* across multiple downstream users. Intuitively, users with similar tasks must be able to assist each other in boosting the effective fine-tuning data size. We propose *Collaborative Low-Rank Adaptation*, or CoLoRA, which exploits task similarity to collaboratively and efficiently fine-tune personalized foundation models. The main idea in CoLoRA is to train one shared adapter capturing underlying task similarities across all tasks, and personalized adapters tailored to user-specific tasks. We theoretically study CoLoRA on heterogeneous linear regression and provide provable guarantees for ground truth recovery. We also conduct several natural language experiments with varying task similarity, which further demonstrate that when trained together with similar tasks, individual performances are significantly boosted.

## 1. Introduction

Foundation models (FMs) such as large language models (LLMs) are the essential horsepower of modern AI systems and their remarkable advances. These are large and general-purpose models that are pre-trained on massive corpora of public data such as the Internet. One key characteristic of foundation models is their *adaptability* to likely unseen user-specific tasks, also known as downstream tasks (Bommasani et al., 2021). More precisely, pre-trained FMs can adapt to new tasks with minimal *fine-tuning* to task-specific data. This remarkable feature facilitates computation-efficient adaptation of foundation models without retraining all parameters, which is clearly infeasible. Parameter-efficient fine-tuning (PEFT) is, in fact, a fast-growing area of research that devises such techniques, including the celebrated Low-Rank Adaptation (LoRA) method (Hu et al., 2022).

Supervised fine-tuning data is typically a limited set of labeled samples drawn from a distribution specific to a particular user task. For instance, suppose an AI user aims to fine-tune a pre-trained Llama model for text summarization in English. A typical fine-tuning dataset is CNN/DailyMail benchmark which consists of news articles paired with multi-sentence summaries. High-quality fine-tuning data, however, is expensive to collect, resulting in a critical bottleneck in the widespread deployability of foundation models known as *data scarcity* (Chen et al., 2023; Szep et al., 2024).

To mitigate data scarcity in fine-tuning of foundation models, we propose to leverage *task similarity* across downstream users. Consider a geographically scattered pool of users seeking text summarizer AI assistants in different languages such as English, Spanish, etc. One simple approach is for each user to individually and locally fine-tune a pre-trained Llama model using their language-specific text summarization samples which, however, suffers from data scarcity. A key point here is that these are *similar* tasks, though in different languages. For instance, one would expect similar content in summaries of an article written in English and its Spanish copy. In other words, summarization tasks across languages share a largely language-agnostic semantic representation. By leveraging this shared structure, labeled data from different languages can be pooled, alleviating data scarcity and improving fine-tuning.

In a nutshell, we propose utilizing fine-tuning data across similar tasks to capture the underlying common transformations, enabling a potentially much larger data pool. In addition, task-specific parameters help personalize and tailor the fine-tuned model to each individual user task. Now the central question becomes:

Authors are with Laboratory for Information & Decision Systems (LIDS), Massachusetts Institute of Technology (MIT).

*How can we exploit task similarity to collaboratively and efficiently fine-tune personalized foundation models?*

We propose *Collaborative Low-Rank Adaptation*, or **CoLoRA** in short, and describe it in the following. The main idea behind CoLoRA is to train two sets of fine-tuning adapters: common (or global) adapters used in all tasks which capture the underlying task similarities; and task-specific (or personalized) ones that tailor the fine-tuned models for each downstream task.

**Main contributions.** Here, we summarize the main contributions of the paper:

- We introduce CoLoRA, a collaborative (and distributed) fine-tuning approach with minimal parameter overhead. The key idea is to leverage the underlying similarities in downstream tasks and train common adapters useful across all tasks.

- We investigate the notation of "task similarity" for language tasks. To the best of our knowledge, there has been no robust similarity notion in this context, as opposed to standard deep learning tasks, such as image classification. We introduce preliminary similarity notions based on the task's adapter and utilize them both in our theoretical and empirical discussions.

- To analyse CoLoRA theoretically, we connect it to a heterogeneous matrix linear regression problem as they share their optimization objective. We utilize the Alternating Minimization (AltMin) approach and provide rigorous reconstruction error and sample complexity.

- We implement CoLoRA for collaboratively fine-tuning language tasks. Our results demonstrate significant improvement in downstream tasks when trained together with "similar" tasks. Moreover, we compare CoLoRA against multiple federated and collaborative fine-tuning baselines.

## 2. Preliminaries

Modern large foundation models consist of a pre-trained model comprising billions of weight parameters which could be key, query and value weight matrices of several layers of Transformer models. We denote a fixed layer of the pre-trained model parameterized with a weight matrix $W_0 \in \mathbb{R}^{d \times d}$ by $f_{W_0}$. Fine-tuning a pre-trained model refers to updating the model parameters to adapt to a new downstream task. More precisely, the pre-trained weight matrix $W_0$ is updated to $W_0 + \Delta W$ using the downstream task data. In supervised scenarios, such data is typically a collection of task-specific context-target pairs denoted by $\mathcal{D} = \{(x_1, y_1), \cdots, (x_N, y_N)\}$. In text summarization, for

instance, $x_i$ contains a news article paired with its summary $y_i$. For a proper choice of the loss function $\ell(\cdot)$, fine-tuning can be expressed as the following optimization problem:

$$\min_{\Delta W} \sum_{(x,y) \in \mathcal{D}} \ell(f_{W_0 + \Delta W}(x), y). \qquad (1)$$

For instance, for a pre-trained autoregressive language model $p_{W_0}(y|x)$, the loss function is the negative log-likelihood over next-token predictions, i.e. $-\sum_t \log p_{W_0 + \Delta W}(y_t | x, y_{<t})$.

### 2.1. LoRA

Updating all model parameters induces a massive computation cost and renders model adaptation inefficient and impractical. Low-Rank Adaptation (LoRA) is a parameter-efficient fine-tuning method that proposes low-rank structures for the model update. In our setting, LoRA considers the adapter $\Delta W = BA$ to be composed of low-rank matrices $B \in \mathbb{R}^{d \times r}$ and $A \in \mathbb{R}^{r \times d}$ of rank $r \ll d$, that is,

$$\min_{A,B} \sum_{(x,y) \in \mathcal{D}} \ell(f_{W_0 + BA}(x), y). \qquad (2)$$

Consequently, the number of training parameters is dramatically reduced to linear $\mathcal{O}(2rd)$ from quadratic $\mathcal{O}(d^2)$ in (1), facilitating more efficient fine-tuning.

### 2.2. Adapting to multiple tasks

Many AI applications rely on adapting *one* pre-trained model to not one but *multiple* downstream tasks. Consider $k$ tasks and their corresponding fine-tuning data $\mathcal{D}_1, \cdots, \mathcal{D}_k$. Applying LoRA (2) on each task individually leads to disjointly training $k$ pairs of matrices $(A_1, B_1), \cdots, (A_k, B_k)$ as follows

$$\min_{A_i, B_i} \sum_{(x,y) \in \mathcal{D}_i} \ell(f_{W_0 + B_i A_i}(x), y), \qquad i = 1, \cdots, k. \quad (3)$$

Although LoRA significantly contributes to parameter efficiency via small ranks $r$, still, the number of training parameters $\mathcal{O}(krd)$ scales directly with dimensions $d, r$ and the number of tasks $k$. As a result, purely disjoint fine-tuning on individual tasks prohibits task scalability.

### 2.3. Task similarity

As we observed above, in general, employing local LoRA without any collaboration between the users is not scalable. Now, what if user tasks are somehow related or similar? How can we exploit such *task similarity*? Let us elaborate on our idea with a simple example. Consider a scenario in which a pretrained model has to be fine-tuned to the following two tasks: "Task $T_1$: *Count alphabetical elements in list.*" and "Task $T_2$: *Count numerical elements in list.*".

It is fairly reasonable to consider these two tasks *similar* since they both rely on classifying elements alphabetically or numerically and counting them.

We fine-tune a base model separately to these two tasks using the local LoRA method (3) resulting in the optimized adapters $(A_1, B_1)$ and $(A_2, B_2)$ and fine-tuned models $W_0 + B_1 A_1$ and $W_0 + B_2 A_2$. If we could find shared adapters $A, B$ and task-specific ones $\Lambda_1, \Lambda_2 \in \mathbb{R}^{r \times r}$ such that

$$B_1 A_1 \approx B \Lambda_1 A \qquad \text{and} \qquad B_2 A_2 \approx B \Lambda_2 A, \quad (4)$$

then the two tasks effectively can be effectively represented with fewer parameters, $4dr$ vs. $2dr + 2r^2$. Here, $A, B$ capture the underlying common structures between the two similar tasks, and the personalized ones $\Lambda_1, \Lambda_2$ reflect the task-specific characteristics. We examine this in the following: we utilize (column) subspace similarity which measures how much column subspaces of the fine-tuned adapters overlap across different tasks. We use subspace distance as a proxy for similarity of the corresponding tasks.

**Definition 1** (Column subspace similarity). *Let $A_i, B_i$ and $A_j, B_j$ denote the fine-tuned adapters of tasks $T_i$ and $T_j$ respectively. The column subspace similarity of the two tasks is defined as*

$$\text{sim}_c(T_i, T_j) := \frac{1}{\sqrt{r}} \|U_i^\top U_j\|_F, \quad (5)$$

*where $U_i$ denotes an orthonormal basis for the column subspace of $B_i A_i$.*

Note that $0 \leq \text{sim}(\cdot, \cdot) \leq 1$. Also, larger values of subspace similarity $\text{sim}(T_1, T_2)$ imply that the two matrices $B_2 A_2$ and $B_1 A_1$ have greater overlaps between their column subspaces, hinting of a common matrix $B$ that aims at satisfying (4) approximately. Also, we can likewise define row subspace similarity[1]. We further note that to the best of our knowledge, there has been no robust and efficient task similarity measure in the context of language tasks. In contrast, notions such as Task2Vec based on Fisher Information Matrix (FIM) have been well studied for standard deep learning tasks such as image classification (Achille et al., 2019). Such notions are deemed infeasible in language tasks due to large model sizes.

We utilize the similarity metric defined in (5) to order six tasks which results in the following grid. We considered six tasks and LoRA fine-tuned a model for each task independently. Figure 1 demonstrates the subspace similarity measured across these tasks as defined in (5).

---

[1]It has been observed that in standard LoRA, trained $A_i$s tend to remain close to their initial value (Ban & Ji, 2025). Therefore, with the same initializations across different tasks, we only consider the column subspaces ($B_i$ matrices) in our similarity measure. We also observed empirically that this similarity is robust to initialization.

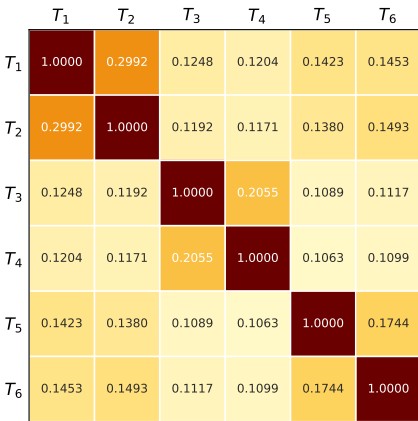

*Figure 1.* Column subspace similarity averaged across all layers.

*Table 1.* Task descriptions.

| | |
|---|---|
| $T_1$ | *Count alphabetical elements in a list.* |
| $T_2$ | *Count numerical elements in a list.* |
| $T_3$ | *In a list, multiply positives by 2, negatives by $-3$.* |
| $T_4$ | *In a list, divide evens by 4, multiply odds by 4, add 2.* |
| $T_5$ | *Count frequency of a letter in sentence.* |
| $T_6$ | *Count vowels and consonants in sentence.* |

This experiment reveals several interesting insights. By reading the task descriptions one could realize that the six tasks share similarity. For instance, tasks $T_1$ and $T_2$ are significantly similar as both involve classifying elements to alphabetical or numerical and counting them. Similarly, tasks $(T_3, T_4)$ and $(T_5, T_6)$ form similar pairs of tasks. Intrestingly, such pairs demonstrate relatively large signals in the grid above; see the similarities $\text{sim}(T_1, T_2)$, $\text{sim}(T_3, T_4)$ and $\text{sim}(T_5, T_6)$. It is also worth noting that tasks $T_1, T_2$ and $T_5, T_6$ show a similarity in the grid which can be attributed to the fact that both tasks involve counting.

**Takeaway:** This experiment further corroborates our initial hypothesis that similar tasks tend to have common underlying structures in the sense of (4) when fine-tuned with LoRA. Consequently, we could utilize this observation for a novel adapter structure when fine-tuning for several relevant tasks. This paves the way for our proposed framework detailed in the next section.

## 3. Collaborative Low-Rank Adaptation

To recap the discussion in the previous section, we hypothesize that if $n$ downstream tasks are *similar*, their fine-tuned adapters would have significantly overlapped row and column subspaces. Consequently, there exist common adapters $A, B$ and personalized ones $\Lambda_1, \cdots, \Lambda_k \in \mathbb{R}^{r \times r}$ such that

$$B_i A_i \approx B \Lambda_i A \qquad \text{for } i = 1, \cdots, k. \quad (6)$$

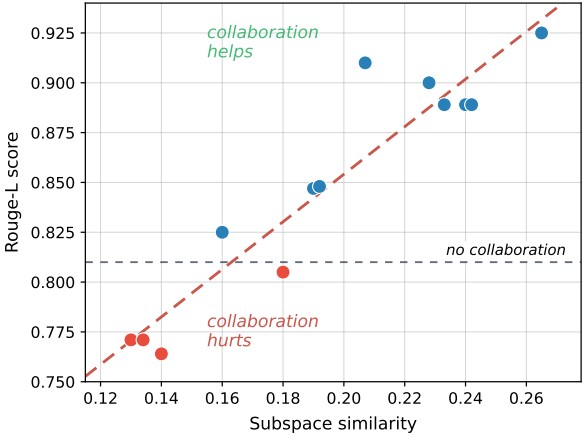

*Figure 2.* CoLoRA for Task $T$: *Given a list of integers, remove all the even elements.* Each point corresponds to task $T$'s Rouge-L score jointly trained with three other particular tasks. The black dashed line indicates the score of task $T$ when trained exclusively.

This is the main idea in our proposed Collaborative Low-Rank Adaptation (CoLoRA) method that solves the following optimization problem

$$\min_{\substack{A,B \\ \Lambda_1, \cdots, \Lambda_k}} \sum_{1 \leq i \leq k} \sum_{(x,y) \in \mathcal{D}_i} \ell(f_{W_0 + B\Lambda_i A}(x), y). \quad \text{(CoLoRA)}$$

The main motivating ideas for us to consider this particular formulation are two-fold:

**Leveraging task similairity**: As we elaborated in Section 2.3, more similar tasks tend to have lower subspace distance as illustrated in Figure 1. The proposed adapter in CoLoRA leverages task similarity by training common $A, B$ for all tasks, capturing the underlying subspace overlaps. Each $\Lambda_i$ on the other hand, tailors local adapters to specific downstream task, resulting in personalized models.

To demonstrate the task similarity point above, we focus on a particular task and monitor its performance in the data scarce regime, in two scenarios: i) collaboratively fine-tuned together with other similar tasks, and ii) fine-tuned locally without any collaboration. Let us fix task $T$: *Given a list of integers, remove all the even elements.* We experiment several instances where in each instance, task $T$ is joined by three other tasks with varying similarities to $T$. Each instance is represented by a dot in Figure 2. As illustrated, as task $T$ is trained together with more similar tasks (right side of the plot), it enjoys higher performance gains compared to training exclusively on its own data (dashed line). This experiment highlights how CoLoRA leverages task similarity and enables scalable and parameter-efficient collaborative fine-tuning. We provide further experimental results and their details in Section 5.

**Parameter count:** CoLoRA enables personalized collaborative fine-tuning with a total model size that grows as

$\mathcal{O}(dr + kr^2)$. The shared global component requires $\mathcal{O}(dr)$ parameters, while each additional user contributes only $\mathcal{O}(r^2)$ parameters. Since $r \ll d$, (typically $r = 4, 8, 16$), this growth is mild even for thousands of users, making CoLoRA practical for collaborative and federated learning applications at scale.

**Related work.** We briefly discuss some of the main related works here and defer the reader to subsection A.1 for further details.

**Federated LoRA.** Among the first attempts to adapt foundation models to federated fine-tuning is Zhang et al. (2023) where LoRA adapters are updated across users using the standard FEDAVG algorithm. The naive aggregation approach in this method, however, leads to inferior task performance because averaging LoRA factors independently across clients produces an inexact global update, as the true LoRA perturbation depends on their product. Several follow-up works address this issue by aggregating the effective LoRA updates rather than the individual low-rank factors (Bai et al., 2024; Wang et al., 2024; Singhal et al., 2024; Cho et al., 2024). Specifically, client updates are combined at the level of the full LoRA perturbation, after which client adapters are recovered via a low-rank approximation using SVD. By adopting different adapter ranks across users, this approach further facilitates adaptation to heterogeneous resource or data constraints. Alternatively, Singhal et al. (2024) keeps track of both separate and simultaneous aggregations and updates the pretrained model to compensate for the residual error.

Several alternatives avoid the costly SVD required to project the aggregated LoRA update back to a low-rank representation, instead adopting more strategic aggregation schemes. Among those, Sun et al. (2024) suggests freezing the $A_i$ adapters at initialization and only training and aggregating the $B_i$ matrices. In contrast, Guo et al. (2025) argues that the $A_i$ matrices primarily encode global knowledge, while the $B_i$ adapters capture user-specific variations, and therefore propose aggregating only $A_i$ while keeping $B_i$ personalized. A subsequent work by Ban & Ji (2025) challenges this interpretation, suggesting that the apparent similarity of the $A_i$ matrices does not arise from shared global knowledge, but from their limited deviation from a common initialization. Consequently, they advocate aggregating only the $B_i$ dapters, which they identify as more critical for effective client-to-client knowledge transfer. Furthermore, Yang et al. (2024) introduce a framework that maintains both global and personalized adapters, alternating their updates while aggregating only the global adapters at each communication round. However, this framework still suffers from the aforementioned issue: the adapters are aggregated independently, leading to an inexact global update. Chen et al. (2025)

propose alternating minimization over $A_i$ and $B_i$ and provide theoretical guarantees in the simplified case of rank-1 adapters. More related to our work is Singhal et al. (2025) that aims to substantially reduce communication costs. It proposes an adapter structure $BR_iA$, where the matrices $B$ and $A$ are kept frozen, and only the smaller adapters $R_i$ are trained and communicated.

It is worth emphasizing that the works mentioned above either do not incorporate personalization, or their effective number of parameters scales as $O(kd)$, where $k$ denotes the number of clients and $d$ the problem dimension. In this work, we propose a parameter-efficient personalization approach, thereby achieving the best of both worlds.

We also note that Brüel-Gabrielsson et al. (2025) studies a decomposition of the form $B_iA_i \approx U\Lambda_iV^\top$, with the primary goal of compressing large collections of LoRA adapters. Their experimental results further demonstrate that LoRA adapters share significant commonalities in structure that can be leveraged to mitigate memory overhead.

**Linear Representation Learning.** Our theoretical framework is related to the literature on multitask linear representation learning (Collins et al., 2021; Du et al., 2021; Thekumparampil et al., 2021; Collins et al., 2022; Tripuraneni et al., 2021; Park et al., 2024). Similar to Collins et al. (2021), we employ an alternating optimization approach. However, motivated by the LoRA structure, our setting introduces two shared representations instead of a single one, which introduces an additional layer of non-convexity and changes the problem structure.

Last but not least, it is worth noting that addressing data scarcity and model personalization is not a new direction and has been studied extensively in collaborative and federated learning literature (Fallah et al., 2020; Tan et al., 2023; Hanzely & Richtárik, 2020; Farnia et al., 2022; Mansour et al., 2020; Deng et al., 2020). The typical theme in this line of work is to train a global model using the collection of data across users and personalize it via local training, something we also utilize in our framework. While we adopt a similar global–local training paradigm, our focus is on achieving this personalization in a parameter-efficient manner suitable for federated fine-tuning.

## 4. Theoretical Understanding of CoLoRA with Linear Regression

In this section, we focus on a rigorous understanding of CoLoRA's convergence properties via a simple problem: linear regression. Consider a collection of $k$ users, each observing linear regression samples $\mathcal{D}_i = \{(G_j^i, y_j^i)\}$ governed by

$$y_j^i = \langle G_j^i, M^{i^*} \rangle, \tag{7}$$

where entries of every $G$ are i.i.d standard Gaussian. We are particularly interested in the case that $d \times d$ ground truth $M^{i^*}$s are related via their column and row subspaces. Roughly speaking, there exist common orthonormal $d \times r$ matrices $U^*, V^*$ and user-specific $r \times r$ ones $\Lambda^{i^*}$ such that

$$M^{i^*} \approx U^*\Lambda^{i^*}V^{*\top}. \tag{8}$$

More precisely, we aim to solve the following optimization problem

$$\min_{\substack{U,V \\ \Lambda^1,\cdots,\Lambda^k}} \sum_{1 \le i \le k} \sum_{(G,y) \in \mathcal{D}_i} \left(y - \langle G, U\Lambda^iV^\top \rangle\right)^2. \tag{9}$$

This problem is, in fact, similar to CoLoRA's objective in (CoLoRA) in Section 3 if we pick the loss $\ell$ to be mean-squared error. Therefore, our linear regression problem (9) is identical to an instance of CoLoRA's optimization problem for specific choices of the mapping and the loss. Consequently, we set our goal in the rest of the section to solve and analyse (9) where for each user $i$, the underlying matrix $M^{i^*}$ identifies its "task" $T_i$. To formalize how these tasks are related, we introduce task similarity based on subspace similarity.

**Definition 2** (Task similarity)**.** *For a collection of tasks* $T_1, \cdots, T_k$ *coresponding to matrices* $M^{1^*}, \cdots, M^{k^*}$*, we define the task similarity as the largest $\xi$ such that $\forall i, j$,*

$$\text{sim}_c(T_i, T_j), \text{sim}_r(T_i, T_j) \ge \xi. \tag{10}$$

*Here,* $\text{sim}_c$ *and* $\text{sim}_r$ *denote column and row subspace similarity defined in Definition* (1)*, that is,*

$$\text{sim}_c(T_i, T_j) := \frac{1}{\sqrt{r}}\|U^{i^*\top}U^{j^*}\|_F, \tag{11}$$

$$\text{sim}_r(T_i, T_j) := \frac{1}{\sqrt{r}}\|V^{i^*\top}V^{j^*}\|_F, \tag{12}$$

*where* $U^{i^*}, V^{i^*}$ *are orthonormal bases for the column and row subspaces of* $M^{i^*}$*.*

The tasks similarity notion defined above yields that for any $\xi$, there exist reference matrices $U^*, V^*$ such that for all $i$,

$$\text{dist}(U^*, U^{i^*}), \text{dist}(V^*, V^{i^*}) \le \sqrt{r(1 - \xi^2)}. \tag{13}$$

where $\text{dist}(\cdot, \cdot)$ denotes the subspace distance (See Section B.1). This further formalizes the notion in (8) and our optimization problem in (9) aims to find such $U^*, V^*$.

Note that when $\xi = 1$, (8) holds with equality for all users. In words, columns (and rows) of $M^{i^*}$s generate identical subspaces. Moreover, our optimization problem (9) covers classical and well-studies *low-rank matrix sensing* (Recht et al., 2010) for one user ($k = 1$) and low-rank ground truth $M^* = U^*\Lambda^*V^{*\top}$ ($\xi = 1$). Alternative minimization (AltMin) is a general approach for solving the low-rank matrix sensing problem Jain et al. (2013).

## 4.1. Our approach: Collaborative AltMin

Following the classical AltMin method for (one-user) matrix-sensing, we present a similar yet collaborative approach to solve our optimization problem (9). The main idea is that our optimization variables serve distinct purposes and therefore should be treated differently. On the one hand, two matrices $U, V$ will be used in the final recovered model *for all the users*, hence called *global* variables. On the other hand, each $\Lambda^i$ is exclusively used by user $i$, thus we call them *personalized* variables. All in all, we present a collaborative alternative minimization method, namely CoAltMin, in which global variables are updated using the samples from all users, while personalized ones utilize user-specific samples (Algorithm 1).

---

**Algorithm 1** CoAltMin

---

1: Initialize $U_0, V_0$ according to (14).
2: **for** $t = 0, \cdots, T - 1$ **do**
3:     **for** $i = 1, \cdots, k$ **do**
4:        $\Lambda_{t+1}^i = \arg\min_{\Lambda} l_t^i(U_t, V_t, \Lambda)$.
5:     **end for**
6:     $\widehat{V}_{t+1} = \arg\min_V \sum_{i=1}^k l^i(U_t, V, \Lambda_{t+1}^i)$.
7:     $V_{t+1}, \cdot = \mathsf{QR}(\widehat{V}_{t+1})$
8:     $\widehat{U}_{t+1} = \arg\min_U \sum_{i=1}^k l^i(U, V_t, \Lambda_{t+1}^i)$
9:     $U_{t+1}, \cdot = \mathsf{QR}(\widehat{U}_{t+1})$
10: **end for**
11: **return** $U_T, V_T, \Lambda_{T-1}^1, \cdots, \Lambda_{T-1}^k$

---

**Data Splitting:** For analysis purposes, we split the data samples of each user to several batches: one "large" batch of size $N$ that defines the loss $l^i$, and $T$ "small" disjoint batches of size $n$ making up losses $l_t^i$. That is,

$$l^i(U, V, \Lambda^i) = \sum_{j=1}^N (y_j^i - \langle G_j^i, U\Lambda^i V^\top \rangle)^2$$

denotes the loss associated with the large batches of size $N$ for user $i \in [k]$. Small batch losses $l_t^i$ are defined similarly for each $t \in [T]$ and $i \in [k]$. Moreover, we pick a particular initialization for global variables as follows

$$\widehat{M} := \frac{1}{kN} \sum_{i=1}^k \sum_{j=1}^N y_j^i G_j^i, \ U_0 \Lambda_0 V_0^\top = \mathsf{SVD}(\widehat{M}). \ (14)$$

Note that here, we use large batches across all users for this initialization. We defer the justification of such a choice to the appendix.

## 4.2. Theoretical results

Before presenting guarantees for CoAltMin, let us introduce a few notations and assumptions. First, note that $U^*, V^*$

are rank-$r$, therefore, in order to have nontrivial solutions, we need $M^{i*}$s to be rank-$r$ as well. To capture how well-conditioned the problem is, we let $\kappa$ denote the worst-case global conditioning among $M^{i*}$s. Moreover, we quantify the degree of alignment among them by $\gamma$ as follows

$$\kappa := \frac{\max_{i=1}^k \sigma_1(M^{i*})}{\min_{i=1}^k \sigma_r(M^{i*})}, \quad \gamma := \frac{\max_{i=1}^k \sigma_1(M^{i*})}{\sigma_r(\frac{1}{k}\sum_{i=1}^k M^{i*})}.$$

Here, $\sigma_i(\cdot)$ denotes the $i$-th largest singular value and note that $\kappa, \gamma \geq 1$. Next, we present our main result.

**Theorem 1.** *Assume that large and small batch sizes are*

$$N = \varrho^4 \min(\varrho^4 r^2, rk)(dr/k + r^2)\widetilde{\Theta}(1), \ n = \kappa^4 r^3 \widetilde{\Theta}(1),$$

*where $\varrho = \max(\kappa, \gamma)$ and $\widetilde{\Theta}(\cdot)$ hides logarithmic factors. Moreover, suppose that task similarity $\xi$ is large enough s.t.*

$$\xi^2 \geq 1 - \frac{\Theta(1)}{\kappa^2 \varrho^2 r(1 + rd/N)}.$$

*Then for any $\varepsilon > 0$ and with high probability, CoAltMin recovers $U^*, V^*$ after $T = \Theta(\log(1/\varepsilon))$ iterations with*

$$\mathrm{dist}(U_T, U^*), \ \mathrm{dist}(V_T, V^*) \leq \varepsilon + \kappa^2 r \sqrt{1 - \xi^2} \mathcal{O}(1).$$

*Proof.* We defer the proof to Section B.2. □

Let us highlight several insights provided by this result. First, the optimization error in recovering the matrices $U^*, V^*$ scales as $\varepsilon + \mathcal{O}(\sqrt{1 - \xi^2})$ which contains an irreducible error depending on task similarity $\xi$ which decreases as the tasks are more similar. This is expected since matrices with distinct column (or row) subspaces ($\xi < 1$) can not be represented by a single subspace of the same rank. Second, particularly for $\xi = 1$, i.e. $M^{i*}$s with identical column (and row) subspaces, the irreducible error vanishes and the underlying ground truth is recovered up to adjustable error $\varepsilon$. Lastly, the total sample complexity guaranteed by Theorem 1 is at most

$$\varrho^4 \min(\varrho^4 r^2, rk)(dr + r^2 k)\Theta(1) + \kappa^4 r^3 k \log(1/\varepsilon)\widetilde{\Theta}(1).$$

To gain a better understanding of this sample complexity, consider the simple case of one user, i.e., $k = 1$. In this particular case, our proof strategy can be tailored, resulting in improved sample complexity $\Theta(dr^2)$ which tightens the classical AltMin guarantee in the matrix-sensing literature (Jain et al., 2013), i.e. $\mathcal{O}(dr^3)$. We defer further additional discussion to Section A.1 and B.5.

The following corollary of the main theorem shows that $\Lambda_{T-1}^1, \cdots, \Lambda_{T-1}^k$ resulted from CoAltMin (Algorithm 1) together with $U_T, V_T$ generate the underlying ground truths $M^{i*}$s with bounded error in the matrix norm sense.

**Corollary 1.** *Under the setting of Theorem 1, global matrices $\boldsymbol{U}_T, \boldsymbol{V}_T$ and personalized ones $\boldsymbol{\Lambda}_{T-1}^1, \cdots, \boldsymbol{\Lambda}_{T-1}^k$ resulted from the CoAltMin gaurantee reconstruction error*

$$\|\boldsymbol{U}_T \boldsymbol{\Lambda}_{T-1}^i \boldsymbol{V}_T - \boldsymbol{M}^{i^*}\|_2 \le \mathcal{O}(\|\boldsymbol{M}^{i^*}\|_F)(\varepsilon + \kappa^2 r \sqrt{1-\xi^2}).$$

*Proof.* We defer the proof to Section B.4.     □

Similar to matrix-sensing problems and AltMin-type methods, our main tool for analysing CoAltMin builds on the Restricted Isometry Property (RIP).

### 4.3. Generalized Restricted Isometry Property

RIP is a condition imposed on the linear measurement operator to guarantee that low-rank matrices can be recovered from a small number of measurements.

**Definition 3** (RIP (Candès & Tao, 2005; Recht et al., 2010)). *The ensemble $\{\boldsymbol{G}_j : j \in [N]\}$ satisfies $r$-RIP with constant $\delta$, if for any matrix $\boldsymbol{X}$ of at most rank $r$,*

$$\left| \frac{1}{N} \sum_{j=1}^N \langle \boldsymbol{G}_j, \boldsymbol{X} \rangle^2 - \|\boldsymbol{X}\|_F^2 \right| \le \delta \|\boldsymbol{X}\|_F^2.$$

Several well-known random ensembles satisfy RIP. For instance, if $N = \Omega(dr/\delta^2)$ and entries of $\boldsymbol{G}_j$ are i.i.d samples from a zero mean sub-Gaussian distribution, $r$-RIP holds with high probability. Next, we introduce the Generalized Restricted Isometry Property (GRIP).

**Definition 4** (GRIP). *We say that the ensemble $\{\boldsymbol{G}_j^i : i \in [k], j \in [N]\}$ satisfies $r$-GRIP with constant $\delta$, if for any collection of matrices $\boldsymbol{U}, \boldsymbol{V} \in \mathbb{R}^{d \times r}$ and $\{\boldsymbol{\Lambda}^i\}_{i=1}^k \in \mathbb{R}^{r \times r}$,*

$$\left| \frac{1}{kN} \sum_{i=1}^k \sum_{j=1}^N \langle \boldsymbol{G}_j^i, \boldsymbol{U}\boldsymbol{\Lambda}^i \boldsymbol{V}^\top \rangle^2 - \frac{1}{k} \sum_{i=1}^k \left\| \boldsymbol{U}\boldsymbol{\Lambda}^i \boldsymbol{V}^\top \right\|_F^2 \right|$$
$$\le \delta \max_{i \in [k]} \left\| \boldsymbol{U}\boldsymbol{\Lambda}^i \boldsymbol{V}^\top \right\|_F^2.$$

In the following, we establish that the collection of large batches of size $N$ across all $k$ users used in CoAltMin guarantees $r$-GRIP with high probability.

**Proposition 1.** *Consider the random ensemble of matrices $\boldsymbol{G}_j^i$ with i.i.d sub-Gaussian entries. Then, for any $\delta > 0$ and sample size*

$$kN = \frac{dr + kr^2}{\bar{\delta}^2} \log \left( r/\bar{\delta} \right) \Omega(1), \ \ \bar{\delta} = \delta \max(\delta, 1/\sqrt{k}),$$

*the ensemble $\{\boldsymbol{G}_j^i : i \in [k], j \in [N]\}$ satisfies $r$-GRIP with constant $\delta$, with probability at least $1 - \exp(-\Theta(kN\bar{\delta}^2))$.*

*Proof.* We defer the proof to Section D.12.     □

With the choice of $N$ prescribed in Theorem 1, Proposition 1 ensures $r$-GRIP with high probability.

## 5. CoLoRA in Experiments

In this section, we investigate CoLoRA's performance in federated fine-tuning settings. We conduct experiments on NATURAL INSTRUCTIONS (Wang et al., 2022; Mishra et al., 2022). The benchmark is organized into *metatasks*, each comprising multiple related tasks (see Fig. 2 of Wang et al. 2022). We use 140 tasks in total. We primarily focus on the *Program Execution* metatask, but additionally include tasks from other metatasks. For all experiments, we use Qwen2.5-1.5B-Instruct (Qwen Team, 2025) as the base model. We freeze the base model weights and fine-tune LoRA adapters on all attention and MLP layers. We use and modify the PEFT library (Mangrulkar et al., 2022) for our experiments.

Now we introduce CoLoRA-Alt , which is our algorithm for the federated setting. Pseudocode is given in Algorithm 2. We initialize $\boldsymbol{A}, \boldsymbol{B}, \{\boldsymbol{\Lambda}^i\}_{i=1}^k$ (the base model weights are frozen), then alternate local updates and aggregation of LoRA factors. We (i) train $\boldsymbol{\Lambda}^i$ jointly with $\boldsymbol{B}$ while keeping $\boldsymbol{A}$ fixed and average $\boldsymbol{B}$ across clients, then (ii) train $\boldsymbol{\Lambda}^i$ jointly with $\boldsymbol{A}$ while keeping $\boldsymbol{B}$ fixed and average $\boldsymbol{A}$.

To simulate a data-scarce setting, we fix the number of training datapoints to 50 for all clients across all our experiments.

**Impact of task similarity**. In this section, we consider two target tasks from *Program Exeuction* metatask: $T_1^1$: *Given a list remove all the even elements*, and $T_2^1$: *Given a list of lists, multiply all odd elements in each list.* We investigate how task similarity affects collaborative learning. Specifically, for each target task $T_1^i$, we evaluate its performance when jointly trained with three auxiliary tasks exhibiting varying levels of similarity (See the tasks in Section E.1.). More precisely, given a target task $T_1^i$, we construct sets of collaborators $(T_2^i, T_3^i, T_4^i)$ such that $1/3 \sum_{j=2}^4 \text{sim}_c(T_1^i, T_j^i)$ spans a range from low to high values. We then apply CoLoRA to collaboratively train each subset $\{T_j^i\}_{j=1}^4$ and measure the resulting performance of the target task $T_1^i$. As shown in Figure 3 and Table 2, Table 3, performance is correlated with similarity, which is predicted in our theory.

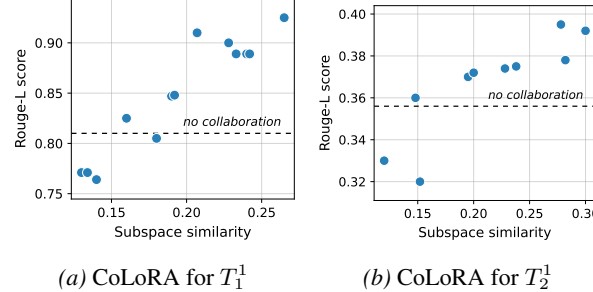

*(a)* CoLoRA for $T_1^1$     *(b)* CoLoRA for $T_2^1$

*Figure 3.* Performance of CoLoRA for a fixed task w.r.t different levels of similarity. The dashed line is the performance when only using local data.

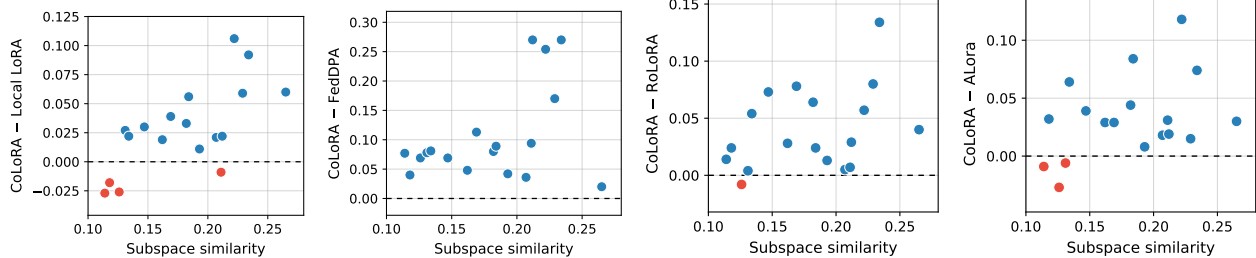

*Figure 4.* Performance difference between CoLoRA and baseline methods. For each experiment, we compute the average performance of all clients on their respective tasks and plot the difference between CoLoRA and each baseline. A positive score difference indicates superior performance of CoLoRA over the baseline.

**Baselines.** Across our experiments, we compare CoLoRA against the following baselines:

- **Local LoRA.** Each client independently fine-tunes LoRA adapters on its private data; no parameters are communicated.

- **RoLoRA** (Chen et al., 2025). It alternates between local training of the $B$ matrices followed by averaging, and local training of the $A$ matrices followed by averaging.

- **FedDPA** (Yang et al., 2024). It maintains both global $(B, A)$ and personalized $(C, D)$ adapters for each client; all adapters are trained locally, but only the global adapters are communicated and aggregated, while the personalized ones remain private.

- **ALoRA** (Ban & Ji, 2025). It train both $A$ and $B$ adapters locally, but keep the $A$ adapters private while communicating and aggregating the $B$ adapters.

- **RAVAN** (Raje et al., 2025). It uses the adapter structure $\sum_{i=1}^{h} B_i R_i A_i$, where $B_i$ and $A_i$ are frozen and only $R_i$ are aggregated across clients. The rank for RAVAN is selected such that the total number of learnable paramaters matches the other baselines.

To ensure an evaluation across varying levels of task similarity, we take task sets $\{T_j^i\}_{j=1}^4$ such that average similarity $1/6 \sum_{j<k} \mathrm{sim}_c(T_j^i, T_k^i)$ spans a wide range as $i$ varies. See the tasks in subsection E.1.

For each group of four clients, we apply CoLoRA as well as all baseline methods under identical federated training conditions. Performance is measured as the average rougle-L score across the four clients in each group. We then report, for each experiment, the performance difference between CoLoRA and the baselines. A positive score difference indicates superior performance of CoLoRA over the baseline.

Across most experimental settings, CoLoRA surpasses the baselines (see Figure 4 and Table 4). When task similarity is

very low, however, Local LoRA is superior, as anticipated.

**Experiments with 20 users**. We also conducted a large-scale experiment involving 20 similar yet distinct tasks as demonstrated in Figure 5 (RougeL score). We further provide exact matching scores in Appendix A.5.

**Experiments with Qwen2.5-3B-Instruct**. In addition to the Qwen2.5-1.5B model, we examined a larger model Qwen2.5-3B-Instruc and a higher rank $r = 16$ as shown in Figures 7 (RougeL score) and 8 (exact matching). See subsection A.4 for numerical scores.

## 6. Discussion

In this paper, we introduced CoLoRA, a collaborative fine-tuning approach with minimal parameter overhead, suitable for distributed and federated settings. CoLoRA stands on leveraging similarities in the downstream tasks. Consequently, we investigated the similarity notion, particularly in language tasks. We provided a preliminary and simple similarity metric, however, developing robust and efficient similarity measures for language tasks remains an interesting future direction.

Through experiments on federated fine-tuning of large language models across diverse tasks, we demonstrated that CoLoRA consistently outperforms existing baselines, with particularly strong gains in regimes where tasks are highly related. Finally, we provided a theoretical analysis of CoLoRA via heterogeneous linear regression. By extending techniques from the matrix sensing literature, we derived sample complexity guarantees for recovering the underlying ground truth.

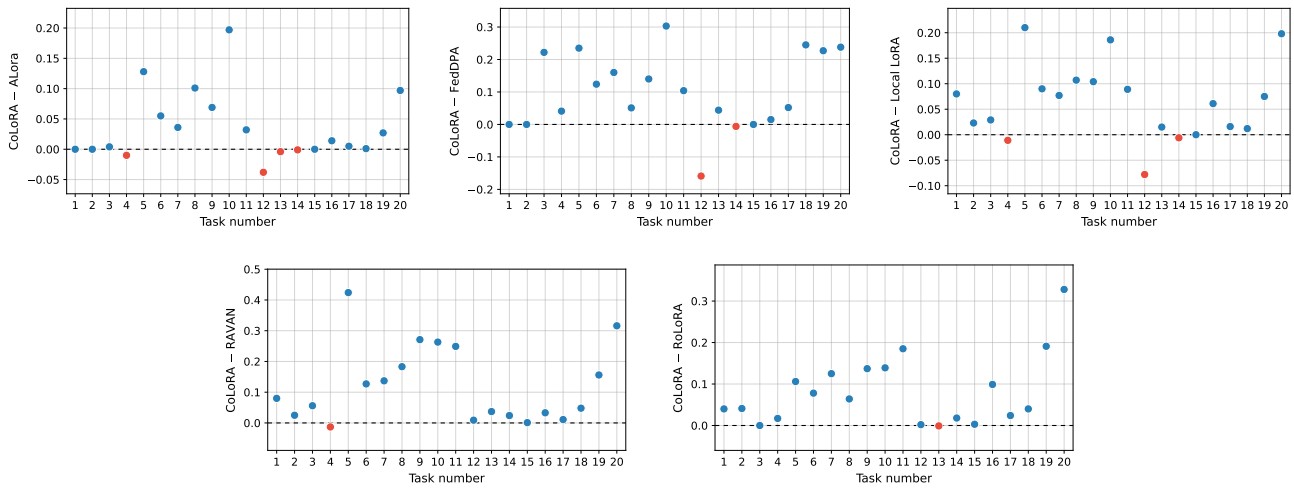

*Figure 5.* Per-task accuracy difference between CoLoRA and each baseline for **20** similar yet different tasks.

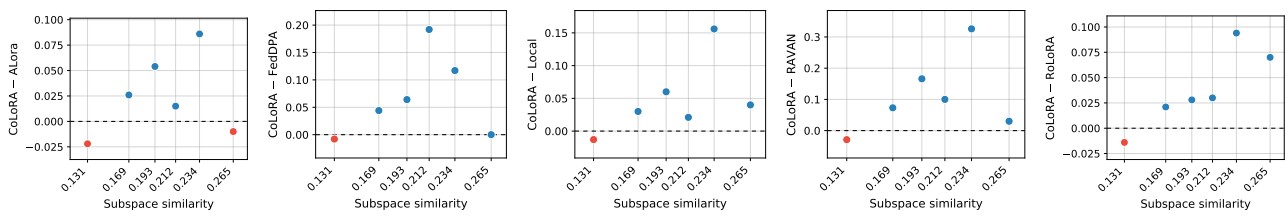

*Figure 6.* Accuracy gain of CoLoRA over each baseline as a function of subspace similarity. Blue: CoLoRA $\geq$ baseline; red: CoLoRA $\leq$ baseline. The dashed line marks zero difference. Rank $r = 16$, rougeL score, model = Qwen2.5-1.5B-Instruct

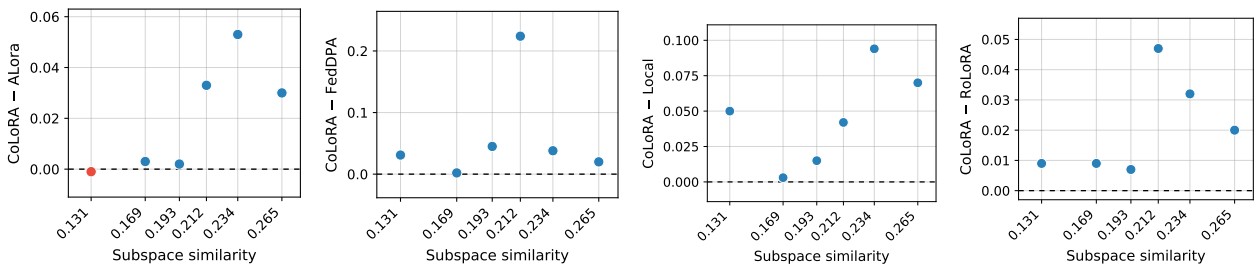

*Figure 7.* Accuracy gain of CoLoRA over each baseline as a function of subspace similarity. Blue: CoLoRA $\geq$ baseline; red: CoLoRA $\leq$ baseline. The dashed line marks zero difference. Rank $r = 4$, rougeL score, model = Qwen2.5-3B-Instruct

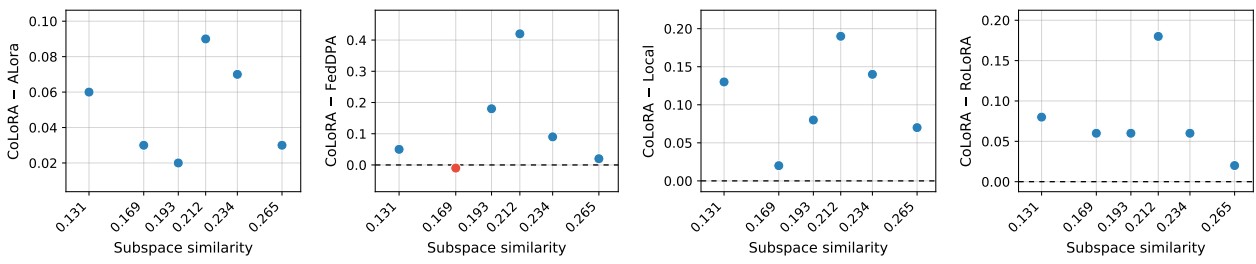

*Figure 8.* Accuracy gain of CoLoRA over each baseline as a function of subspace similarity. Blue: CoLoRA $\geq$ baseline; red: CoLoRA $\leq$ baseline. The dashed line marks zero difference. Rank $r = 4$, exact matching score, model = Qwen2.5-3B-Instruct

## Impact Statement

This work aims to advance machine learning methods in collaborative settings. While such methods may have broader societal implications, we do not identify any specific societal risks or consequences that require special discussion.

## Acknowledgements

Gagik Magakyan was supported by Aeropuertos Argentina 2000 Fellowship. The authors acknowledge the MIT SuperCloud and Lincoln Laboratory Supercomputing Center for providing computing resources that have contributed to the research results reported within this paper.

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

---

**Algorithm 2** CoLoRA-Alt: Alternating collaborative LoRA with personalized $\mathbf{\Lambda}^i$

---

1: **Input:** clients $i = 1, \ldots, k$; comm rounds $T$
2: Initialize $\boldsymbol{A}, \boldsymbol{B}, \{\boldsymbol{\Lambda}^i\}_{i=1}^k$
3: **for** $t = 0$ to $T - 1$ **do**
4:    **if** $t$ even **then**
5:      **for** each client $i = 1, \ldots, k$ in parallel **do**
6:        `// update` $\mathbf{\Lambda}^i, B$`; hold` $A$ `fixed`
7:        $(\boldsymbol{B}^i, \boldsymbol{\Lambda}^i) \leftarrow \text{TRAINLOCAL}(\boldsymbol{B}^i, \boldsymbol{\Lambda}^i;$ client $i$ data; $\boldsymbol{A}^i$ frozen$)$
8:      **end for**
9:    $\bar{\boldsymbol{B}} \leftarrow \text{AVERAGE}(\{\boldsymbol{B}^i\}_{i=1}^k);$    $\boldsymbol{B}^i \leftarrow \bar{\boldsymbol{B}}$ for all $i$
10:   **else**
11:      `// update` $\mathbf{\Lambda}^i, A$`; hold` $B$ `fixed`
12:      **for** each client $i = 1, \ldots, k$ in parallel **do**
13:        $(\boldsymbol{A}^i, \boldsymbol{\Lambda}^i) \leftarrow \text{TRAINLOCAL}(\boldsymbol{A}^i, \boldsymbol{\Lambda}^i;$ client $i$ data; $\boldsymbol{B}^i$ frozen$)$
14:      **end for**
15:    $\bar{\boldsymbol{A}} \leftarrow \text{AVERAGE}(\{\boldsymbol{A}^i\}_{i=1}^k);$    $\boldsymbol{A}^i \leftarrow \bar{\boldsymbol{A}}$ for all $i$
16:   **end if**
17: **end for**

---

# A. Appendix

## A.1. Additional related work

**Matrix sensing.** (Recht et al., 2010) introduced the Restricted Isometry Property (RIP) framework for low-rank matrix recovery and analyzed the matrix sensing problem under this assumption. They proved that if the rank-$r$ RIP constant $\delta_r \leq c$, then the solution to the trace-minimization program exactly recovers the minimum-rank matrix. Furthermore, they established that Gaussian measurement ensembles satisfy this condition with high probability when the number of measurements scales as $\Omega(dr)$. Building upon, (Jain et al., 2010) proposed a projected gradient descent algorithm that achieves similar recovery guarantees as (Recht et al., 2010). However, their method requires computing a full singular value decomposition (SVD) at each iteration, which limits its computational efficiency. (Jain et al., 2013) developed a more scalable alternating minimization approach. Their theoretical analysis requires a stronger condition, namely $\delta_{4r} \leq c/r$, which leads to a higher sample complexity of order $\Omega(dr^3)$. Later, (Tu et al., 2016) demonstrated that a simple gradient descent algorithm can achieve recovery under the milder condition $\delta_{6r} \leq c$, reducing the sample complexity back to $\Omega(dr)$ while remaining computationally efficient.

**Linear Representation Learning.** Our framework is related to the literature on multitask linear representation learning (Collins et al., 2021; Du et al., 2021; Thekumparampil et al., 2021; Collins et al., 2022; Tripuraneni et al., 2021). These works typically consider the data model

$$y_j^i = \left\langle \boldsymbol{x}_j^i, \boldsymbol{B}^* \boldsymbol{w}^{i*} \right\rangle, \quad i = 1, \ldots, k, \quad j = 1, \ldots, n, \tag{15}$$

where $k$ denotes the number of clients, $n$ the number of datapoints per client, $\boldsymbol{x}_j^i$ are the input features, $\boldsymbol{B}^* \in \mathbb{R}^{d \times r}$ is the shared representation, and $\boldsymbol{w}^i$ are client-specific linear heads. In particular, Collins et al. (2021) show that if each client has

$$n \gtrsim r^2 \left( \frac{d}{k} + r \log k \right) \log \frac{1}{\epsilon},$$

then their alternating minimization algorithm recovers $\boldsymbol{B}^*$ up to $\epsilon$ accuracy in subspace distance.

Recall that in the special case $\beta = 0$, our data model reduces to

$$y_j^i = \left\langle \boldsymbol{G}_j^i, \boldsymbol{U}^* \boldsymbol{\Lambda}^{i*} \boldsymbol{V}^{*\top} \right\rangle, \quad i = 1, \ldots, k, \quad j = 1, \ldots, n,$$

which can be rewritten in the form of (15) as follows:

$$
\begin{aligned}
y_j^i &= \left\langle \boldsymbol{G}_j^i, \boldsymbol{U}^* \boldsymbol{\Lambda}^{i^*} \boldsymbol{V}^{*\top} \right\rangle \\
&= \left\langle \mathrm{vec}\left(\boldsymbol{G}_j^i\right), \mathrm{vec}\left(\boldsymbol{U}^* \boldsymbol{\Lambda}^{i^*} \boldsymbol{V}^{*\top}\right) \right\rangle \\
&= \left\langle \mathrm{vec}\left(\boldsymbol{G}_j^i\right), \left(\boldsymbol{V}^* \otimes \boldsymbol{U}^*\right) \mathrm{vec}\left(\boldsymbol{\Lambda}^{i^*}\right) \right\rangle,
\end{aligned}
$$

where $\boldsymbol{B}^* = \boldsymbol{V}^* \otimes \boldsymbol{U}^* \in \mathbb{R}^{d^2 \times r^2}$ and $(\boldsymbol{w}^i)^* = \mathrm{vec}\left(\boldsymbol{\Lambda}^{i^*}\right) \in \mathbb{R}^{r^2}$. Hence, under this transformation, our formulation corresponds to a linear representation learning model in which the shared representation exhibits a Kronecker product structure. Directly applying existing results without accounting for this structure would lead to bounds scaling with $d^2$, which are therefore suboptimal.

**Personalized Federated Learning.** (Mishchenko et al., 2025; Pillutla et al., 2022) propose frameworks that combine global and local parameters, along with alternating optimization framework to jointly learn them. Fallah et al. (2020) adopts a meta-learning perspective, aiming to learn a global model that performs well on each local task after a single gradient update. While these methods are related to our setting, their analyses focus on general non-convex objectives and establish convergence under that regime. In contrast, we study a linear model, which allows us to derive exact sample complexity guarantees. Moreover, applying these methods directly to LoRA adapters would lead to inexact parameter averaging, typically resulting in degraded performance. Conceptually, their techniques draw from the classical federated learning literature, whereas ours are rooted in matrix sensing theory. A deeper connection between the two perspectives may exist under certain conditions, which we leave as an interesting direction for future work. Hanzely & Richtárik (2020) propose a formulation that interpolates between local and global models by introducing a regularization term. However, their approach differs from ours in two key aspects. First, they employ a quadratic regularizer, which is incompatible with our setting, as our notion of distance is based on subspace distance. Second, their theoretical analysis assumes strongly convex objectives, whereas our problem involves non-convex optimization.

**Task similarity.** Task relatedness and similarity have been extensively studied in both computer vision and natural language processing through the lens of transfer learning (Zamir et al., 2018; ?; Standley et al., 2020; Aribandi et al., 2022). These works typically either train a model on one task and fine-tune it on another to assess transferability, or jointly train the tasks and evaluate the resulting performance gains. A complementary line of research (Achille et al., 2019; Vu et al., 2020) measures task similarity via task embeddings derived from the Fisher Information Matrix (FIM). Our similarity measure is conceptually closer to this second line of work, but measures similarity based on the LoRA adapters. While prior methods compute the FIM by fine-tuning a full model, one could, in principle, apply the same approach to LoRA adapters. Nonetheless, such an extension is not invariant to the transformations $\boldsymbol{B} \leftarrow \boldsymbol{B}\boldsymbol{R},\ \boldsymbol{A} \leftarrow \boldsymbol{R}^{-1}\boldsymbol{A}$, which makes the method unsuitable in the LoRA setting.

## A.2. Details on experiments

**Optimization and hyperparameters** We mirror the defaults in our code for reproducibility. We used AdamW Optimizer (Loshchilov & Hutter, 2019) with learning rate $1.0 \times 10^{-4}$. Training batch size was 4, communication total rounds is set as 50, and we set local epochs between communications as 1. We used rouge-L and exact matching for evaluation metrics. For all the comparisons experiments, we used the same seed in the initialization to ensure consistency. All experiments were run on a single NVIDIA L40 GPU, except the large-scale experiments, which used a single NVIDIA H200 GPU.

**Similarity calculation protocol.** Following Bai et al. (2024) and Mishra et al. (2022), we do not use the full data for training adapters for reducing computational cost. For each task, we use 600 datapoints to train the task-specific LoRA adapters, starting from the same initialization. Empirically, we find that our similarity measure is robust to initialization and preserves the relative distance between tasks.

Note that the values in Figure 1 are obtained by averaging the similarity across all matrices in the pretrained model. Here, we report the similarity values for individual matrices across all layers for the $Q$ and MLP components between task pairs $(T_1, T_2)$ and $(T_3, T_4)$. We observe that for more similar tasks, the similarity tends to increase in the later layers of the network, and this trend is more pronounced for the $Q$ matrices. This suggests that the early layers capture general representations, while the later layers become increasingly task-specific. Consequently, it may be advantageous to define task similarity using only the later layers of the network—a direction we leave for future investigation.

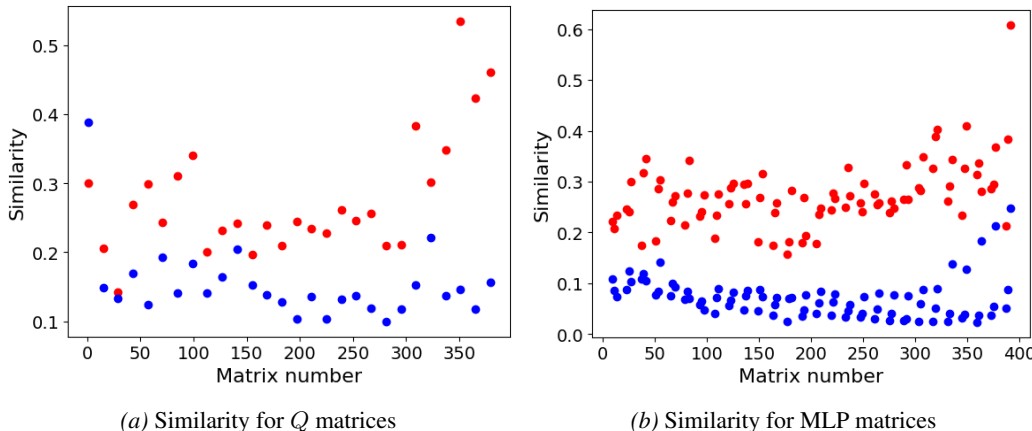

*(a) Similarity for $Q$ matrices*  *(b) Similarity for MLP matrices*

*Figure 9.* Similarity across network layers. Smaller matrix indices correspond to parameters from earlier layers in the network. Red points denote the similarity between tasks $T_1$ and $T_2$, while blue points correspond to $T_1$ and $T_3$.

## A.3. Experiment tables

*Table 2.* Similarity and performance values for $T_1^1$.

| Subspace similarity | 0.129 | 0.133 | 0.138 | 0.163 | 0.180 | 0.192 | 0.207 | 0.230 | 0.234 | 0.242 | 0.263 |
|---|---|---|---|---|---|---|---|---|---|---|---|
| Rougle-L score | 0.772 | 0.786 | 0.760 | 0.823 | 0.805 | 0.847 | 0.910 | 0.899 | 0.890 | 0.891 | 0.924 |

*Table 3.* Similarity and performance values for $T_1^2$.

| **Task 2** | | | | | | | |
|---|---|---|---|---|---|---|---|
| Subspace similarity | 0.114 | 0.136 | 0.151 | 0.196 | 0.238 | 0.280 | 0.287 | 0.301 |
| Rougle-L score | 0.333 | 0.362 | 0.322 | 0.372 | 0.373 | 0.377 | 0.399 | 0.391 |

*Table 4.* Performance comparison across varying similarity levels. The bolded values indicate the best performance for each similarity level. Oracle values denote the performance of local fine-tuning in the data-rich regime (using 600 datapoints).

| Subspace similarity | 0.114 | 0.118 | 0.126 | 0.131 | 0.134 | 0.147 | 0.162 | 0.169 | 0.182 | 0.184 | 0.193 | 0.207 | 0.211 | 0.212 | 0.222 | 0.229 | 0.234 | 0.265 |
|---|---|---|---|---|---|---|---|---|---|---|---|---|---|---|---|---|---|---|
| Local LoRA | **0.405** | **0.677** | **0.593** | 0.806 | 0.543 | 0.381 | 0.711 | 0.905 | 0.789 | 0.779 | 0.937 | 0.958 | **0.742** | 0.939 | 0.527 | 0.618 | 0.635 | 0.910 |
| FedDPA | 0.301 | 0.619 | 0.498 | 0.755 | 0.484 | 0.342 | 0.682 | 0.831 | 0.742 | 0.746 | 0.906 | 0.943 | 0.639 | 0.691 | 0.379 | 0.507 | 0.457 | 0.95 |
| RoLoRA | 0.364 | 0.635 | 0.575 | 0.829 | 0.511 | 0.338 | 0.702 | 0.866 | 0.758 | 0.811 | 0.935 | 0.974 | 0.726 | 0.932 | 0.576 | 0.597 | 0.593 | 0.930 |
| ALoRA | 0.387 | 0.627 | 0.594 | **0.839** | 0.501 | 0.372 | 0.701 | 0.915 | 0.778 | 0.751 | 0.940 | 0.961 | 0.702 | 0.942 | 0.515 | 0.662 | 0.653 | 0.940 |
| CoLoRA | 0.378 | 0.659 | 0.567 | 0.833 | **0.565** | **0.411** | **0.730** | **0.944** | **0.822** | **0.835** | **0.948** | **0.979** | 0.733 | **0.961** | **0.633** | **0.677** | **0.727** | **0.970** |
| Oracle | 0.512 | 0.763 | 0.696 | 0.908 | 0.680 | 0.580 | 0.827 | 0.982 | 0.950 | 0.936 | 0.977 | 0.981 | 0.976 | 0.993 | 0.886 | 0.893 | 0.925 | 0.987 |

## A.4 Bigger rank and larger base model experiments

*Table 5.* Rank $r = 16$, rougeL score, model = Qwen2.5-1.5B-Instruct

| Subspace similarity | 0.131 | 0.169 | 0.193 | 0.212 | 0.234 | 0.265 |
|---|---|---|---|---|---|---|
| Local | 0.782 | 0.938 | 0.919 | 0.941 | 0.657 | 0.920 |
| FedDPA | 0.777 | 0.924 | 0.915 | 0.770 | 0.696 | 0.960 |
| RoLoRA | 0.783 | 0.947 | 0.951 | 0.932 | 0.719 | 0.890 |
| ALoRA | **0.791** | 0.942 | 0.925 | 0.947 | 0.727 | **0.970** |
| RAVAN | 0.798 | 0.895 | 0.813 | 0.862 | 0.487 | 0.930 |
| CoLoRA | 0.769 | **0.968** | **0.979** | **0.962** | **0.813** | 0.960 |

*Table 6.* Rank $r = 16$, exact matching score, model = Qwen2.5-1.5B-Instruct

| Subspace similarity | 0.131 | 0.169 | 0.193 | 0.212 | 0.234 | 0.265 |
|---|---|---|---|---|---|---|
| Local | 0.620 | 0.720 | 0.690 | 0.650 | 0.380 | 0.920 |
| FedDPA | 0.630 | 0.700 | 0.700 | 0.390 | 0.400 | 0.960 |
| RoLoRA | 0.640 | 0.760 | 0.810 | 0.670 | 0.500 | 0.890 |
| ALoRA | **0.660** | 0.800 | 0.760 | 0.710 | 0.500 | **0.970** |
| RAVAN | 0.700 | 0.610 | 0.550 | 0.480 | 0.180 | 0.930 |
| CoLoRA | **0.660** | **0.870** | **0.910** | **0.780** | **0.750** | 0.960 |

*Table 7.* Rank $r = 4$, rougeL score, model = Qwen2.5-3B-Instruct

| Subspace similarity | 0.131 | 0.169 | 0.193 | 0.212 | 0.234 | 0.265 |
|---|---|---|---|---|---|---|
| Local | 0.785 | 0.973 | 0.949 | 0.922 | 0.689 | 0.880 |
| FedDPA | 0.804 | **0.974** | 0.919 | 0.740 | 0.745 | 0.930 |
| RoLoRA | 0.826 | 0.967 | 0.957 | 0.917 | 0.751 | 0.930 |
| ALoRA | **0.836** | 0.973 | 0.962 | 0.931 | 0.730 | 0.920 |
| CoLoRA | 0.835 | 0.976 | **0.964** | **0.964** | **0.783** | **0.950** |

*Table 8.* Rank $r = 4$, exact matching score, model = Qwen2.5-3B-Instruct

| Subspace similarity | 0.131 | 0.169 | 0.193 | 0.212 | 0.234 | 0.265 |
|---|---|---|---|---|---|---|
| Local | 0.640 | 0.830 | 0.830 | 0.620 | 0.580 | 0.880 |
| FedDPA | 0.720 | **0.860** | 0.730 | 0.390 | 0.630 | 0.930 |
| RoLoRA | 0.690 | 0.790 | 0.850 | 0.630 | 0.660 | 0.930 |
| ALoRA | 0.710 | 0.820 | 0.890 | 0.720 | 0.650 | 0.920 |
| CoLoRA | **0.770** | 0.850 | **0.910** | **0.810** | **0.720** | **0.950** |

## A.5 Large number of clients experiment

*Table 9.* Individual rougeL scores for the experiment with 20 clients

| Task number | T1 | T2 | T3 | T4 | T5 | T6 | T7 | T8 | T9 | T10 | T11 | T12 | T13 | T14 | T15 | T16 | T17 | T18 | T19 | T20 |
|---|---|---|---|---|---|---|---|---|---|---|---|---|---|---|---|---|---|---|---|---|
| Local LoRA | 0.920 | 0.937 | 0.962 | 0.972 | 0.786 | 0.848 | 0.787 | 0.790 | 0.853 | 0.664 | 0.876 | 0.253 | 0.917 | **0.996** | **1.000** | 0.924 | 0.972 | 0.974 | 0.376 | 0.331 |
| FedDPA | **1.000** | 0.960 | 0.769 | 0.920 | 0.761 | 0.814 | 0.704 | 0.846 | 0.817 | 0.547 | 0.861 | **0.334** | 0.888 | **0.996** | **1.000** | 0.970 | 0.936 | 0.741 | 0.224 | 0.291 |
| RoLoRA | 0.960 | 0.919 | **0.991** | 0.944 | 0.890 | 0.860 | 0.739 | 0.833 | 0.820 | 0.711 | 0.780 | 0.173 | 0.933 | 0.972 | 0.997 | 0.886 | 0.964 | 0.946 | 0.260 | 0.201 |
| ALoRA | **1.000** | 0.960 | 0.987 | 0.971 | 0.868 | 0.883 | 0.828 | 0.796 | 0.888 | 0.653 | 0.933 | 0.213 | **0.936** | 0.991 | **1.000** | 0.971 | 0.983 | 0.985 | 0.424 | 0.432 |
| RAVAN | 0.920 | 0.935 | 0.935 | **0.974** | 0.572 | 0.811 | 0.727 | 0.714 | 0.686 | 0.587 | 0.716 | 0.166 | 0.895 | 0.966 | 0.999 | 0.952 | 0.977 | 0.938 | 0.295 | 0.213 |
| CoLoRA | **1.000** | **0.960** | **0.991** | 0.961 | **0.996** | **0.938** | **0.864** | **0.897** | **0.957** | **0.850** | **0.965** | 0.175 | 0.932 | 0.990 | **1.000** | **0.985** | **0.988** | **0.986** | **0.451** | **0.529** |

*Table 10.* Individual exact matching scores for the experiment with 20 clients

| Task number | T1 | T2 | T3 | T4 | T5 | T6 | T7 | T8 | T9 | T10 | T11 | T12 | T13 | T14 | T15 | T16 | T17 | T18 | T19 | T20 |
|---|---|---|---|---|---|---|---|---|---|---|---|---|---|---|---|---|---|---|---|---|
| Local LorRA | 0.920 | 0.880 | 0.760 | **0.760** | 0.240 | 0.400 | 0.280 | 0.560 | 0.320 | 0.240 | 0.320 | 0.120 | 0.520 | **0.960** | **1.000** | **0.667** | **0.840** | 0.720 | 0.000 | 0.000 |
| FedDPA | **1.000** | 0.960 | 0.200 | 0.480 | 0.280 | **0.360** | 0.200 | **0.800** | 0.200 | 0.320 | 0.320 | **0.160** | 0.320 | **0.960** | **1.000** | 0.762 | 0.360 | 0.160 | 0.000 | 0.000 |
| RoLoRA | 0.920 | 0.760 | **0.920** | 0.600 | 0.520 | 0.560 | 0.240 | 0.640 | 0.240 | 0.320 | 0.200 | 0.040 | **0.560** | 0.800 | 0.920 | 0.571 | 0.640 | 0.560 | 0.000 | 0.000 |
| ALoRA | **1.000** | 0.960 | 0.880 | **0.760** | 0.480 | 0.480 | 0.440 | 0.560 | 0.400 | 0.160 | 0.440 | 0.080 | **0.560** | **0.960** | **1.000** | **0.667** | **0.840** | 0.840 | **0.040** | 0.000 |
| CoLoRA | **1.000** | **1.000** | **0.920** | 0.720 | **0.960** | 0.680 | **0.600** | **0.920** | **0.640** | **0.760** | **0.720** | 0.040 | **0.560** | 0.920 | **1.000** | 0.810 | **0.840** | **0.880** | **0.040** | 0.000 |

# B. Appendix

### B.1. On the subspace distance

We consider the subspace distance (Golub & Van Loan, 2013; Jain et al., 2013) defined as:

**Definition 5.** *Let* $\boldsymbol{X}_1, \boldsymbol{X}_2 \in R^{d \times r}$ *and* $\boldsymbol{Q}_1, \boldsymbol{Q}_2 \in R^{d \times r}$ *be the orthonormal basis of* $\mathrm{span}(\boldsymbol{X}_1), \mathrm{span}(\boldsymbol{X}_2)$ *respectively. The subspace distance between* $\boldsymbol{X}_1$ *and* $\boldsymbol{X}_2$ *is given by:*

$$
\begin{aligned}
dist(\boldsymbol{X}_1, \boldsymbol{X}_2) &\overset{\text{def}}{=} \left\| \left( I - \boldsymbol{Q}_1 \boldsymbol{Q}_1^\top \right) \boldsymbol{Q}_2 \right\|_2 \\
&= \left\| \left( I - \boldsymbol{Q}_2 \boldsymbol{Q}_2^\top \right) \boldsymbol{Q}_1 \right\|_2 .
\end{aligned}
$$

The subspace distance is connected to *principal angles* (Golub & Van Loan, 2013) between two subspaces as follows. If $\boldsymbol{Q}_1^\top \boldsymbol{Q}_2 = \boldsymbol{U} \boldsymbol{S} \boldsymbol{V}^\top$ is the singular value decomposition, then the singular values $\mathrm{diag}(\boldsymbol{S}) = (\cos(\boldsymbol{\theta}_1), \cdots, \cos(\boldsymbol{\theta}_r))$, where $\boldsymbol{\theta}_1 \geq \cdots \geq \boldsymbol{\theta}_r$ are called the principal angles between subspaces $\mathrm{span}(\boldsymbol{Q}_1)$ and $\mathrm{span}(\boldsymbol{Q}_2)$. One can show

$$
\mathrm{dist}(\boldsymbol{X}_1, \boldsymbol{X}_2) = \sin(\boldsymbol{\theta}_1).
$$

Recall that our subspace similarity notion defined in 11, 12 is equal to

$$
\mathrm{sim}(\boldsymbol{X}_1, \boldsymbol{X}_2) = \frac{\left\| \boldsymbol{Q}_1^\top \boldsymbol{Q}_2 \right\|_F}{\sqrt{r}} = \sqrt{\frac{\sum_{i=1}^{r} \cos(\boldsymbol{\theta}_i)^2}{r}} .
$$

Hence, we have:

$$
\mathrm{dist}(\boldsymbol{X}_1, \boldsymbol{X}_2)^2 = \sin(\boldsymbol{\theta}_1)^2 \leq \sum_{i=1}^{r} \sin(\boldsymbol{\theta}_i)^2 = r - \sum_{i=1}^{r} \cos(\boldsymbol{\theta}_i)^2 = r(1 - \mathrm{sim}(\boldsymbol{X}_1, \boldsymbol{X}_2)^2) \tag{16}
$$

### B.2. Main Theorem

For the proof of our main theorem we are going to work with subspace distance rather then subspace similarity. We define

$$
\beta := \sqrt{r} \cdot \sqrt{1 - \xi^2} \tag{17}
$$

Recall by 13, for all $i \in [k]$ we have:

$$
\mathrm{dist}(\boldsymbol{U}^*, \mathrm{col}(\boldsymbol{M}^{i^*})), \mathrm{dist}(\boldsymbol{V}^*, \mathrm{row}(\boldsymbol{M}^{i^*})) \leq \beta, \tag{18}
$$

where $\mathrm{col}(\boldsymbol{M}^{i^*}), \mathrm{row}(\boldsymbol{M}^{i^*})$ are the column and row subspaces of $\boldsymbol{M}^{i^*}$ respectively.

We define

$$
A_{N,d,r} := C_2 \frac{N + rd}{N}, \quad B_{N,d,r} = 10^4 \cdot \sqrt{r} \kappa^2 \sqrt{A_{N,d,r}},
$$

where $C_2$ is an absolute constant coming from a concentration inequality and will be specified later.

We now state the full version of the main theorem.

**Theorem 2.** *Suppose we have*

$$
\beta \leq \frac{1}{4 \cdot 10^4} \cdot \frac{1}{\sqrt{r}} \cdot \frac{1}{\max(\kappa^2, \kappa\gamma)} \cdot \frac{1}{\sqrt{A_{N,d,r}}} \tag{19}
$$

*Let $\delta_{3r}, \delta'_{2r}$ be such that:*

$$\delta_{3r} \leq \frac{1}{10^4} \cdot \frac{1}{\sqrt{r}} \min\left(\kappa^{-2}, \gamma^{-1}\right) \tag{20}$$

$$\delta'_{2r} \leq \frac{1}{5 \cdot 10^5} \cdot \frac{1}{\sqrt{r}} \kappa^{-2} \tag{21}$$

*Let $\bar{\delta}_{3r} = \max\left(\left(\frac{\delta_{3r}}{50}\right)^2, \frac{\delta_{3r}}{50\sqrt{k}}\right)$. Suppose we have*

$$N \geq \frac{2(\frac{dr}{k} + r^2)}{c\bar{\delta}_{3r}^2} \log\left(\frac{9\sqrt{r}}{\bar{\delta}_{3r}}\right) \text{ and } n \geq \frac{162r^2}{c(\delta'_{2r})^2} \log(\frac{27}{\delta'_{2r}}). \tag{22}$$

*Then, the iterates of Algorithm 1 satisfy*

$$dist(\boldsymbol{U}_T, \boldsymbol{U}^*) + dist(\boldsymbol{V}_T, \boldsymbol{V}^*) \leq \left(4 \cdot 10^4 \sqrt{r}\kappa^2(\delta_{3r} + \delta'_{2r})\right)^T + 8\beta B_{N,d,r},$$

*with probability at least:*

$$1 - C_1 exp\left(-\frac{cm\bar{\delta}_{3r}^2}{2}\right) - 2k \cdot exp\left(-dr\right) - 3C_1 \cdot Tk \cdot exp\left(-\frac{cn_2(\delta'_{2r})^2}{162}\right)$$

*Proof.* See subsection B.3 □

By the main theorem , the concentration parameters can be chosen as

$$\delta_{3r} = \frac{1}{10^5} \frac{1}{\sqrt{r}} \min\left(\kappa^{-2}, \gamma^{-1}\right), \qquad \delta'_{2r} = \frac{1}{5 \cdot 10^5} \frac{1}{\sqrt{r}} \kappa^{-2},$$

which in turn yields:

$$N \gtrsim \left(\frac{dr}{k} + r^2\right) \min\left(r^2 \max\left(\kappa^8, \gamma^4\right), rk \max\left(\kappa^4, \gamma^2\right)\right), \qquad n \gtrsim \kappa^4 r^3.$$

Hence, for $\varepsilon > 0$, we need in total

$$k(N + Tn_2) = O\left(\left(dr + kr^2\right) \cdot \min\left(r^2 \max\left(\kappa^8, \gamma^4\right), rk \max\left(\kappa^4, \gamma^2\right)\right) + kr^4 \cdot \kappa^4 \cdot \log\left(\frac{1}{\varepsilon}\right)\right)$$

datapoints to obtain

$$dist(\boldsymbol{U}_T, \boldsymbol{U}^*) + dist(\boldsymbol{V}_T, \boldsymbol{V}^*) \leq \varepsilon + 8\beta B_{N,d,r}.$$

Plugging $\beta$ from 17, we obtain the form in our informal version.

## B.3. Proof of Theorem 2

We first introduced notations that will be frequently used throughout the proof.

For a matrix $\boldsymbol{A} \in R^{m \times n}$, we denote by $vec(\boldsymbol{A}) \in R^{mn}$ its vectorization obtained by concatenating its columns. Conversely, for a vector $\boldsymbol{v} \in R^{mn}$, we write $mat(\boldsymbol{v}) \in R^{m \times n}$ for the matrix such that $vec(mat(\boldsymbol{v})) = \boldsymbol{v}$.

We use $\|\boldsymbol{A}\|_F$ and $\|\boldsymbol{A}\|_2$ to denote the Frobenius and spectral norms, respectively. Note that $\|vec(\boldsymbol{A})\|_2 = \|\boldsymbol{A}\|_F$. The singular values of $\boldsymbol{A}$ are written as $\sigma_1(\boldsymbol{A}) \geq \sigma_2(\boldsymbol{A}) \geq \cdots$, with $\sigma_{\min}(\boldsymbol{A}), \sigma_{\max}(\boldsymbol{A})$ denoting the smallest and largest ones respectively. We have the condition number is $\kappa(\boldsymbol{A}) = \frac{\sigma_{\max}(\boldsymbol{A})}{\sigma_{\min}(\boldsymbol{A})}$.

We will repeatedly use the inequalities

$$\sigma_{min}(\boldsymbol{A}) \|\boldsymbol{B}\|_F \leq \|\boldsymbol{A}\boldsymbol{B}\|_F \leq \|\boldsymbol{A}\|_2 \|\boldsymbol{B}\|_F,$$
$$|\sigma_i(\boldsymbol{X}) - \sigma_i(\boldsymbol{Y})| \leq \sigma_{\max}(\boldsymbol{X} - \boldsymbol{Y}),$$

second of which is called Weil's inequality. For matrices $\boldsymbol{A}, \boldsymbol{B} \in R^{m \times n}$, their inner product is

$$\langle \boldsymbol{A}, \boldsymbol{B} \rangle = \langle \text{vec}\,(\boldsymbol{A}), \text{vec}\,(\boldsymbol{B}) \rangle = \text{tr}\,(\boldsymbol{A}^\top \boldsymbol{B}).$$

We denote by $\text{B}(r, n)$ the Euclidean ball of radius $r$ in $R^n$, and by $\text{St}(d, r)$ the Stiefel manifold, i.e. the set of orthonormal matrices in $R^{d \times r}$. $\text{col}(\boldsymbol{X})$ and $\text{row}(\boldsymbol{X})$ mean the column and row subspaces of the matrix $\boldsymbol{X}$, respectively.

We make frequent use of Kronecker products, relying on the identities

$$(\boldsymbol{A} \otimes \boldsymbol{B}) \text{vec}\,(\boldsymbol{C}) = \text{vec}\,(\boldsymbol{BCA}^\top), \tag{23}$$

$$(\boldsymbol{A} \otimes \boldsymbol{B})^\top = \boldsymbol{A}^\top \otimes \boldsymbol{B}^\top. \tag{24}$$

Finally, for $\boldsymbol{A} \in R^{m \times r_1}$ and $\boldsymbol{B} \in R^{m \times r_2}$, we use $[\boldsymbol{A}, \boldsymbol{B}]$ to denote their concatenation across column axis.

WLOG, we can assume that $\boldsymbol{U}^*, \boldsymbol{V}^* \in \text{St}(d, r)$. Since the column and row spaces of $(\boldsymbol{M}^i)^*$ are close to those of $\boldsymbol{U}^*$ and $\boldsymbol{V}^*$ in subspace distance, one can choose bases representing these spaces that are also close to $\boldsymbol{U}^*$ and $\boldsymbol{V}^*$ in spectral norm. The following lemma formalizes this observation.

**Lemma 1.** *There is a factorization*

$$\boldsymbol{M}^{i*} = \boldsymbol{U}^{i*} \boldsymbol{\Lambda}^{i*} \boldsymbol{V}^{i*\top} \text{ for } i = 1, \cdots, k$$

*where $\boldsymbol{U}^{i*}, \boldsymbol{V}^{i*} \in R^{d \times r}, \boldsymbol{\Lambda}^{i*} \in R^{r \times r}$ satisfy*

$$\left\| \boldsymbol{U}^* - \boldsymbol{U}^{i*} \right\|_2, \left\| \boldsymbol{V}^* - \boldsymbol{V}^{i*} \right\|_2 \leq \beta. \tag{25}$$

$$\max_{i=1}^k \kappa\left(\boldsymbol{\Lambda}^{i*}\right) \leq \sqrt{2} \max_{i=1}^k \kappa\left(\boldsymbol{M}^{i*}\right) = \sqrt{2}\kappa. \tag{26}$$

*Proof.* See subsection D.1 $\qquad\qquad\qquad\qquad\qquad\qquad\qquad\qquad\qquad\qquad\qquad\qquad\qquad\quad$ $\square$

We slightly abuse notation by reusing $\boldsymbol{U}^{i*}$ and $\boldsymbol{V}^{i*}$. These should not be confused with the matrices appearing in Definition 2, where they denote arbitrary orthonormal bases. Here, $\boldsymbol{U}^{i*}$ and $\boldsymbol{V}^{i*}$ refer to specific bases chosen to be close to the shared representations $\boldsymbol{U}^*$ and $\boldsymbol{V}^*$, respectively, in spectral distance.

First, we want to separate out the concentration arguments from the main body of the proof to have a more readable and user-friendly proof. Let $T$ be the number of iterations of the algorithm.

From Proposition 1 we have that $\{\boldsymbol{G}_j^i\}$ satisfies 3r-GRIP with coefficient $\delta_{3r}$ with probability at least

$$1 - C_1 \exp\left(-\frac{cm\bar{\delta}_{3r}^2}{2}\right) \tag{27}$$

Additionally, from Lemma 14, we have that $\{\boldsymbol{G}_j^i\}$ is sub-isometric (see Definition 7) with coefficient $A_{N,d,r}$, with probability at least

$$1 - 2k \cdot \exp\left(-dr\right) \tag{28}$$

For each iteration we also need concentration inequalities for client-specific $\boldsymbol{\Lambda}$ minimization part. For that we define a notion of $(\boldsymbol{U}, \boldsymbol{V})$-RIP in Definition 6. Define

$$\begin{aligned}
\widetilde{\boldsymbol{U}}_{0,t} &= \boldsymbol{U}_t, & \widetilde{\boldsymbol{V}}_{0,t} &= \boldsymbol{V}_t, \\
\widetilde{\boldsymbol{U}}_{1,t} &= \left[\boldsymbol{U}_t, \boldsymbol{U}_t \boldsymbol{U}_t^\top \boldsymbol{U}^{i*}\right], & \widetilde{\boldsymbol{V}}_{1,t} &= \left[\boldsymbol{V}_t, \boldsymbol{V}^t \boldsymbol{V}_t^\top \boldsymbol{V}^{i*} - \boldsymbol{V}^{i*}\right], \\
\widetilde{\boldsymbol{U}}_{2,t} &= \left[\boldsymbol{U}_t, \boldsymbol{U}_t \boldsymbol{U}_t^\top \boldsymbol{U}^{i*} - \boldsymbol{U}^{i*}\right], & \widetilde{\boldsymbol{V}}_{2,t} &= \left[\boldsymbol{V}_t, \boldsymbol{V}^{i*}\right],
\end{aligned}$$

where $[\boldsymbol{A}, \boldsymbol{B}]$ is concatenation across column axis.

The next lemma shows that across all iterations the "local" concentrations happen with high probability.

**Lemma 2.** *With probability at least:*

$$1 - 3C \cdot Tk \cdot exp\left(-\frac{cn_2(\delta'_{2r})^2}{162}\right) \tag{29}$$

*the ensemble $\{\boldsymbol{G}^i_{j,t}\}^n_{j=1}$ satisfies $(\widetilde{\boldsymbol{U}}_{\ell,t}, \widetilde{\boldsymbol{V}}_{\ell,t})$-RIP with coefficient $\delta'_{2r}$ for all $\ell = 0, 1, 2, i = 1, \cdots, k$ and $t = 0, 1, \cdots T-1$.*

*Proof.* Since $(\boldsymbol{U}_t, \boldsymbol{V}_t)$ depends only on $\{\boldsymbol{G}^i_j\}$ and $\{\boldsymbol{G}^i_{j,s}\}$ for $s \leq t-1$, it follows that $\{\boldsymbol{G}^i_{j,t}\}$ is independent of $(\boldsymbol{U}_t, \boldsymbol{V}_t)$. Therefore, for a fixed $\ell, i, t$, from Proposition 5 we have that with probability at least:

$$1 - 3C \cdot \exp\left(-\frac{nc(\delta'_{2r})^2)}{162}\right),$$

the ensemble $\{\boldsymbol{G}^i_{j,t}\}^n_{j=1}$ satisfies $(\widetilde{\boldsymbol{U}}_{\ell,t}, \widetilde{\boldsymbol{V}}_{\ell,t}$-RIP. Applying union bound completes the proof. □

Finally, using union bound we obtain that with probability at least:

$$1 - C\exp\left(-\frac{cm\bar{\delta}^2_{3r}}{2}\right) - Ck\exp\left(-dr\right) - 3C \cdot Tk \cdot \exp\left(-\frac{nc(\delta'_{2r})^2)}{162}\right)$$

we have that:

$$\{\boldsymbol{G}^i_j\} \text{ satisfies 3r-GRIP with coefficient } \delta_{3r} \tag{30}$$
$$\text{and}$$
$$\{\boldsymbol{G}^i_j\} \text{ is sub-isometric with coefficient } A_{N,d,r} \tag{31}$$
$$\text{and}$$
$$\{\boldsymbol{G}^i_{j,t}\}^n_{j=1} \text{ satisfies } (\widetilde{\boldsymbol{U}}_{\ell,t}, \widetilde{\boldsymbol{V}}_{\ell,t}\text{-RIP with coefficient } \delta'_{2r}$$
$$\text{for all } \ell \in \{0, 1, 2\}, i \in \{1, \cdots, k\} \text{ and } t \in \{0, 1, \cdots T-1\}. \tag{32}$$

**From now on we work on this high-probability event.**

We are going to inductively prove the following proposition which will complete the proof of the main theorem:

**Proposition 2.** *Given the initialization from Lemma 3, for all $t = 0, 1, \cdots T$ the following inequalities hold:*

$$\left\|\boldsymbol{\Lambda}^i_{t+1} - \boldsymbol{U}_t^\top \boldsymbol{U}^{i*} \boldsymbol{\Lambda}^{i*} \boldsymbol{V}^{i*\top} \boldsymbol{V}_t\right\|_2 \leq 72\delta'_{2r} \left\|\boldsymbol{\Lambda}^{i*}\right\|_F \left(dist(\boldsymbol{U}_t, \boldsymbol{U}^{i*}) + dist(\boldsymbol{V}_t, \boldsymbol{V}^{i*})\right) \tag{33}$$

$$dist(\boldsymbol{V}_{t+1}, \boldsymbol{V}^*) \leq 2 \cdot 10^4 \sqrt{r}\kappa^2(\delta_{3r} + \delta'_{2r})\left(dist(\boldsymbol{U}_t, \boldsymbol{U}^*) + dist(\boldsymbol{V}_t, \boldsymbol{V}^*)\right) + 2\beta B_{N,d,r} \tag{34}$$

$$dist(\boldsymbol{U}_{t+1}, \boldsymbol{U}^*) \leq 2 \cdot 10^4 \sqrt{r}\kappa^2(\delta_{3r} + \delta'_{2r})\left(dist(\boldsymbol{U}_t, \boldsymbol{U}^*) + dist(\boldsymbol{V}_t, \boldsymbol{V}^*)\right) + 2\beta B_{N,d,r} \tag{35}$$

### B.3.1. INITIALIZATION

We follow the standard initialization used in the matrix sensing literature, which is equivalent performing one step of projected gradient descent from 0 initialization Jain et al. (2010) .

**Lemma 3.** *Denote the estimator*

$$\widehat{\boldsymbol{M}} := \frac{1}{m}\sum^k_{i=1}\sum^N_{j=1}\boldsymbol{y}^i_j\boldsymbol{G}^i_j,$$

*and let $\boldsymbol{U}_0\boldsymbol{\Lambda}_0\boldsymbol{V}_0^\top = \widehat{\boldsymbol{M}}_r$ be the rank-$r$ SVD of $\boldsymbol{M}$. Then, the following inequality holds:*

$$dist(\boldsymbol{U}_0, \boldsymbol{U}^*), dist(\boldsymbol{V}_0, \boldsymbol{V}^*) \leq \frac{1}{20\kappa},$$

*Proof.* See subsection D.4 □

B.3.2. ANALYSIS OF $\boldsymbol{\Lambda}$ MINIMIZATION

In this section, we analyze line 4 in Algorithm 1. Note, because of linear invariance, we cannot aim for a bound $\left\| \boldsymbol{\Lambda}_t^i - \boldsymbol{\Lambda}^{i*} \right\|_2$ and therefore we give a bound relative to our estimates of the ground-truth factors, as captured in the next proposition.

**Proposition 3.** *The following inequality holds:*

$$\left\| \boldsymbol{\Lambda}_{t+1}^i - \boldsymbol{U}_t^\top \boldsymbol{U}^{i*} \boldsymbol{\Lambda}^{i*} \boldsymbol{V}^{i*\top} \boldsymbol{V}_t \right\|_2 \leq 72\delta_{2r}' \left\| \boldsymbol{\Lambda}^{i*} \right\|_F \left( dist(\boldsymbol{U}_t, \boldsymbol{U}^{i*}) + dist(\boldsymbol{V}_t, \boldsymbol{V}^{i*}) \right).$$

In this section, we will prove Proposition 3 and some additional helper results.

Writing down the definition of the loss function we get:

$$l_t^i(\boldsymbol{U}_t, \boldsymbol{V}_t, \boldsymbol{\Lambda}^i) = \sum_{j=1}^n \left( \boldsymbol{y}_{j,t}^i - \left\langle \boldsymbol{G}_{j,t}^i, \boldsymbol{U}_t \boldsymbol{\Lambda}^i \boldsymbol{V}_t^\top \right\rangle \right)^2 = \sum_{j=1}^n \left( \left\langle \boldsymbol{G}_{j,t}^i, \boldsymbol{U}^{i*} \boldsymbol{\Lambda}^{i*} \boldsymbol{V}^{i*\top} \right\rangle - \left\langle \boldsymbol{G}_{j,t}^i, \boldsymbol{U}_t \boldsymbol{\Lambda}^i (\boldsymbol{V}_t)^\top \right\rangle \right)^2$$

We modify the inner products to get vectorized objects as follows:

$$\left\langle \boldsymbol{G}_{j,t}^i, \boldsymbol{U}^{i*} \boldsymbol{\Lambda}^{i*} (\boldsymbol{V}^{i*})^\top \right\rangle = \left\langle \mathrm{vec}\left(\boldsymbol{G}_{j,t}^i\right), \mathrm{vec}\left(\boldsymbol{U}^{i*} \boldsymbol{\Lambda}^{i*} (\boldsymbol{V}^{i*})^\top\right) \right\rangle$$
$$= \left\langle \mathrm{vec}\left(\boldsymbol{G}_{j,t}^i\right), \left(\boldsymbol{V}^{i*} \otimes \boldsymbol{U}^{i*}\right) \mathrm{vec}\left(\boldsymbol{\Lambda}^{i*}\right) \right\rangle$$
$$= \mathrm{vec}\left(\boldsymbol{G}_{j,t}^i\right)^\top \left(\boldsymbol{V}^{i*} \otimes \boldsymbol{U}^{i*}\right) \mathrm{vec}\left(\boldsymbol{\Lambda}^{i*}\right)$$

Similarly, we have:

$$\left\langle \boldsymbol{G}_{j,t}^i, \boldsymbol{U}_t \boldsymbol{\Lambda}^i (\boldsymbol{V}_t)^\top \right\rangle = \mathrm{vec}\left(\boldsymbol{G}_{j,t}^i\right)^\top (\boldsymbol{V}_t \otimes \boldsymbol{U}_t) \mathrm{vec}\left(\boldsymbol{\Lambda}^i\right)$$

Since $\boldsymbol{\Lambda}_{t+1}^i$ minimizes the quadratic loss function, writing down the first-order condition

$$0 = \nabla_{\boldsymbol{\Lambda}^i} l_t^i(\boldsymbol{U}_t, \boldsymbol{V}_t, \boldsymbol{\Lambda}_{t+1}^i) = \nabla_{\boldsymbol{\Lambda}^i} \sum_{j=1}^n \left( \mathrm{vec}\left(\boldsymbol{G}_{j,t}^i\right)^\top (\boldsymbol{V}_t \otimes \boldsymbol{U}_t) \mathrm{vec}\left(\boldsymbol{\Lambda}^i\right) - \mathrm{vec}\left(\boldsymbol{G}_{j,t}^i\right)^\top \left(\boldsymbol{V}^{i*} \otimes \boldsymbol{U}^{i*}\right) \mathrm{vec}\left(\boldsymbol{\Lambda}^{i*}\right) \right)^2,$$

gives us:

$$\sum_{j=1}^n \left( \mathrm{vec}\left(\boldsymbol{G}_{j,t}^i\right)^\top (\boldsymbol{V}_t \otimes \boldsymbol{U}_t) \right)^\top \left( \mathrm{vec}\left(\boldsymbol{G}_{j,t}^i\right)^\top (\boldsymbol{V}_t \otimes \boldsymbol{U}_t) \mathrm{vec}\left(\boldsymbol{\Lambda}^i\right) - \mathrm{vec}\left(\boldsymbol{G}_{j,t}^i\right)^\top \left(\boldsymbol{V}^{i*} \otimes \boldsymbol{U}^{i*}\right) \mathrm{vec}\left(\boldsymbol{\Lambda}^{i*}\right) \right) = 0$$

Therefore, with the notations:

$$\boldsymbol{B}_t = \sum_{j=1}^n (\boldsymbol{V}_t)^\top \otimes (\boldsymbol{U}_t)^\top \mathrm{vec}\left(\boldsymbol{G}_{j,t}^i\right) \mathrm{vec}\left(\boldsymbol{G}_{j,t}^i\right)^\top \boldsymbol{V}_t \otimes \boldsymbol{U}_t,$$

$$\boldsymbol{C}_t = \sum_{j=1}^n (\boldsymbol{V}_t)^\top \otimes (\boldsymbol{U}_t)^\top \mathrm{vec}\left(\boldsymbol{G}_{j,t}^i\right) \mathrm{vec}\left(\boldsymbol{G}_{j,t}^i\right)^\top \boldsymbol{V}^{i*} \otimes \boldsymbol{U}^{i*},$$

we obtain:

$$\boldsymbol{B}_t \mathrm{vec}\left(\boldsymbol{\Lambda}_{t+1}^i\right) = \boldsymbol{C}_t \mathrm{vec}\left((\boldsymbol{\Lambda}^i)^*\right).$$

Hence, the vectorized $\boldsymbol{\Lambda}_{t+1}^i$ satisfies the equation:

$$\mathrm{vec}\left(\boldsymbol{\Lambda}_{t+1}^i\right) = (\boldsymbol{B}_t)^{-1} \boldsymbol{C}_t \mathrm{vec}\left(\boldsymbol{\Lambda}^*\right)$$
$$= ((\boldsymbol{V}_t)^\top \boldsymbol{V}^{i*}) \otimes ((\boldsymbol{U}_t)^\top \boldsymbol{U}^{i*}) \mathrm{vec}\left(\boldsymbol{\Lambda}^{i*}\right)$$
$$- (\boldsymbol{B}_t)^{-1} \left( \boldsymbol{B}_t ((\boldsymbol{V}_t)^\top \boldsymbol{V}^{i*}) \otimes ((\boldsymbol{U}_t)^\top \boldsymbol{U}^{i*}) - \boldsymbol{C}_t \right) \mathrm{vec}\left(\boldsymbol{\Lambda}^{i*}\right),$$

where the second line is just an algebraic manipulation. From this we obtain the desired difference we want to bound:

$$\text{vec}\left(\boldsymbol{\Lambda}_{t+1}^{i}\right) - \left((\boldsymbol{V}_t)^\top \boldsymbol{V}^{i*}\right) \otimes \left((\boldsymbol{U}_t)^\top \boldsymbol{U}^{i*}\right)\text{vec}\left(\boldsymbol{\Lambda}^{i*}\right)$$
$$= (\boldsymbol{B}_t)^{-1}\left(\boldsymbol{B}_t((\boldsymbol{V}_t)^\top \boldsymbol{V}^{i*}) \otimes ((\boldsymbol{U}_t)^\top \boldsymbol{U}^{i*}) - \boldsymbol{C}_t\right)\text{vec}\left(\boldsymbol{\Lambda}^{i*}\right).$$

In order to bound the $\ell_2$ norm of LHS, we will bound the following terms:

$$\left\|(\boldsymbol{B}_t)^{-1}\right\|_2, \left\|\left(\boldsymbol{B}_t((\boldsymbol{V}_t)^\top \boldsymbol{V}^{i*}) \otimes ((\boldsymbol{U}_t)^\top \boldsymbol{U}^{i*}) - \boldsymbol{C}_t\right)\text{vec}\left(\boldsymbol{\Lambda}^{i*}\right)\right\|_2 \tag{36}$$

**For the ease of notation, we are going to remove the subscript $t$ from our variables**.

We proceed proving upper bounds for our desired terms in 36 with the following two lemmas:

**Lemma 4.** *We have that:*

$$\left\|\boldsymbol{B}^{-1}\right\|_2 \leq \frac{1}{(1 - \delta_{2r}')n}.$$

*Proof.* See subsection D.5. $\square$

**Lemma 5.** *We have that:*

$$\left\|\left(\boldsymbol{B}(\boldsymbol{V}^\top \boldsymbol{V}^{i*}) \otimes (\boldsymbol{U}^\top \boldsymbol{U}^{i*}) - \boldsymbol{C}\right)\text{vec}\left(\boldsymbol{\Lambda}^{i*}\right)\right\|_2 \leq 72\delta_{2r}'n\left\|\boldsymbol{\Lambda}^{i*}\right\|_F\left(\text{dist}(\boldsymbol{U}, \boldsymbol{U}^{i*}) + \text{dist}(\boldsymbol{V}, \boldsymbol{V}^{i*})\right)$$

*Proof.* See subsection D.6. $\square$

Combining Lemma 4 and Lemma 5 gives us:

$$\left\|\boldsymbol{\Lambda}^i - \boldsymbol{U}^\top \boldsymbol{U}^* \boldsymbol{\Lambda}^{i*}(\boldsymbol{V}^{i*})^\top \boldsymbol{V}\right\|_2 \leq \left\|\boldsymbol{\Lambda}^i - \boldsymbol{U}^\top \boldsymbol{U}^* \boldsymbol{\Lambda}^{i*}(\boldsymbol{V}^{i*})^\top \boldsymbol{V}\right\|_F = \left\|\text{vec}\left(\boldsymbol{\Lambda}^i - \boldsymbol{U}^\top \boldsymbol{U}^* \boldsymbol{\Lambda}^{i*}(\boldsymbol{V}^{i*})^\top \boldsymbol{V}\right)\right\|_F$$
$$= \left\|\text{vec}\left(\boldsymbol{\Lambda}^i\right) - (\boldsymbol{V}^\top \boldsymbol{V}^{i*}) \otimes (\boldsymbol{U}^\top \boldsymbol{U}^{i*})\text{vec}\left(\boldsymbol{\Lambda}^{i*}\right)\right\|_2$$
$$\leq \left\|\boldsymbol{B}^{-1}\right\|_2\left\|\left(\boldsymbol{B}(\boldsymbol{V}^\top \boldsymbol{V}^{i*}) \otimes (\boldsymbol{U}^\top \boldsymbol{U}^{i*}) - \boldsymbol{C}\right)\text{vec}\left(\boldsymbol{\Lambda}^{i*}\right)\right\|_2$$
$$\leq \frac{72\delta_{2r}'}{1 - \delta_{2r}'}\left\|\boldsymbol{\Lambda}^{i*}\right\|_F\left(\text{dist}(\boldsymbol{U}, \boldsymbol{U}^{i*}) + \text{dist}(\boldsymbol{V}, \boldsymbol{V}^{i*})\right)$$
$$\leq 144\delta_{2r}'\left\|\boldsymbol{\Lambda}^{i*}\right\|_F\left(\text{dist}(\boldsymbol{U}, \boldsymbol{U}^{i*}) + \text{dist}(\boldsymbol{V}, \boldsymbol{V}^{i*})\right). \tag{37}$$

This complete the proof of Proposition 3.

For the subsequent analysis, we need bounds on $\sigma_{\min}(\boldsymbol{\Lambda}_t^i)$ and $\sigma_{\max}(\boldsymbol{\Lambda}_t^i)$, which we provide in the following lemma:

**Lemma 6.** *For $i = 1, \cdots, r$, the following inequalities holds:*

$$\frac{1}{2}\sigma_i(\boldsymbol{\Lambda}^{i*}) \leq \sigma_i(\boldsymbol{\Lambda}_t^i) \leq 2\sigma_i(\boldsymbol{\Lambda}^{i*}) \tag{38}$$

$$\frac{1}{2}\left\|\boldsymbol{\Lambda}^{i*}\right\|_F \leq \left\|\boldsymbol{\Lambda}_t^i\right\|_F \leq 2\left\|\boldsymbol{\Lambda}^{i*}\right\|_F. \tag{39}$$

*Proof.* See subsection D.7. $\square$

B.3.3. ANALYSIS OF $U, V$ MINIMIZATION

We continue analyzing lines 6-9 in Algorithm 1.

**Proposition 4.** *The following inequalities hold:*

$$dist(\boldsymbol{V}_{t+1}, \boldsymbol{V}^*) \leq 2 \cdot 10^4 \sqrt{r}\kappa^2(\delta_{3r} + \delta'_{2r}) \left(dist(\boldsymbol{U}_t, \boldsymbol{U}^*) + dist(\boldsymbol{V}_t, \boldsymbol{V}^*)\right) + 2\beta B_{N,d,r}, \tag{40}$$
$$dist(\boldsymbol{U}_{t+1}, \boldsymbol{U}^*) \leq 2 \cdot 10^4 \sqrt{r}\kappa^2(\delta_{3r} + \delta'_{2r}) \left(dist(\boldsymbol{U}_t, \boldsymbol{U}^*) + dist(\boldsymbol{V}_t, \boldsymbol{V}^*)\right) + 2\beta B_{N,d,r} \tag{41}$$

Recall that the loss function is defined as follows:

$$\sum_{i=1}^{k} l^i(\boldsymbol{U}_t, \boldsymbol{V}, \boldsymbol{\Lambda}_{t+1}^i) = \sum_{i=1}^{k}\sum_{j=1}^{N} \left(\boldsymbol{y}_j^i - \left\langle \boldsymbol{G}_j^i, \boldsymbol{U}_t\boldsymbol{\Lambda}_{t+1}^i(\boldsymbol{V})^\top \right\rangle\right)^2$$
$$= \sum_{i=1}^{k}\sum_{j=1}^{N} \left(\left\langle \boldsymbol{G}_j^i, \boldsymbol{U}^{i*}\boldsymbol{\Lambda}^{i*}(\boldsymbol{V}^{i*})^\top \right\rangle - \left\langle \boldsymbol{G}_j^i, \boldsymbol{U}_t\boldsymbol{\Lambda}_{t+1}^i(\boldsymbol{V})^\top \right\rangle\right)^2$$

As before, we write down the inner product in terms of the vectorized objects as follows:

$$\left\langle \boldsymbol{G}_j^i, \boldsymbol{U}_t\boldsymbol{\Lambda}_{t+1}^i(\boldsymbol{V})^\top \right\rangle = \left\langle (\boldsymbol{G}_j^i)^\top, \boldsymbol{V}\left(\boldsymbol{U}_t\boldsymbol{\Lambda}_{t+1}^i\right)^\top \right\rangle$$
$$= \left\langle \text{vec}\left((\boldsymbol{G}_j^i)^\top\right), \text{vec}\left(\boldsymbol{V}\left(\boldsymbol{U}_t\boldsymbol{\Lambda}_{t+1}^i\right)^\top\right) \right\rangle$$
$$= \text{vec}\left((\boldsymbol{G}_j^i)^\top\right)^\top \text{vec}\left(\boldsymbol{V}\left(\boldsymbol{U}_t\boldsymbol{\Lambda}_{t+1}^i\right)^\top\right)$$
$$= \text{vec}\left((\boldsymbol{G}_j^i)^\top\right)^\top \left((\boldsymbol{U}_t\boldsymbol{\Lambda}_{t+1}^i) \otimes \boldsymbol{I}_d\right) \text{vec}\left(\boldsymbol{V}\right).$$

Similarly, we have:

$$\left\langle \boldsymbol{G}_j^i, \boldsymbol{U}^{i*}\boldsymbol{\Lambda}^{i*}(\boldsymbol{V}^{i*})^\top \right\rangle = \text{vec}\left((\boldsymbol{G}_j^i)^\top\right)^\top \left((\boldsymbol{U}^{i*}\boldsymbol{\Lambda}^{i*}) \otimes \boldsymbol{I}_d\right) \text{vec}\left(\boldsymbol{V}^{i*}\right).$$

Now, since $\boldsymbol{V}_{t+1}$ minimizes the quadratic loss function, writing down the first-order condition

$$0 = \nabla_{\text{vec}(\boldsymbol{V})} l^i(\boldsymbol{U}_t, \boldsymbol{V}, \boldsymbol{\Lambda}_{t+1}^i) = \nabla_{\text{vec}(\boldsymbol{V})} \sum_{i=1}^{k}\sum_{j=1}^{N} \left(\text{vec}\left((\boldsymbol{G}_j^i)^\top\right)^\top \left((\boldsymbol{U}^{i*}\boldsymbol{\Lambda}^{i*}) \otimes \boldsymbol{I}_d\right) \text{vec}\left(\boldsymbol{V}^{i*}\right)\right.$$
$$\left. - \text{vec}\left((\boldsymbol{G}_j^i)^\top\right)^\top \left((\boldsymbol{U}_t\boldsymbol{\Lambda}_{t+1}^i) \otimes \boldsymbol{I}_d\right) \text{vec}\left(\boldsymbol{V}\right)\right)^2,$$

gives us:

$$0 = \sum_{i=1}^{k}\sum_{j=1}^{N} \left(\text{vec}\left((\boldsymbol{G}_j^i)^\top\right)^\top \left((\boldsymbol{U}_t\boldsymbol{\Lambda}_{t+1}^i) \otimes \boldsymbol{I}_d\right)\right)^\top \left(\text{vec}\left((\boldsymbol{G}_j^i)^\top\right)^\top \left((\boldsymbol{U}^{i*}\boldsymbol{\Lambda}^{i*}) \otimes \boldsymbol{I}_d\right) \text{vec}\left(\boldsymbol{V}^{i*}\right)\right.$$
$$\left. - \text{vec}\left((\boldsymbol{G}_j^i)^\top\right)^\top \left((\boldsymbol{U}_t\boldsymbol{\Lambda}_{t+1}^i) \otimes \boldsymbol{I}_d\right) \text{vec}\left(\boldsymbol{V}\right)\right)$$

Therefore, with the notations:

$$\boldsymbol{B}_t^i = \sum_{j=1}^{N} (\boldsymbol{U}_t\boldsymbol{\Lambda}_t^i)^\top \otimes \boldsymbol{I}_d \cdot \text{vec}\left((\boldsymbol{G}_j^i)^\top\right) \text{vec}\left((\boldsymbol{G}_j^i)^\top\right)^\top \cdot (\boldsymbol{U}_t\boldsymbol{\Lambda}_{t+1}^i) \otimes \boldsymbol{I}_d,$$

$$\boldsymbol{C}_t^i = \sum_{j=1}^{N} (\boldsymbol{U}_t\boldsymbol{\Lambda}_t^i)^\top \otimes \boldsymbol{I}_d \cdot \text{vec}\left((\boldsymbol{G}_j^i)^\top\right) \text{vec}\left((\boldsymbol{G}_j^i)^\top\right)^\top \cdot (\boldsymbol{U}^{i*}(\boldsymbol{\Lambda}^i)^*) \otimes \boldsymbol{I}_d,$$

we get:

$$\left(\sum_{i=1}^{k} \boldsymbol{B}_t^i\right) \text{vec}\left(\widehat{\boldsymbol{V}}_{t+1}\right) = \left(\sum_{i=1}^{k} \boldsymbol{C}_t^i\right) \text{vec}\left(\boldsymbol{V}^{i*}\right), \tag{42}$$

Let $\boldsymbol{D}_t = (\boldsymbol{V}^*)^\top \boldsymbol{V}_t$. Using our induction statement we have:

$$\begin{aligned}
\text{dist}(\boldsymbol{V}_t, \boldsymbol{V}^*) &\leq \text{dist}(\boldsymbol{V}_t, \boldsymbol{V}^*) + \text{dist}(\boldsymbol{U}_t, \boldsymbol{U}^*) \\
&\leq \left(4 \cdot 10^3 \sqrt{r}\kappa^2(\delta_{3r} + \delta'_{2r})\right)^t \left(\text{dist}(\boldsymbol{V}^0, \boldsymbol{V}^*) + \text{dist}(\boldsymbol{U}^0, \boldsymbol{U}^*)\right) + 8\beta B_{N,d,r} \\
&\overset{(\xi_1)}{\leq} \left(\text{dist}(\boldsymbol{V}^0, \boldsymbol{V}^*) + \text{dist}(\boldsymbol{U}^0, \boldsymbol{U}^*)\right) + 8\beta B_{N,d,r} \\
&\overset{(\xi_2)}{\leq} \frac{1}{2},
\end{aligned}$$

where $(\xi_1)$ follows using 20, 21 and $(\xi_2)$ from Lemma 3 and 19. Now, using Lemma 15 we obtain:

$$\left\|(\boldsymbol{D}_t)^{-1}\right\|_2 \leq 2. \tag{43}$$

**For the ease of notation, from now on we will drop the dependence on $t$.**

Note, vectorized $\widehat{\boldsymbol{V}}$ satisfies the equation:

$$\begin{aligned}
\text{vec}(\widehat{\boldsymbol{V}}) &= \left(\sum_{i=1}^{k} \boldsymbol{B}^i\right)^{-1} \left(\sum_{i=1}^{k} \boldsymbol{C}^i\right) \text{vec}(\boldsymbol{V}^*) \\
&= \left(\sum_{i=1}^{k} \boldsymbol{B}^i\right)^{-1} \left(\sum_{i=1}^{k} \boldsymbol{B}^i \cdot \boldsymbol{D}^{-1} \otimes \boldsymbol{I}_d - \sum_{i=1}^{k} \boldsymbol{B}^i \cdot \boldsymbol{D}^{-1} \otimes \boldsymbol{I}_d + \sum_{i=1}^{k} \boldsymbol{C}^i\right) \text{vec}\left(\boldsymbol{V}^*\right) \\
&= \boldsymbol{D}^{-1} \otimes \boldsymbol{I}_d \cdot \text{vec}\left(\boldsymbol{V}^*\right) - \left(\sum_{i=1}^{k} \boldsymbol{B}^i\right)^{-1} \left(\sum_{i=1}^{k} \boldsymbol{B}^i \cdot \boldsymbol{D}^{-1} \otimes \boldsymbol{I}_d - \sum_{i=1}^{k} \boldsymbol{C}^i\right) \text{vec}\left(\boldsymbol{V}^*\right) \\
&= \boldsymbol{D}^{-1} \otimes \boldsymbol{I}_d \cdot \text{vec}\left(\boldsymbol{V}^*\right) - \boldsymbol{H} \tag{44}
\end{aligned}$$

where we defined:

$$\boldsymbol{H} := \left(\sum_{i=1}^{k} \boldsymbol{B}^i\right)^{-1} \left(\sum_{i=1}^{k} \boldsymbol{B}^i \cdot \boldsymbol{D}^{-1} \otimes \boldsymbol{I}_d - \sum_{i=1}^{k} \boldsymbol{C}^i\right) \text{vec}\left(\boldsymbol{V}^*\right).$$

Next, we will show that $\|\boldsymbol{H}\|_2$ is small. First, we will upper bound $\left\|\left(\sum_{i=1}^{k} \boldsymbol{B}^i\right)^{-1}\right\|_2$ with the following lemma:

**Lemma 7.** *The following bound holds:*

$$\left\|\left(\sum_{i=1}^{k} \boldsymbol{B}^i\right)^{-1}\right\|_2 \leq \frac{8}{m} \cdot \frac{1}{\min_{i=1}^{k} \sigma_{\min}(\boldsymbol{\Lambda}^{i*})^2}.$$

*Proof.* See subsection D.8 □

Now, we will upper-bound our second desired term:

$$\left\|\left(\sum_{i=1}^{k} \boldsymbol{B}^i \cdot \boldsymbol{D}^{-1} \otimes \boldsymbol{I}_d - \sum_{i=1}^{k} \boldsymbol{C}^i\right) \text{vec}\left(\boldsymbol{V}^*\right)\right\|_2.$$

Using the definitions of $\boldsymbol{B}^i$ and $\boldsymbol{C}^i$, we further decompose it into two parts as follows:

$$
\left( \sum_{i=1}^{k} \boldsymbol{B}^i \cdot \boldsymbol{D}^{-1} \otimes \boldsymbol{I}_d - \sum_{i=1}^{k} \boldsymbol{C}^i \right) \text{vec}\left(\boldsymbol{V}^*\right)
$$

$$
= \sum_{i=1}^{k} \sum_{j=1}^{N} (\boldsymbol{U}\boldsymbol{\Lambda}^i)^\top \otimes \boldsymbol{I}_d \cdot \text{vec}\left(\boldsymbol{G}_j^i\right) \text{vec}\left(\boldsymbol{G}_j^i\right)^\top \cdot \left( (\boldsymbol{U}\boldsymbol{\Lambda}^i \boldsymbol{D}^{-1}) \otimes \boldsymbol{I}_d - \boldsymbol{U}^{i*}\boldsymbol{\Lambda}^{i*} \otimes \boldsymbol{I}_d \right) \text{vec}\left(\boldsymbol{V}^*\right)
$$

$$
= \underbrace{\sum_{i=1}^{k} \sum_{j=1}^{N} (\boldsymbol{U}\boldsymbol{\Lambda}^i)^\top \otimes \boldsymbol{I}_d \cdot \text{vec}\left(\boldsymbol{G}_j^i\right) \text{vec}\left(\boldsymbol{G}_j^i\right)^\top \cdot \left( (\boldsymbol{U}\boldsymbol{\Lambda}^i \boldsymbol{D}^{-1}) \otimes \boldsymbol{I}_d - \boldsymbol{U}\boldsymbol{U}^\top \boldsymbol{U}^{i*}\boldsymbol{\Lambda}^{i*} \otimes \boldsymbol{I}_d \right) \text{vec}\left(\boldsymbol{V}^*\right)}_{\boldsymbol{F}_1}
$$

$$
+ \underbrace{\sum_{i=1}^{k} \sum_{j=1}^{N} (\boldsymbol{U}\boldsymbol{\Lambda}^i)^\top \otimes \boldsymbol{I}_d \cdot \text{vec}\left(\boldsymbol{G}_j^i\right) \text{vec}\left(\boldsymbol{G}_j^i\right)^\top \cdot \left( \boldsymbol{U}\boldsymbol{U}^\top \boldsymbol{U}^{i*}\boldsymbol{\Lambda}^{i*} \otimes \boldsymbol{I}_d - \boldsymbol{U}^{i*}\boldsymbol{\Lambda}^{i*} \otimes \boldsymbol{I}_d \right) \text{vec}\left(\boldsymbol{V}^*\right)}_{\boldsymbol{F}_2}
$$

First, we bound $\|\boldsymbol{F}_1\|$.

**Lemma 8.** *The following bound holds:*

$$
\|\boldsymbol{F}_1\|_2 \leq 1152m \cdot \max_{i=1}^{k} \left( \left\|\boldsymbol{\Lambda}^{i*}\right\|_F, \left\|\boldsymbol{\Lambda}^{i*}\right\|_2 \right) \cdot \left( \delta'_{2r} \left( dist(\boldsymbol{U}, \boldsymbol{U}^*) + dist(\boldsymbol{V}, \boldsymbol{V}^*) \right) + \beta\sqrt{A_{N,d,r}} \right)
$$

*Proof.* See subsection D.9. $\qquad\square$

Now, we bound $\|\boldsymbol{F}_2\|_2$.

**Lemma 9.** *The following bound holds:*

$$
\|\boldsymbol{F}_2\|_2 \leq 36m \cdot \max_{i=1}^{k} \left( \left\|\boldsymbol{\Lambda}^{i*}\right\|_2 \left\|\boldsymbol{\Lambda}^{i*}\right\|_F \right) \cdot \left( \delta_{3r} dist(\boldsymbol{U}, \boldsymbol{U}^*) + \beta\sqrt{A_{N,d,r}} \right)
$$

*Proof.* See subsection D.10 $\qquad\square$

Using our results we bound $\|\boldsymbol{H}\|_2$ as follows:

$$
\|\boldsymbol{H}\|_2 = \left\| \left( \sum_{i=1}^{k} \boldsymbol{B}^i \right)^{-1} \left( \sum_{i=1}^{k} \boldsymbol{B}^i \cdot \boldsymbol{D}^{-1} \otimes \boldsymbol{I}_d - \sum_{i=1}^{k} \boldsymbol{C}^i \right) \text{vec}\left(\boldsymbol{V}^*\right) \right\|_2
$$

$$
\leq \left\| \left( \sum_{i=1}^{k} \boldsymbol{B}^i \right)^{-1} \right\|_2 \left\| \left( \sum_{i=1}^{k} \boldsymbol{B}^i \cdot \boldsymbol{D}^{-1} \otimes \boldsymbol{I}_d - \sum_{i=1}^{k} \boldsymbol{C}^i \right) \text{vec}\left(\boldsymbol{V}^*\right) \right\|_2
$$

$$
\leq \left\| \left( \sum_{i=1}^{k} \boldsymbol{B}^i \right)^{-1} \right\|_2 \left( \|\boldsymbol{F}_1\|_2 + \|\boldsymbol{F}_2\|_2 \right).
$$

Combigning Lemma 7, Lemma 8, Lemma 9, gives us:

$$\|\boldsymbol{H}\|_2 \leq 8 \frac{\max_{i=1}^k \left(\|\boldsymbol{\Lambda}^{i*}\|_2 \|\boldsymbol{\Lambda}^{i*}\|_F\right)}{\min_{i=1}^k \sigma_{min}(\boldsymbol{\Lambda}^{i*})^2} \left((1152\delta'_{2r} + 36\delta_{3r})\,\mathrm{dist}(\boldsymbol{U}, \boldsymbol{U}^*) + 1152\delta'_{2r}\mathrm{dist}(\boldsymbol{V}, \boldsymbol{V}^*) + 600\beta\sqrt{A_{N,d,r}}\right)$$

$$\leq 16\sqrt{r}\kappa^2 \left((1152\delta'_{2r} + 36\delta_{3r})\,\mathrm{dist}(\boldsymbol{U}, \boldsymbol{U}^*) + 576\delta'_{2r}\mathrm{dist}(\boldsymbol{V}, \boldsymbol{V}^*) + 600\beta\sqrt{A_{N,d,r}}\right)$$

$$\leq 10^4\sqrt{r}\kappa^2(\delta_{3r} + \delta'_{2r})\left(\mathrm{dist}(\boldsymbol{U}, \boldsymbol{U}^*) + \mathrm{dist}(\boldsymbol{V}, \boldsymbol{V}^*)\right) + \beta B_{N,d,r} \tag{45}$$

$$\tag{46}$$

where second follows from Lemma 16. To avoid confusion about the iterates, we reintroduce the superscript $t$ for $\widehat{V}_{t+1}$ and $V_{t+1}$. Remember that from 44 we have:

$$\mathrm{vec}\left(\widehat{V}_{t+1}\right) = (\boldsymbol{D}_t)^{-1} \otimes \boldsymbol{I}_d \cdot \mathrm{vec}\left(\boldsymbol{V}^*\right) - \boldsymbol{H}_t,$$

from which we obtain:

$$\widehat{V}_{t+1} = \boldsymbol{V}^*((\boldsymbol{D}_t)^{-1})^\top - \mathrm{mat}(\boldsymbol{H}_t).$$

Since $V_{t+1}$ is obtained from $\widehat{V}_{t+1}$ by QR factorization, there exists a matrix $\boldsymbol{R}_{t+1} \in \mathbb{R}^{r \times r}$ such that $\widehat{V}_{t+1} = V_{t+1}\boldsymbol{R}_{t+1}$. Let $\boldsymbol{V}_\perp^*$ be a matrix with columns forming an orthonormal basis for $\mathrm{span}(V_{t+1})^\perp$. Recalling 49, we can bound the distance as follows:

$$\mathrm{dist}(V_{t+1}, \boldsymbol{V}^*) = \left\|(\boldsymbol{V}_\perp^*)^\top V_{t+1}\right\|_2 = \left\|(\boldsymbol{V}_\perp^*)^\top \widehat{V}_{t+1}(\boldsymbol{R}_{t+1})^{-1}\right\|_2$$

$$= \left\|(\boldsymbol{V}_\perp^*)^\top \left(\boldsymbol{V}^*((\boldsymbol{D}_t)^{-1})^\top - \mathrm{mat}(\boldsymbol{H}_t)\right)(\boldsymbol{R}_{t+1})^{-1}\right\|_2$$

$$= \left\|(\boldsymbol{V}_\perp^*)^\top \mathrm{mat}(\boldsymbol{H}^t)(\boldsymbol{R}_t)^{-1}\right\|_2 \leq \|\mathrm{mat}(\boldsymbol{H}_t)\|_2 \left\|(\boldsymbol{R}_{t+1})^{-1}\right\|_2. \tag{47}$$

For bounding the first term, using 46, we get:

$$\|\mathrm{mat}(\boldsymbol{H}_t)\|_2 \leq \|\mathrm{mat}(\boldsymbol{H}_t)\|_F = \|\boldsymbol{H}_t\|_2 \leq 10^3\sqrt{r}\kappa^2(\delta_{3r} + \delta'_{2r})\left(\mathrm{dist}(\boldsymbol{U}_t, \boldsymbol{U}^*) + \mathrm{dist}(\boldsymbol{V}_t, \boldsymbol{V}^*)\right) + \beta B_{N,d,r}. \tag{48}$$

To complete the argument, it remains to bound $\left\|(\boldsymbol{R}_{t+1})^{-1}\right\|$. The next lemma provides such a bound.

**Lemma 10.** *The following inequality holds:*

$$\sigma_{\min}(\boldsymbol{R}_{t+1}) \geq 1 - \|mat(\boldsymbol{H}_t)\|_2.$$

*Proof.* See subsection D.11 □

Finnaly, combining 48, 47 and Lemma 10 gives us:

$$\mathrm{dist}(V_{t+1}, \boldsymbol{V}^*) \leq \frac{10^4\sqrt{r}\kappa^2(\delta_{3r} + \delta'_{2r})\left(\mathrm{dist}(\boldsymbol{U}_t, \boldsymbol{U}^*) + \mathrm{dist}(\boldsymbol{V}_t, \boldsymbol{V}^*)\right) + \beta B_{N,d,r}}{1 - 10^4\sqrt{r}\kappa^2(\delta_{3r} + \delta'_{2r})\left(\mathrm{dist}(\boldsymbol{U}_t, \boldsymbol{U}^*) + \mathrm{dist}(\boldsymbol{V}_t, \boldsymbol{V}^*)\right) - \beta B_{N,d,r}}$$

$$\leq 2 \cdot 10^4\sqrt{r}\kappa^2(\delta_{3r} + \delta'_{2r})\left(\mathrm{dist}(\boldsymbol{U}_t, \boldsymbol{U}^*) + \mathrm{dist}(\boldsymbol{V}_t, \boldsymbol{V}^*)\right) + 2\beta B_{N,d,r},$$

where for the second inequality we lower bounded the denominator using Lemma 3 and inequalities 19, 20, 21.

Similarly, we have:

$$\mathrm{dist}(\boldsymbol{U}_{t+1}, \boldsymbol{U}^*) \leq 2 \cdot 10^4\sqrt{r}\kappa^2(\delta_{3r} + \delta'_{2r})\left(\mathrm{dist}(\boldsymbol{U}_t, \boldsymbol{U}^*) + \mathrm{dist}(\boldsymbol{V}^t, \boldsymbol{V}^*)\right) + 2\beta B_{N,d,r},$$

and therefore the proof is completed.

**B.4. Proof of Corollary 1**

By triangle inequality we have:

$$\left\| U_t \Lambda^i_{t+1} V_t^\top - M^{i*} \right\|_2 \leq \left\| U_t \Lambda^i_{t+1} V_t^\top - U_t U_t^\top U^{i*} \Lambda^{i*} V^{i*^\top} V_t V_t^\top \right\|_2$$
$$+ \left\| U_t U_t^\top U^{i*} \Lambda^{i*} V^{i*^\top} V_t V_t^\top - U^{i*} \Lambda^{i*} V^{i*^\top} \right\|_2.$$

For the first part, using Proposition 3 we obtain:

$$\left\| U_t \Lambda^i_{t+1} V_t^\top - U_t U_t^\top U^{i*} \Lambda^{i*} V^{i*^\top} V_t V_t^\top \right\|_2 \left\| \Lambda^i_{t+1} - U_t^\top U^{i*} \Lambda^{i*} V^{i*^\top} V_t \right\|_2$$
$$\leq 144 \delta'_{2r} \left\| \Lambda^{i*} \right\|_F \left( \text{dist}(U_t, U^{i*}) + \text{dist}(V_t, V^{i*}) \right)$$
$$= 144 \delta'_{2r} \left\| M^{i*} \right\|_F \left( \text{dist}(U_t, U^{i*}) + \text{dist}(V_t, V^{i*}) \right).$$

For the second part, we have:

$$\left\| U_t U_t^\top U^{i*} \Lambda^{i*} V^{i*^\top} V_t V_t^\top - U^{i*} \Lambda^{i*} V^{i*^\top} \right\|_2 = \left\| U_t U_t^\top U^{i*} \Lambda^{i*} V^{i*^\top} V_t V_t^\top - U^{i*} \Lambda^{i*} V^{i*^\top} V_t V_t^\top \right\|_2$$
$$+ \left\| U^{i*} \Lambda^{i*} V^{i*^\top} V_t V_t^\top - U^{i*} \Lambda^{i*} V^{i*^\top} \right\|_2$$
$$\leq \left\| (U_t U_t^\top - I) U^{i*} \right\|_2 \left\| \Lambda^{i*} \right\|_2 \left\| V^{i*^\top} V_t V_t^\top \right\|_2$$
$$+ \left\| U^{i*} \Lambda^{i*} V^{i*^\top} V_t V_t^\top - U^{i*} \Lambda^{i*} V^{i*^\top} \right\|_2$$

For the first term, we obtain:

$$\left\| (U_t U_t^\top - I) U^{i*} \right\|_2 \left\| \Lambda^{i*} \right\|_2 \left\| V^{i*^\top} V_t V_t^\top \right\|_2 \leq 4 \left\| M^{i*} \right\|_F \text{dist}(U_t, U^{i*}).$$

Similarly, we have the analogue bound for the second term. Combining both and using 21, gives us:

$$\left\| U_t \Lambda^i_{t+1} V_t^\top - M^{i*} \right\|_2 \leq 5 \left\| M^{i*} \right\|_F \left( \text{dist}(U_t, U^{i*}) + \text{dist}(V_t, V^{i*}) \right)$$
$$\leq 5 \left\| M^{i*} \right\|_F \left( \text{dist}(U_t, U^*) + \text{dist}(V_t, V^*) + \text{dist}(U^{i*}, U^*) + \text{dist}(V^{i*}, V^*) \right)$$
$$\leq 5 \left\| M^{i*} \right\|_F \left( \text{dist}(U_t, U^*) + \text{dist}(V_t, V^*) + 2\beta \right).$$

Plugging in the value of $\beta$ completes the proof.

**B.5. Removing logarithmic factor**

Now, we focus on the case where $k = O(1)$; for simplicity, take $k = 1$. Recall that in the general setting, we needed the chunk $\{(G^i_{j,t}, y^i_{j,t})\}^n_{j=1}$ to ensure independence from $(\widetilde{U}^t_l, \widetilde{V}^t_l)$, which was required in establishing the local concentration result 32. However, when there is only a single client, the GRIP condition 30 already guarantees that the sample set $\{G_j\}^N_{j=1}$ satisfies the $(U, V)$-RIP for all $U, V \in \mathbb{R}^{d \times r}$. Consequently, we can update the factor $\Lambda$ directly on the large subset without needing the chunks.

In this case, setting $\beta = 0$, the resulting sample complexity reduces to

$$O(\kappa^4 dr^2),$$

which improves upon the rate established by Jain et al. (2013) by a factor of $r$.

# C. Appendix

## C.1. On the subspace distance

The subspace distance above can appear in different forms in the literature. Let $X_1^\perp, X_2^\perp$ be orthonormal basis for $\mathrm{span}(Q_1)^\perp, \mathrm{span}(Q_2)^\perp$. Note that since $(I - Q_1 Q_1^\top)$ is the projection matrix to $\mathrm{span}(Q_1)^\perp$ then we have $(I - Q_1 Q_1^\top) = Q_1 Q_1^\perp$ and therefore

$$\left\| \left( Q_1 Q_1^\top - I \right) Q_2 \right\|_2 = \left\| Q_1^\perp (Q_1^\perp)^\top Q_2 \right\|_2 = \left\| (Q_1^\perp)^\top Q_2 \right\|_2 . \tag{49}$$

It is also equal to the distance of projection matrices:

$$\mathrm{dist}(X_1, X_2) = \left\| Q_1 (Q_1)^\top - Q_2 (Q_2)^\top \right\|_2 = \left\| P_1 - P_2 \right\|_2 .$$

Hence, it satisfies the distance properties such as triangle inequality.

## C.2. Classical Gaussian Concentration

We are using the following classical concentration result several times in the manuscript.

**Lemma 11.** *Let $n$ be a positive integer. Suppose $G_1, \ldots, G_n$ are i.i.d and for every $i$, the entroes of $J_i$ are i.i.d from 0 mean sub-Gaussian distribution. Then, for fixed matrices $X_1, \ldots, X_n \in R^{d \times d}$ and $t \in (0, 1)$, we have:*

$$\mathbf{P} \left[ \left| \sum_{i=1}^{n} \langle G_i, X_i \rangle^2 - \sum_{i=1}^{n} \| X_i \|_F^2 \right| \geq nt \max_{i=1}^{n} \| X_i \|_F^2 \right] \leq C_1 e^{-cnt^2},$$

*where $C_1, c$ are constants depending on the sub-Gaussian parameter.*

*Proof.* See (Wainwright, 2019). □

## C.3. (U,V)-RIP

Beyond the GRIP concentration that controls the updates of shared $U$ and $V$, we also need concentration bounds specific to local parameters $\Lambda^i$. This motivates the following definition of $(U, V)$-RIP:

**Definition 6.** *Suppose we have fixed matrices $U, V \in R^{d \times r}$. Given the ensemble $\{G_j\}_{j=1}^n$, we say it satisfies $(U, V)$-RIP with coefficient $\delta_r$, if for any $\Lambda \in R^{r \times r}$, we have the following:*

$$\left| \sum_{j=1}^{n} \langle G_j, U \Lambda V^\top \rangle^2 - \sum_{j=1}^{n} \left\| U \Lambda V^\top \right\|_F^2 \right| \leq n \delta_r \left\| U \Lambda V^\top \right\|_F^2 . \tag{50}$$

This definition can be transformed to standard RIP condition in $R^{r \times r}$. Write $U = \bar{U} R_1, V = \bar{V} R_2$, where $\bar{U}, \bar{V}$ are orthonormal. For simplicity, assume $G_j$ are i.i.d standard Gaussian. Then

$$\langle G_j, U \Lambda (V)^\top \rangle = \mathrm{tr} \left( G_j^\top U \Lambda (V)^\top \right) = \mathrm{tr} \left( G_j^\top \bar{U} R_1 \Lambda R_2^\top \bar{V}^\top \right)$$
$$= \mathrm{tr} \left( \bar{V}^\top G_j^\top \bar{U} R_1 \Lambda R_2^\top \right) = \langle \bar{U}^\top G_j \bar{V}, R_1 \Lambda R_2^\top \rangle ,$$

and

$$\left\| U \Lambda V^\top \right\|_F = \left\| \bar{U} R_1 \Lambda (R_2)^\top \bar{V}^\top \right\|_F = \left\| R_1 \Lambda R_2^\top \right\|_F .$$

Now, define $\bar{G}_j := \bar{U}^\top G_j \bar{V} \in R^{r \times r}$, which are i.i.d standard Gaussian by orthogonal invariance, and $\bar{\Lambda} := R_1 \Lambda R_2^\top \in R^{r \times r}$. Then the condition 50 reduces to

$$\left| \sum_{j=1}^{n} \langle \bar{G}_j, \bar{\Lambda} \rangle^2 - \sum_{j=1}^{n} \left\| \bar{\Lambda} \right\|_F^2 \right| \leq n \delta_r \left\| \bar{\Lambda} \right\|_F^2 ,$$

which is exactly the standard RIP in $R^{r \times r}$ (see Definition 3).

For completeness, we state a proposition, saying that a sufficiently large ensemble of matrices satisfies the $(U, V)$-RIP.

**Proposition 5.** *Let $U, V \in \mathbb{R}^{d \times r}$ be fixed matrices. Suppose we have the random ensemble $\{G_i\}_{i=1}^n$, where for each $i$, the entries of $G_i$ are i.i.d from a 0-mean sub-Gaussian distribution. Then, if*

$$n \geq \frac{162r^2 \log(\frac{27}{\delta_r})}{\delta_r^2}.$$

*the ensemble $\{G_j\}_{j=1}^n$ satisfies $(U, V)$-RIP with coefficient $\delta_r$ with probability at least*

$$1 - C_1 exp\left(-\frac{cn\delta_r^2}{162}\right).$$

*Proof.* The proof is similar and motivated from (Candès & Plan, 2011). See subsection D.14 □

### C.4. Inner products

A very natural interpretation of the RIP condition is that the linear map

$$\mathcal{A}(X) = \begin{pmatrix} \frac{1}{\sqrt{n}} \langle G_1, X \rangle \\ \frac{1}{\sqrt{n}} \langle G_2, X \rangle \\ \vdots \\ \frac{1}{\sqrt{n}} \langle G_n, X \rangle \end{pmatrix},$$

is nearly an isometry ((Recht et al., 2010)). Hence, it preserves inner products up to some error((Jain et al., 2013) Lemma B.1). We state and prove inner-product preservation lemmas for our GRIP condition.

**Lemma 12.** *Let $\{G_j^i\}$ be an ensemble of matrices indexed by $i = 1, \ldots, k$ and $j = 1, \ldots, n$, satisfying the $3r$-GRIP (Generalized Restricted Isometry Property) with constant $\delta_{3r}$. Then, for any collection of matrices $U_\ell, V_\ell$ and $\{\Lambda_\ell^i\}_{i=1}^k$ for $\ell = 1, 2, 3$, define*

$$X^i = U_1 \Lambda_1^i V_1^\top, \quad Y^i = U_2 \Lambda_2^i V_2^\top + U_3 \Lambda_3^i V_3^\top.$$

*Then the following inequality holds:*

$$\left| \sum_{i=1}^k \sum_{j=1}^n \langle G_j^i, X^i \rangle \langle G_j^i, Y^i \rangle - \sum_{i=1}^k \langle X^i, Y^i \rangle \right| \leq 18m\delta_{3r} \max_{i=1,\ldots,k} \|X^i\|_F \|Y^i\|_F.$$

*Proof.* See subsection D.13 □

**Lemma 13.** *Suppose we have matrices $U_1, U_2, V_1, V_2 \in R^{d \times r}$. Assume that the ensemble $\{G_i\}_{i=1}^n$ satisfies $(U, V)$-RIP with coefficient $\delta_{2r}$, where $U, V$ are defined as follows:*

$$U = [U_1, U_2], V = [V_1, V_2].$$

*Now, given $\Lambda_1, \Lambda_2 \in R^{r \times r}$, define $X = U_1 \Lambda_1 (V_1)^\top$ and $Y = U_2 \Lambda_2 (V_2)^\top$. Than, the following inequality holds:*

$$\left| \sum_{i=1}^n \langle G_i, X \rangle \langle G_i, Y \rangle - \sum_{i=1}^n \langle X, Y \rangle \right| \leq 18n\delta_{2r} \|X\|_F^2 \|Y\|_F^2$$

*Proof.* The proof is a simplified version of subsection D.13. □

### C.5. Sub-isometric ensemble

We define a notion of a sub-isometric ensemble.

**Definition 7.** *The random ensemble $\{G_j^i\}$, indexed by $j = 1, \cdots, n$ and $i = 1, \cdots, k$ and $G_j^i \in R^{d \times d}$ is sub-isometric with coefficient $A_{d,r,n}$, if for matrices $X^1, \cdots, X^k$ of at most rank $r$, the following inequality holds:*

$$\sum_{i=1}^k \sum_{j=1}^n \langle G_j^i, X^i \rangle^2 \leq m A_{n,d,r} \max_{i=1}^k \|X^i\|_F^2.$$

Compared to the generalized RIP condition in Definition 4, the sub-isometric property imposes only an upper bound without assuming any shared low-rank factorization of the form

$$X^i = U \Lambda_i V^\top,$$

and thus applies more generally to arbitrary low-rank matrices.

The following proposition will help us to give a high-probability bound on the sub-isometric coefficient $A_{d,r,n}$.

**Proposition 6.** *Suppose $\{G_i\}_{i=1}^n$ with $G_i \in R^{d \times d}$ is an ensemble of i.i.d Gaussian matrices. Then, with probability at least,*

$$1 - 2e^{-dr},$$

*for all $X \in R^{d \times d}$ of at most rank $r$ we have that*

$$\sum_{i=1}^n \langle G_i, X \rangle^2 \le C_2 (n + \sqrt{nrd} + rd) \|X\|_F^2.$$

*Proof.* The proof follows from **Remark 9.1.4** in (**?**) using that the gaussian width

$$w(T) \lesssim \sqrt{rd},$$

where $T = \{X \in R^{d \times d} : \mathrm{rank}(X) \le r, \|X\|_F \le 1\}$ □

**Lemma 14.** *The random ensemble $\{G_j^i\}$ is sub-isometric with coefficient*

$$A_{n,d,r} := C_2 \frac{n + \sqrt{nrd} + rd}{n},$$

*with probability at least*

$$1 - 2k e^{-dr}.$$

*Proof.* The proof follows from Proposition 6 and applying union bound. □

## D. appendix

### D.1. Proof of Lemma 1

We first state a lemma which will be used several times in the proof of the main theorem.

**Lemma 15.** *Suppose $dist(V_1, V_2) < 1$. Than, $(V_2^\top V_1)^{-1}$ exists, and the following inequality holds:*

$$\left\| (V_2^\top V_1)^{-1} \right\|_2 \le \frac{1}{\sqrt{1 - dist(V_1, V_2)}}.$$

*Proof.* See subsection D.2. □

Let

$$M^{i*} = (\bar{U}^{i*})^* (\bar{\Lambda}^i)^* (\bar{V}^i)^{*\top}$$

be the SVD of $M^{i*}$.

Denote $(\bar{R}^i)^* := (\bar{U}^i)^{*\top} U^*$. Since

$$\mathrm{dist}((\bar{U}^i)^*, U^*) = \mathrm{dist}(\mathrm{col}\left(M^{i*}\right), U^*) = \beta < 1,$$

Lemma 15 implies that $(\bar{R}^i)^*$ is invertible and

$$\left\|\left((\bar{R}^i)^*\right)^{-1}\right\|_2 \leq \frac{1}{\sqrt{1 - \text{dist}((\bar{U}^i)^*, U^*)}} \leq \frac{1}{\sqrt{1-\beta}} \leq 2^{\frac{1}{4}}, \tag{51}$$

where the last inequality follows from 19. Note that:

$$\left\|(\bar{U}^i)^*(\bar{R}^i)^* - U^*\right\|_2 = \left\|(\bar{U}^i)^*(\bar{U}^i)^{*\top}U^* - U^*\right\|_2 = \text{dist}(U^*, (\bar{U}^i)^*) \leq \beta. \tag{52}$$

Similarly, we define $(\bar{T}^i)^* := (\bar{V}^i)^{*\top}V^*$. Hence, under the transformation

$$\boldsymbol{M}^{i*} = (\bar{U}^i)^*(\bar{\Lambda}^i)^*(\bar{V}^i)^{*\top} = \underbrace{(\bar{U}^i)^*(\bar{R}^i)^*}_{:=U^{i*}}\underbrace{(\bar{R}^i)^{*-1}(\bar{\Lambda}^i)^*\left((\bar{T}^i)^{*\top}\right)^{-1}}_{:=\Lambda^{i*}}\underbrace{((\bar{V}^i)^*(\bar{T}^i)^*)^\top}_{:=(V^{i*})^\top}$$

$$= U^{i*}\Lambda^{i*}V^{i*\top},$$

we have that

$$\left\|U^* - U^{i*}\right\|_2, \left\|V^* - V^{i*}\right\|_2 \leq \beta. \tag{53}$$

Now we state and proof a lemma which will give our condition number bound.

**Lemma 16.** *The following inequalities holds:*

$$\sigma_{\max}\left(\Lambda^{i*}\right) \leq \sqrt{2}\sigma_{\max}\left((\bar{\Lambda}^i)^*\right) = \sqrt{2}\sigma_{\max}\left(\boldsymbol{M}^{i*}\right), \tag{54}$$

$$\sigma_{\min}\left(\Lambda^{i*}\right) \geq \sigma_{\min}\left((\bar{\Lambda}^i)^*\right) = \sigma_{\min}\left(\boldsymbol{M}^{i*}\right) \tag{55}$$

*Proof.* See subsection D.3 □

Using Lemma 16, gives us:

$$\kappa\left(\Lambda^{i*}\right) \leq \sqrt{2}\kappa\left((\bar{\Lambda}^i)^*\right) = \sqrt{2}\kappa\left(\boldsymbol{M}^{i*}\right),$$

and therefore

$$\max_{i=1}^{k}\kappa\left(\Lambda^{i*}\right) \leq \sqrt{2}\max_{i=1}^{k}\kappa\left((\boldsymbol{M}^i)^*\right) \leq \sqrt{2}\kappa. \tag{56}$$

**D.2. Proof of Lemma 15**

We have:

$$\sigma_{min}\left(V_2^\top V_1\right) = \sqrt{\lambda_{min}\left(V_2^\top V_1 V_1^\top V_2\right)}$$

Now, note that:

$$\lambda_{\min}\left(V_2^\top V_1 V_1^\top V_2\right) = \min_{\|\boldsymbol{x}\|=1}\boldsymbol{x}^\top V_2^\top V_1 V_1^\top V_2 \boldsymbol{x}$$

$$= 1 + \min_{\|\boldsymbol{x}\|=1}\boldsymbol{x}^\top V_2^\top V_1 V_1^\top V_2 \boldsymbol{x} - \boldsymbol{x}^\top \boldsymbol{x}$$

$$= 1 + \min_{\|\boldsymbol{x}\|=1}\boldsymbol{x}^\top\left(V_2^\top V_1 V_1^\top V_2 - V_2^\top V_2\right)\boldsymbol{x}$$

$$\geq 1 - \left\|V_2^\top V_1 V_1^\top V_2 - V_2^\top V_2\right\|_2$$

$$\geq 1 - \left\|V_1 V_1^\top V_2 - V_2\right\|_2$$

$$= 1 - \text{dist}(V_1, V_2).$$

Therefore we get:

$$\sigma_{\min}\left(V_2^\top V_1\right) \geq \sqrt{1 - \operatorname{dist}(V_1, V_2)},$$

from which the following holds:

$$\left\|\left(V_2^\top V_1\right)^{-1}\right\|_2 \leq \frac{1}{\sqrt{1 - \operatorname{dist}(V_1, V_2)}}.$$

and completes the proof.

### D.3. Proof of Lemma 16

We have:

$$
\begin{aligned}
\sigma_{\max}\left(\Lambda^{i*}\right) &= \sigma_{\max}\left(((\bar{R}^i)^*)^{-1}(\bar{\Lambda}^i)^*\left((\bar{T}^i)^{*\top}\right)^{-1}\right) \\
&\leq \sigma_{\max}\left(((\bar{R}^i)^*)^{-1}\right)\sigma_{\max}\left((\bar{T}^i)^{*-1}\right)\sigma_{\max}\left((\bar{\Lambda}^i)^*\right) \\
&\leq \sqrt{2}\sigma_{\max}\left((\bar{\Lambda}^i)^*\right),
\end{aligned}
$$

where the second inequality follows from 51 and from its analogous version for $(\bar{T}^i)^*$. Similarly, we get:

$$
\begin{aligned}
\sigma_{\min}\left(\Lambda^{i*}\right) &= \sigma_{\min}\left(((\bar{R}^i)^*)^{-1}(\bar{\Lambda}^i)^*\left((\bar{T}^i)^{*\top}\right)^{-1}\right) \\
&\geq \sigma_{\min}\left(((\bar{R}^i)^*)^{-1}\right)\sigma_{\min}\left(((\bar{T}^i)^*)^{-1}\right)\sigma_{\min}\left((\bar{\Lambda}^i)^*\right) \\
&= \frac{1}{\left\|(\bar{R}^i)^*\right\|_2}\frac{1}{\left\|(\bar{T}^i)^*\right\|_2}\sigma_{\min}\left((\bar{\Lambda}^i)^*\right) \\
&\geq \sigma_{\min}\left((\bar{\Lambda}^i)^*\right).
\end{aligned}
$$

### D.4. Proof of Lemma 3

We define

$$
\begin{aligned}
\Lambda_{\text{avg}}^* &:= \frac{1}{k}\sum_{i=1}^k \Lambda^{i*} \\
M^* &:= U^*\Lambda_{\text{avg}}^* V^{*\top} \\
\Delta^{i*} &:= U^*\Lambda^{i*}V^{*\top} - U^{i*}\Lambda^{i*}V^{i*\top}.
\end{aligned}
$$

Note that, we have:

$$
\begin{aligned}
\left\|\Delta^{i*}\right\|_2 &\leq \left\|\left(U^* - U^{i*}\right)\Lambda^{i*}V^{*\top}\right\|_2 + \left\|U^{i*}\Lambda^{i*}(V^* - V^{i*\top})\right\|_F \\
&\leq \left(\left\|U^* - U^{i*}\right\|_2 + \left\|U^{i*}\right\|_2\left\|V^* - V^{i*}\right\|_2\right)\left\|\Lambda^{i*}\right\|_2 \\
&\leq 2\beta\left\|\Lambda^{i*}\right\|_2,
\end{aligned}
$$

and similarly $\left\|\Delta^{i*}\right\|_F \leq 2\beta\left\|\Lambda^{i*}\right\|_F$. Let $M^* = U^*\Lambda_{\text{avg}}^*(V^*)^\top$. Using this inequality, we obtain:

$$\left\|\boldsymbol{M}^* - \frac{1}{k}\sum_{i=1}^{k}\boldsymbol{M}^{i^*}\right\|_2 \leq \frac{1}{k}\sum_{i=1}^{k}\left\|\boldsymbol{U}^*\boldsymbol{\Lambda}^{i^*}\boldsymbol{V}^{*\top} - \boldsymbol{M}^{i^*}\right\|_2$$

$$\leq \frac{1}{k}\sum_{i=1}^{k}\left\|\boldsymbol{U}^*\boldsymbol{\Lambda}^{i^*}\boldsymbol{V}^{*\top} - (\boldsymbol{U}^i)^*\boldsymbol{\Lambda}^{i^*}\boldsymbol{V}^{i^*\top}\right\|_2$$

$$= \frac{1}{k}\sum_{i=1}^{k}\left\|\boldsymbol{\Delta}^{i^*}\right\|_2$$

$$\leq 2\beta\max_{i=1}^{k}\left\|\boldsymbol{\Lambda}^{i^*}\right\|_2$$

$$\leq 3\beta\max_{i=1}^{k}\left\|\boldsymbol{M}^{i^*}\right\|_2,$$

where the last inequality follows from Lemma 16. Hence, applying Weil's inequality yields:

$$\sigma_{\min}(\boldsymbol{\Lambda}^*_{\text{avg}}) = \sigma_{\min}(\boldsymbol{U}^*\boldsymbol{\Lambda}^*_{\text{avg}}\boldsymbol{V}^{*\top}) = \sigma_r(\boldsymbol{M}^*) \geq \sigma_r\left(\frac{1}{k}\sum_{i=1}^{k}\boldsymbol{M}^{i^*}\right) - \left\|\boldsymbol{M}^* - \frac{1}{k}\sum_{i=1}^{k}\boldsymbol{M}^{i^*}\right\|_2$$

$$\geq \sigma_r\left(\frac{1}{k}\sum_{i=1}^{k}\boldsymbol{M}^{i^*}\right) - 3\beta\max_{i=1}^{k}\left\|\boldsymbol{M}^{i^*}\right\|_2. \tag{57}$$

Now, we are going to bound $\left\|\widehat{\boldsymbol{M}} - \boldsymbol{M}^*\right\|_2$. Pick arbitrary $\boldsymbol{x}, \boldsymbol{y} \in R^d$ with $\|\boldsymbol{x}\| = \|\boldsymbol{y}\| = 1$. Using the definition of $\boldsymbol{y}_j^i$, we get:

$$\boldsymbol{x}^\top\widehat{\boldsymbol{M}}\boldsymbol{y} = \frac{1}{m}\sum_{i=1}^{k}\sum_{j=1}^{N}\boldsymbol{y}_j^i\boldsymbol{x}^\top\boldsymbol{G}_j^i\boldsymbol{y} = \frac{1}{m}\sum_{i=1}^{k}\sum_{j=1}^{N}\left\langle\boldsymbol{G}_j^i, (\boldsymbol{U}^i)^*\boldsymbol{\Lambda}^{i^*}\boldsymbol{V}^{i^*\top}\right\rangle\left\langle\boldsymbol{G}_j^i, \boldsymbol{x}\boldsymbol{y}^\top\right\rangle.$$

Similarly, we obtain:

$$\boldsymbol{x}^\top\boldsymbol{M}^*\boldsymbol{y} = \frac{1}{m}\sum_{i=1}^{k}\sum_{j=1}^{N}\boldsymbol{x}^\top\boldsymbol{U}^*\boldsymbol{\Lambda}^{i^*}\boldsymbol{V}^{*\top}\boldsymbol{y} = \frac{1}{m}\sum_{i=1}^{k}\sum_{j=1}^{N}\left\langle\boldsymbol{U}^*\boldsymbol{\Lambda}^{i^*}\boldsymbol{V}^{*\top}, \boldsymbol{x}\boldsymbol{y}^\top\right\rangle.$$

Therefore, we have:

$$\boldsymbol{x}^\top\widehat{\boldsymbol{M}}\boldsymbol{y} - \boldsymbol{x}^\top\boldsymbol{M}^*\boldsymbol{y} = \frac{1}{m}\sum_{i=1}^{k}\sum_{j=1}^{N}\left\langle\boldsymbol{G}_j^i, (\boldsymbol{U}^i)^*\boldsymbol{\Lambda}^{i^*}\boldsymbol{V}^{i^*\top}\right\rangle\left\langle\boldsymbol{G}_j^i, \boldsymbol{x}\boldsymbol{y}^\top\right\rangle - \frac{1}{m}\sum_{i=1}^{k}\sum_{j=1}^{N}\left\langle\boldsymbol{U}^*\boldsymbol{\Lambda}^{i^*}\boldsymbol{V}^{*\top}, \boldsymbol{x}\boldsymbol{y}^\top\right\rangle$$

$$= \underbrace{\frac{1}{m}\sum_{i=1}^{k}\sum_{j=1}^{N}\left\langle\boldsymbol{G}_j^i, \boldsymbol{U}^*\boldsymbol{\Lambda}^{i^*}\boldsymbol{V}^{*\top}\right\rangle\left\langle\boldsymbol{G}_j^i, \boldsymbol{x}\boldsymbol{y}^\top\right\rangle - \frac{1}{m}\sum_{i=1}^{k}\sum_{j=1}^{N}\left\langle\boldsymbol{U}^*\boldsymbol{\Lambda}^{i^*}\boldsymbol{V}^{*\top}, \boldsymbol{x}\boldsymbol{y}^\top\right\rangle}_{①}$$

$$+ \underbrace{\frac{1}{m}\sum_{i=1}^{k}\sum_{j=1}^{N}\left\langle\boldsymbol{G}_j^i, \boldsymbol{\Delta}^{i^*}\right\rangle\left\langle\boldsymbol{G}_j^i, \boldsymbol{x}\boldsymbol{y}^\top\right\rangle}_{②}.$$

Since $\{\boldsymbol{G}_j^i\}$ satisfies the $3r$-GRIP with coefficient $\delta_{3r}$, Lemma 12 yields:

$$① \leq 18\delta_{3r}\max_{i=1}^{k}\left\|\boldsymbol{U}^*\boldsymbol{\Lambda}^{i^*}\boldsymbol{V}^{*\top}\right\|_F\left\|\boldsymbol{x}\boldsymbol{y}^\top\right\|_F = 18\delta_{3r}\left\|\boldsymbol{x}\boldsymbol{y}^\top\right\|_F\max_{i=1}^{k}\left\|\boldsymbol{\Lambda}^{i^*}\right\|_F.$$

For the second term, applying C-S inequality, gives us:

$$\textcircled{2} \leq \sqrt{\frac{1}{m}\sum_{i=1}^{k}\sum_{j=1}^{N}\left\langle \boldsymbol{G}_j^i, \boldsymbol{\Delta}^{i*}\right\rangle^2} \cdot \sqrt{\frac{1}{m}\sum_{i=1}^{k}\sum_{j=1}^{N}\left\langle \boldsymbol{G}_j^i, \boldsymbol{x}\boldsymbol{y}^\top\right\rangle^2}$$

$$\leq \sqrt{A_{n,d,r}\max_{i=1}^{k}\left\|\boldsymbol{\Delta}^{i*}\right\|_F^2} \cdot \sqrt{(1+\delta_{3r})\left\|\boldsymbol{x}\boldsymbol{y}^\top\right\|_F^2}$$

$$\leq 4\beta\max_{i=1}^{k}\left\|\boldsymbol{\Lambda}^{i*}\right\|_F\sqrt{A_{n,d,r}}\cdot\left\|\boldsymbol{x}\boldsymbol{y}^\top\right\|_F,$$

where the second inequality follows since $\{\boldsymbol{G}_j^i\}$ is sub-isometric with coefficient $A_{n,d,r}$ and satisfies $3r$-GRIP with coefficient $\delta_{3r}$.

Combining these bounds, gives us:

$$\left\|\widehat{\boldsymbol{M}}-\boldsymbol{M}^*\right\|_2 = \max_{\|\boldsymbol{x}\|=1,\|\boldsymbol{y}\|=1}\boldsymbol{x}^\top\left(\widehat{\boldsymbol{M}}-\boldsymbol{M}^*\right)\boldsymbol{y}$$

$$= \max_{\|\boldsymbol{x}\|=1,\|\boldsymbol{y}\|=1}\boldsymbol{x}^\top\widehat{\boldsymbol{M}}\boldsymbol{y}-\boldsymbol{x}^\top\boldsymbol{M}^*\boldsymbol{y}$$

$$\leq \max_{\|\boldsymbol{x}\|=1,\|\boldsymbol{y}\|=1}18\delta_{3r}\left\|\boldsymbol{x}\boldsymbol{y}^\top\right\|_F\max_{i=1}^{k}\left\|\boldsymbol{\Lambda}^{i*}\right\|_F + 4\beta\max_{i=1}^{k}\left\|\boldsymbol{\Lambda}^{i*}\right\|_F\sqrt{A_{n,d,r}}\cdot\left\|\boldsymbol{x}\boldsymbol{y}^\top\right\|_F$$

$$= 18\max_{i=1}^{k}\left\|\boldsymbol{\Lambda}^{i*}\right\|_F\left(\delta_{3r}+\beta\cdot\sqrt{A_{n,d,r}}\right)$$

By the triangle inequality,

$$\|\widehat{\boldsymbol{M}_r}-\boldsymbol{M}^*\|_2 \leq \|\widehat{\boldsymbol{M}}-\boldsymbol{M}_r\|_2 + \|\widehat{\boldsymbol{M}}-\boldsymbol{M}^*\|_2.$$

Moreover, since $\boldsymbol{M}_r$ is the best rank-$r$ approximation of $\widehat{\boldsymbol{M}}$ in spectral norm, the Eckart–Young–Mirsky theorem implies

$$\|\widehat{\boldsymbol{M}}-\boldsymbol{M}_r\|_2 \leq \|\widehat{\boldsymbol{M}}-\boldsymbol{M}^*\|_2.$$

Combining the two bounds yields

$$\|\boldsymbol{M}_r-\boldsymbol{M}^*\|_2 \leq 2\|\widehat{\boldsymbol{M}}-\boldsymbol{M}^*\|_2.$$

Finnaly, we obtain:

$$\text{dist}(\boldsymbol{U}_0,\boldsymbol{U}^*)\sigma_{\min}\left(\boldsymbol{\Lambda}_{\text{avg}}^*\right) = \left\|\left(\boldsymbol{U}_0\boldsymbol{U}_0^\top-I\right)\boldsymbol{U}^*\right\|_2\sigma_{\min}\left(\boldsymbol{\Lambda}_{\text{avg}}^*\right)$$

$$\leq \left\|\left(\boldsymbol{U}_0\boldsymbol{U}_0^\top-I\right)\boldsymbol{U}^*\boldsymbol{\Lambda}_{\text{avg}}^*\right\|_2$$

$$= \left\|\left(\boldsymbol{U}_0\boldsymbol{U}_0^\top-I\right)\boldsymbol{U}^*\boldsymbol{\Lambda}_{\text{avg}}^*(\boldsymbol{V}^*)^\top\right\|_2$$

$$= \left\|\left(\boldsymbol{U}_0\boldsymbol{U}_0^\top-I\right)\left(\boldsymbol{U}^*\boldsymbol{\Lambda}_{\text{avg}}^*(\boldsymbol{V}^*)^\top-\boldsymbol{U}_0\boldsymbol{\Lambda}_0\boldsymbol{V}_0^\top\right)\right\|_2$$

$$\leq \left\|\left(\boldsymbol{U}_0\boldsymbol{U}_0^\top-I\right)\right\|_2\left\|\left(\boldsymbol{U}^*\boldsymbol{\Lambda}_{\text{avg}}^*(\boldsymbol{V}^*)^\top-\boldsymbol{U}_0\boldsymbol{\Lambda}_0\boldsymbol{V}_0^\top\right)\right\|_2$$

$$\leq \|\boldsymbol{M}_r-\boldsymbol{M}^*\|_2$$

$$\leq 2\left\|\widehat{\boldsymbol{M}}-\boldsymbol{M}^*\right\|_2$$

$$\leq 36\max_{i=1}^{k}\left\|\boldsymbol{\Lambda}^{i*}\right\|_F\left(r\delta_{3r}+\beta\cdot\sqrt{A_{n,d,r}}\right)$$

$$\leq 51\sqrt{r}\max_{i=1}^{k}\sigma_{\max}\left(\boldsymbol{M}^{i*}\right)\left(\delta_{3r}+\beta\cdot\sqrt{A_{n,d,r}}\right),$$

Hence, using 57, we conclude that:

$$
\begin{aligned}
\operatorname{dist}(\boldsymbol{U}_0, \boldsymbol{U}^*) &\le 51\sqrt{r}\frac{\max_{i=1}^{k}\left\|(\boldsymbol{M}^i)^*\right\|_2}{\sigma_r\left(\frac{1}{k}\sum_{i=1}^{k}(\boldsymbol{M}^i)^*\right) - 3\beta\max_{i=1}^{k}\left\|(\boldsymbol{M}^i)^*\right\|_2} \cdot \left(\delta_{3r} + \beta\cdot\sqrt{A_{n,d,r}}\right) \\
&\le 100\sqrt{r}\frac{\max_{i=1}^{k}\left\|(\boldsymbol{M}^i)^*\right\|_2}{\sigma_r\left(\frac{1}{k}\sum_{i=1}^{k}(\boldsymbol{M}^i)^*\right)} \cdot \left(\delta_{3r} + \beta\cdot\sqrt{A_{n,d,r}}\right) \\
&= 100\sqrt{r}\gamma\left(\delta_{3r} + \beta\cdot\sqrt{A_{n,d,r}}\right) \\
&\le \frac{1}{20\kappa}
\end{aligned}
$$

where we used 20 and 19. Similarly, we have:

$$
\operatorname{dist}(\boldsymbol{V}_0, \boldsymbol{V}^*) \le \frac{1}{20\kappa}.
$$

## D.5. Proof of Lemma 4

Let $\boldsymbol{Z} \in R^{r\times r}$ and $\boldsymbol{z} = \operatorname{vec}(\boldsymbol{Z})$. Than we have:

$$
\begin{aligned}
\sigma_{\min}(\boldsymbol{B}) &= \min_{\|\boldsymbol{z}\|=1} \boldsymbol{z}^\top \boldsymbol{B}\boldsymbol{z} \\
&= \min_{\|\boldsymbol{z}\|=1} \boldsymbol{z}^\top \left(\sum_{j=1}^{n} \boldsymbol{V}^\top \otimes \boldsymbol{U}^\top \operatorname{vec}\left(\boldsymbol{G}_j^i\right)\operatorname{vec}\left(\boldsymbol{G}_j^i\right)^\top \boldsymbol{V}\otimes\boldsymbol{U}\right)\boldsymbol{z} \\
&= \min_{\|\boldsymbol{z}\|=1} \left(\sum_{j=1}^{n}\operatorname{vec}(\boldsymbol{Z})^\top \boldsymbol{V}^\top\otimes\boldsymbol{U}^\top\operatorname{vec}\left(\boldsymbol{G}_j^i\right)\operatorname{vec}\left(\boldsymbol{G}_j^i\right)^\top\boldsymbol{V}\otimes\boldsymbol{U}\operatorname{vec}(\boldsymbol{Z})\right) \\
&= \min_{\|\boldsymbol{z}\|=1}\left(\sum_{j=1}^{n}\operatorname{vec}\left(\boldsymbol{U}\boldsymbol{Z}\boldsymbol{V}^\top\right)^\top\operatorname{vec}\left(\boldsymbol{G}_j^i\right)\operatorname{vec}\left(\boldsymbol{G}_j^i\right)^\top\operatorname{vec}\left(\boldsymbol{U}\boldsymbol{Z}\boldsymbol{V}^\top\right)\right) \\
&= \min_{\|\boldsymbol{z}\|=1}\sum_{j=1}^{n}\left(\operatorname{vec}\left(\boldsymbol{G}_j^i\right)^\top\operatorname{vec}\left(\boldsymbol{U}\boldsymbol{Z}\boldsymbol{V}^\top\right)\right)^2 \\
&= \min_{\|\boldsymbol{z}\|=1}\sum_{j=1}^{n}\left\langle\boldsymbol{G}_j^i, \boldsymbol{U}\boldsymbol{Z}\boldsymbol{V}^\top\right\rangle^2.
\end{aligned}
$$

Now, since $\{\boldsymbol{G}_j^i\}_{j=1}^{n}$ satisfies $(\boldsymbol{U}, \boldsymbol{V})$-RIP with $\delta_{2r}'$, it follows that:

$$
\begin{aligned}
\sum_{j=1}^{n}\left\langle\boldsymbol{G}_j^i, \boldsymbol{U}\boldsymbol{Z}\boldsymbol{V}^\top\right\rangle^2 &\ge \sum_{j=1}^{n}\left\|\boldsymbol{U}\boldsymbol{Z}\boldsymbol{V}^\top\right\|_F^2 - n\delta_{2r}'\left\|\boldsymbol{U}\boldsymbol{Z}\boldsymbol{V}^\top\right\|_F^2 \\
&= n - n\delta_{2r}'\|\boldsymbol{Z}\|_F^2,
\end{aligned}
$$

where the equality follows since $\boldsymbol{U}, \boldsymbol{V}$ are orthonormal matrices and therefore $\left\|\boldsymbol{U}\boldsymbol{Z}\boldsymbol{V}^\top\right\|_F = \|\boldsymbol{Z}\|_F$. Hence, we conclude:

$$
\min_{\|\boldsymbol{z}\|=1}\sum_{j=1}^{n}\left\langle\boldsymbol{G}_j^i, \boldsymbol{U}\boldsymbol{Z}\boldsymbol{V}^\top\right\rangle \ge \min_{\|\boldsymbol{z}\|=1} n\left(1 - \delta_{2r}'\|\boldsymbol{Z}\|_F\right) = n(1 - \delta_{2r}').
$$

## D.6. Proof of Lemma 5

Let $\boldsymbol{Z} \in R^{r\times r}$ and $\boldsymbol{z} = \operatorname{vec}(\boldsymbol{Z})$. Than, we have:

$$\left\| \left( B(V^\top V^{i^*}) \otimes (U^\top U^{i^*}) - C \right) \text{vec} \left( \Lambda^{i^*} \right) \right\|_2 = \max_{\|z\|=1} z^\top \left( B(V^\top V^{i^*}) \otimes (U^\top U^{i^*}) - C \right) \text{vec} \left( \Lambda^{i^*} \right)$$

$$\overset{(\zeta_1)}{=} \max_{\|z\|=1} \sum_{j=1}^n \text{vec} \left( Z \right)^\top V^\top \otimes U^\top \text{vec} \left( G_j^i \right) \text{vec} \left( G_j^i \right)^\top \left( VV^\top V^{i^*} \otimes UU^\top U^{i^*} - V^{i^*} \otimes U^{i^*} \right) \text{vec} \left( \Lambda^{i^*} \right)$$

$$\overset{(\zeta_2)}{=} \max_{\|z\|=1} \sum_{j=1}^n \text{vec} \left( Z \right)^\top V^\top \otimes U^\top \text{vec} \left( G_j^i \right) \text{vec} \left( G_j^i \right)^\top \left( VV^\top V^{i^*} - V^{i^*} \right) \otimes UU^\top U^{i^*} \text{vec} \left( \Lambda^{i^*} \right)$$

$$+ \sum_{j=1}^n \text{vec} \left( Z \right)^\top V^\top \otimes U^\top \text{vec} \left( G_j^i \right) \text{vec} \left( G_j^i \right)^\top V^{i^*} \otimes \left( UU^\top U^{i^*} - U^{i^*} \right) \text{vec} \left( \Lambda^{i^*} \right)$$

$$\overset{(\zeta_3)}{=} \max_{\|z\|=1} \sum_{j=1}^n \langle G_j^i, UZV^\top \rangle \left\langle G_j^i, UU^\top U^{i^*} \Lambda^{i^*} \left( VV^\top V^{i^*} - V^{i^*} \right)^\top \right\rangle$$

$$+ \sum_{j=1}^n \langle G_j^i, UZV^\top \rangle \left\langle G_j^i, \left( UU^\top U^{i^*} - U^{i^*} \right) \Lambda^{i^*} (V^{i^*})^\top \right\rangle,$$

where $\zeta_1$ follows from the definition of $B$ and $C$, $\zeta_2$ is just an algebraic manipulation of adding and subtracting the same term, and $\zeta_3$ follows from identity 23 and writing the inner products in terms of the matrices. Recall that the ensemble $\{G_j^i\}_{j=1}^n$ satisfies $(\widetilde{U}_1, \widetilde{V}_1)$-RIP with coefficient $\delta'_{2r}$, where

$$\widetilde{U}_1 = [U, \ UU^\top U^{i^*}], \qquad\qquad \widetilde{V}_1 = [V, \ VV^\top V^{i^*} - V^{i^*}].$$

Hence, applying Lemma 13, we obtain:

$$\sum_{j=1}^n \langle G_j^i, UZV^\top \rangle \left\langle G_j^i, UU^\top U^{i^*} \Lambda^{i^*} \left( VV^\top V^{i^*} - V^{i^*} \right)^\top \right\rangle$$

$$\le 18n\delta'_{2r} \left\| UZV^\top \right\|_F \left\| UU^\top U^{i^*} \Lambda^{i^*} \left( VV^\top V^{i^*} - V^{i^*} \right)^\top \right\|_F$$

$$= 18n\delta'_{2r} \left\| U^\top U^{i^*} \Lambda^{i^*} \left( VV^\top V^{i^*} - V^{i^*} \right)^\top \right\|_F$$

$$\le 18n\delta'_{2r} \left\| U^\top U^{i^*} \right\|_2 \left\| \Lambda^{i^*} \left( VV^\top V^{i^*} - V^{i^*} \right)^\top \right\|_F$$

$$\le 18n\delta'_{2r} \left\| U^\top \right\|_2 \left\| U^{i^*} \right\|_2 \left\| \Lambda^{i^*} \right\|_F \left\| \left( VV^\top V^{i^*} - V^{i^*} \right)^\top \right\|_2$$

$$= 18n\delta'_{2r} \left\| \Lambda^{i^*} \right\|_F \left\| \left( VV^\top V^{i^*} - V^{i^*} \right)^\top \right\|_2$$

$$\le 18n\delta'_{2r} \left\| \Lambda^{i^*} \right\|_F \left\| \left( VV^\top V^{i^*} - V^{i^*} \right)^\top \right\|_2 \left\| V^{i^*} \right\|_2$$

$$\le 36n\delta'_{2r} \left\| \Lambda^{i^*} \right\|_F \text{dist}(V, V^{i^*}),$$

where we used orthonormality of $U, V$, and also applied $\|AB\|_F \le \|A\|_2 \|B\|_F$ inequality several times. Similarly, using $(\widetilde{U}_2, \widetilde{V}_2)$-RIP, we obtain:

$$\sum_{j=1}^n \langle G_j^i, UZV^\top \rangle \left\langle G_j^i, \left( UU^\top U^{i^*} - U^{i^*} \right) \Lambda^{i^*} (V^{i^*})^\top \right\rangle \le 72n\delta'_{2r} \left\| \Lambda^{i^*} \right\|_F \text{dist}(U, U^{i^*}).$$

Adding these bounds completes the proof.

## D.7. Proof of Lemma 6

For the ease of notation we remove the superscript $t$ from the variables. Hence

$$U = U_t, V = V_t, \Lambda^i = \Lambda^i_{t+1}$$

Applying Weil's inequality yields:

$$
\begin{aligned}
\sigma_i(\Lambda^{i^*}) &\geq \sigma_i\left(U^\top U^{i^*} \Lambda^{i^*}(V^{i^*})^\top V\right) - \left\|\Lambda^i - U^\top U^{i^*}\Lambda^{i^*}(V^{i^*})^\top V\right\|_2 \\
&\overset{(\xi_1)}{\geq} \sigma_i\left(U^\top U^{i^*}\Lambda^{i^*}(V^{i^*})^\top V\right) - 144\delta'_{2r}\left\|\Lambda^{i^*}\right\|_F \left(\operatorname{dist}(U, U^{i^*}) + \operatorname{dist}(V, V^{i^*})\right) \\
&\overset{(\xi_2)}{\geq} \sigma_i\left(U^\top U^{i^*}\Lambda^{i^*}(V^{i^*})^\top V\right) - \sigma_i\left(\Lambda^{i^*}\right) \cdot 144\kappa\sqrt{r}\delta'_{2r}\left(\operatorname{dist}(U, U^{i^*}) + \operatorname{dist}(V, V^{i^*})\right),
\end{aligned}
$$

where $(\xi_1)$ follows from 37 and $(\xi_2)$ from 56. We further bound the first term using Wei'ls inequality as follows:

$$
\begin{aligned}
\sigma_i\left(U^\top U^{i^*}\Lambda^{i^*}(V^{i^*})^\top V\right) &= \sigma_i\left(UU^\top U^{i^*}\Lambda^{i^*}(V^{i^*})^\top V\right) \\
&\geq \sigma_i\left(U^{i^*}\Lambda^{i^*}(V^{i^*})^\top V\right) - \left\|(UU^\top U^{i^*} - U^{i^*})\Lambda^{i^*}(V^{i^*})^\top VV^\top\right\|_2 \\
&\geq \sigma_i\left(\Lambda^{i^*}(V^{i^*})^\top V\right) - \left\|(UU^\top U^{i^*} - U^{i^*})\right\|_2\left\|\Lambda^{i^*}(V^{i^*})^\top VV^\top\right\|_2 \\
&\geq \sigma_i\left(\Lambda^{i^*}(V^{i^*})^\top V\right) - \operatorname{dist}(U, U^{i^*})\left\|\Lambda^{i^*}\right\|_2 \\
&= \sigma_i\left(\Lambda^{i^*}(V^{i^*})^\top VV^\top\right) - \operatorname{dist}(U, U^{i^*})\left\|\Lambda^{i^*}\right\|_2 \\
&\overset{(\xi_1)}{\geq} \sigma_i\left(\Lambda^{i^*}\right) - \operatorname{dist}(V, V^{i^*})\left\|\Lambda^{i^*}\right\|_2 - \operatorname{dist}(U, U^{i^*})\left\|\Lambda^{i^*}\right\|_2 \\
&\overset{(\xi_2)}{\geq} \sigma_i\left(\Lambda^{i^*}\right)\left(1 - 2\kappa \cdot \operatorname{dist}(U, U^{i^*}) - 2\kappa \cdot \operatorname{dist}(V, V^{i^*})\right),
\end{aligned}
$$

where $(\xi_1)$ follows from a similar application of Weil's inequality and $(\xi_2)$ follows since

$$\left\|\Lambda^{i^*}\right\|_2 \leq \kappa\left(\Lambda^{i^*}\right)\sigma_i\left(\Lambda^{i^*}\right) \leq 2\kappa\sigma_i\left(\Lambda^{i^*}\right).$$

Combining these bounds gives:

$$
\begin{aligned}
\sigma_i(\Lambda^i)_2 &\geq \sigma_i(\Lambda^{i^*})\left(1 - (2\kappa + 288\kappa\sqrt{r}\delta'_{2r})\left(\operatorname{dist}(U, U^{i^*}) + \operatorname{dist}(V, V^{i^*})\right)\right) \\
&\geq \sigma_i(\Lambda^{i^*})\left(1 - (2\kappa + 288\kappa\sqrt{r}\delta'_{2r})\left(\operatorname{dist}(U, U^*) + \operatorname{dist}(U^*, U^{i^*}) + \operatorname{dist}(V, V^*) + \operatorname{dist}(V^*, V^{i^*})\right)\right) \\
&\geq \sigma_i(\Lambda^{i^*})\left(1 - (2\kappa + 288\kappa\sqrt{r}\delta'_{2r})\left(\operatorname{dist}(U, U^*) + \operatorname{dist}(V, V^*) + 2\beta\right)\right) \\
&\overset{(\xi_1)}{\geq} \sigma_i(\Lambda^{i^*})\left(1 - (2\kappa + 288\kappa\sqrt{r}\delta'_{2r})\left(\operatorname{dist}(U_0, U^*) + \operatorname{dist}(V_0, V^*) + 10\beta B_{n,d,r}\right)\right) \\
&\overset{(\xi_2)}{\geq} \frac{1}{2}\sigma_{\min}\left(\Lambda^{i^*}\right).
\end{aligned}
$$

where $(\xi_1)$ follows from applying our induction hypothesis Proposition 2 yields:

$$
\begin{aligned}
\operatorname{dist}(U, U^*) + \operatorname{dist}(V, V^*) &= \operatorname{dist}(U_t, U^*) + \operatorname{dist}(V_t, V^*) \\
&\leq \left(\frac{1}{2}\right)^t \left(\operatorname{dist}(U_0, U^*) + \operatorname{dist}(V_0, V^*)\right) + 8\beta B_{n,d,r} \\
&\leq \operatorname{dist}(U_0, U^*) + \operatorname{dist}(V_0, V^*) + 8\beta B_{n,d,r},
\end{aligned}
$$

and $(\xi_2)$ from using Lemma 3 and inequalities 19, 21. The upper bound on $\sigma_i(\Lambda^{i^*})_2$ can be derived similarly. Note that 39 directly follows from 38.

## D.8. Proof of Lemma 7

Note that we have:

$$\sigma_{\min}\left(\sum_{i=1}^{k}\boldsymbol{B}_i\right) = \min_{\|\boldsymbol{z}\|=1}\boldsymbol{z}^{\top}\left(\sum_{i=1}^{k}\boldsymbol{B}_i\right)\boldsymbol{z}.$$

Let $\boldsymbol{Z} \in \mathbb{R}^{r \times r}$ be arbitrary with $\|\boldsymbol{Z}\|_F = 1$, and define $\boldsymbol{z} = \mathrm{vec}\,(\boldsymbol{Z})$. Than we have:

$$\begin{aligned}
\boldsymbol{z}^{\top}\left(\sum_{i=1}^{k}\boldsymbol{B}^i\right)\boldsymbol{z} &= \boldsymbol{z}^{\top}\left(\sum_{i=1}^{k}\sum_{j=1}^{N}(\boldsymbol{U}\boldsymbol{\Lambda}^i)^{\top}\otimes\boldsymbol{I}_d\cdot\mathrm{vec}\left((\boldsymbol{G}_j^i)^{\top}\right)\mathrm{vec}\left((\boldsymbol{G}_j^i)^{\top}\right)^{\top}\cdot(\boldsymbol{U}\boldsymbol{\Lambda}^i)\otimes\boldsymbol{I}_d\right)\boldsymbol{z} \\
&= \left(\sum_{i=1}^{k}\sum_{j=1}^{N}\mathrm{vec}\,(\boldsymbol{Z})^{\top}\cdot(\boldsymbol{U}\boldsymbol{\Lambda}^i)^{\top}\otimes\boldsymbol{I}_d\cdot\mathrm{vec}\left((\boldsymbol{G}_j^i)^{\top}\right)\mathrm{vec}\left((\boldsymbol{G}_j^i)^{\top}\right)^{\top}\cdot(\boldsymbol{U}\boldsymbol{\Lambda}^i)\otimes\boldsymbol{I}_d\cdot\mathrm{vec}\,(\boldsymbol{Z})\right) \\
&= \left(\sum_{i=1}^{k}\sum_{j=1}^{N}\mathrm{vec}\left(\boldsymbol{Z}(\boldsymbol{U}\boldsymbol{\Lambda}^i)^{\top}\right)\cdot\mathrm{vec}\left((\boldsymbol{G}_j^i)^{\top}\right)\mathrm{vec}\left((\boldsymbol{G}_j^i)^{\top}\right)^{\top}\cdot\mathrm{vec}\left(\boldsymbol{Z}(\boldsymbol{U}\boldsymbol{\Lambda}^i)^{\top}\right)\right) \\
&= \sum_{i=1}^{k}\sum_{j=1}^{N}\left\langle(\boldsymbol{G}_j^i)^{\top},\boldsymbol{Z}(\boldsymbol{\Lambda}^i)^{\top}\boldsymbol{U}^{\top}\right\rangle^2
\end{aligned}$$

Now, since $\{(\boldsymbol{G}_j^i)^{\top}\}$ satisfies $3r$-GRIP with $\delta_{3r}$, the last term is lower bounded as follows:

$$\begin{aligned}
&\geq N\cdot\sum_{i=1}^{k}\left\|\boldsymbol{Z}(\boldsymbol{\Lambda}^i)^{\top}\boldsymbol{U}^{\top}\right\|_F^2 - \delta_{3r}m\max_{i=1}^{k}\left\|\boldsymbol{Z}(\boldsymbol{\Lambda}^i)^{\top}\boldsymbol{U}^{\top}\right\|_F^2 \\
&\overset{(\xi_1)}{\geq} N\cdot\sum_{i=1}^{k}\sigma_{\min}^2(\boldsymbol{\Lambda}^i) - \delta_{3r}m\cdot\max_{i=1}^{k}\left\|\boldsymbol{\Lambda}^i\right\|_2^2 \\
&\overset{(\xi_2)}{\geq} \frac{N}{4}\cdot\sum_{i=1}^{k}\sigma_{\min}^2(\boldsymbol{\Lambda}^{i^*}) - 4\delta_{3r}m\cdot\max_{i=1}^{k}\sigma_{\max}^2(\boldsymbol{\Lambda}^{i^*}) \\
&\overset{(\xi_3)}{\geq} \frac{n_1k}{4}\cdot\min_{i=1}^{k}\sigma_{\min}^2(\boldsymbol{\Lambda}^{i^*}) - 4\delta_{3r}m\kappa^2\cdot\max_{i=1}^{k}\sigma_{\min}^2(\boldsymbol{\Lambda}^{i^*}) \\
&= \frac{m}{4}\cdot\min_{i=1}^{k}\sigma_{\min}^2(\boldsymbol{\Lambda}^{i^*})\left(1 - 16\delta_{3r}\kappa^2\right) \\
&\overset{(\xi_4)}{\geq} \frac{m}{8}\cdot\min_{i=1}^{k}\sigma_{\min}^2(\boldsymbol{\Lambda}^{i^*}),
\end{aligned}$$

where $(\xi_1)$ follows since $\boldsymbol{U}$ is orthogonal and $\left\|(\boldsymbol{\Lambda}^i)\boldsymbol{Z}^{\top}\right\|_F \geq \sigma_{\min}(\boldsymbol{\Lambda}^i)\left\|\boldsymbol{Z}^{\top}\right\|_F = \sigma_{\min}(\boldsymbol{\Lambda}^i)$ and $\left\|\boldsymbol{Z}(\boldsymbol{\Lambda}^i)^{\top}\right\|_F \leq \left\|\boldsymbol{\Lambda}^i\right\|_2\|\boldsymbol{Z}\|_F = \left\|\boldsymbol{\Lambda}^i\right\|_2$, $(\xi_2)$ follows from Lemma 6, $(\xi_3)$ follows from the definition of $\kappa$, and finally $(\xi_4)$ holds since from 20 we have $\delta_{3r} \leq \frac{1}{32}\kappa^{-2}$. Since $\boldsymbol{z}$ was arbitrary, this completes the proof.

## D.9. Proof of Lemma 8

Note that we have:

$$\|\boldsymbol{F}_1\| = \max_{\|\boldsymbol{x}\|=1}\boldsymbol{x}^{\top}\boldsymbol{F}_1.$$

Let $\boldsymbol{X} \in \mathbb{R}^{d \times r}$ be arbitrary with $\|\boldsymbol{X}\|_F = 1$, and define $\boldsymbol{x} = \text{vec}\,(\boldsymbol{X})$. We have:

$$\boldsymbol{x}^\top \boldsymbol{F}_1 =$$

$$= \boldsymbol{x}^\top \left( \sum_{i=1}^{k} \sum_{j=1}^{N} (\boldsymbol{U}\boldsymbol{\Lambda}^i)^\top \otimes \boldsymbol{I}_d \cdot \text{vec}\left((\boldsymbol{G}_j^i)^\top\right) \text{vec}\left((\boldsymbol{G}_j^i)^\top\right)^\top \cdot \left( (\boldsymbol{U}\boldsymbol{\Lambda}^i \boldsymbol{D}^{-1}) \otimes \boldsymbol{I}_d - \boldsymbol{U}\boldsymbol{U}^\top \boldsymbol{U}^{i*} \boldsymbol{\Lambda}^{i*} \otimes \boldsymbol{I}_d \right) \right) \text{vec}\,(\boldsymbol{V}^*)$$

$$= \sum_{i=1}^{k} \sum_{j=1}^{N} \text{vec}\,(\boldsymbol{X})^\top (\boldsymbol{U}\boldsymbol{\Lambda}^i)^\top \otimes \boldsymbol{I}_d \cdot \text{vec}\left((\boldsymbol{G}_j^i)^\top\right) \text{vec}\left((\boldsymbol{G}_j^i)^\top\right)^\top \cdot \left( (\boldsymbol{U}\boldsymbol{\Lambda}^i \boldsymbol{D}^{-1}) \otimes \boldsymbol{I}_d - \boldsymbol{U}\boldsymbol{U}^\top \boldsymbol{U}^{i*} \boldsymbol{\Lambda}^{i*} \otimes \boldsymbol{I}_d \right) \text{vec}\,(\boldsymbol{V}^*)$$

$$= \sum_{i=1}^{k} \sum_{j=1}^{N} \text{vec}\left(\boldsymbol{X}(\boldsymbol{\Lambda}^i)^\top \boldsymbol{U}^\top\right)^\top \cdot \text{vec}\left((\boldsymbol{G}_j^i)^\top\right) \text{vec}\left((\boldsymbol{G}_j^i)^\top\right)^\top \cdot \text{vec}\left( \boldsymbol{V}^* \left( \boldsymbol{U}\boldsymbol{\Lambda}^i \boldsymbol{D}^{-1} - \boldsymbol{U}\boldsymbol{U}^\top \boldsymbol{U}^{i*} \boldsymbol{\Lambda}^{i*} \right)^\top \right)$$

$$= \sum_{i=1}^{k} \sum_{j=1}^{N} \langle (\boldsymbol{G}_j^i)^\top, \boldsymbol{X}(\boldsymbol{\Lambda}^i)^\top \boldsymbol{U}^\top \rangle \left\langle (\boldsymbol{G}_j^i)^\top, \boldsymbol{V}^* \left( \boldsymbol{U}\boldsymbol{\Lambda}^i \boldsymbol{D}^{-1} - \boldsymbol{U}\boldsymbol{U}^\top \boldsymbol{U}^{i*} \boldsymbol{\Lambda}^{i*} \right)^\top \right\rangle$$

$$= \underbrace{\sum_{i=1}^{k} \sum_{j=1}^{N} \langle (\boldsymbol{G}_j^i)^\top, \boldsymbol{X}(\boldsymbol{\Lambda}^i)^\top \boldsymbol{U}^\top \rangle \left\langle (\boldsymbol{G}_j^i)^\top, \boldsymbol{V}^* \left( \boldsymbol{U}\boldsymbol{\Lambda}^i \boldsymbol{D}_i^{-1} - \boldsymbol{U}\boldsymbol{U}^\top \boldsymbol{U}^{i*} \boldsymbol{\Lambda}^{i*} \right)^\top \right\rangle}_{①}$$

$$+ \underbrace{\sum_{i=1}^{k} \sum_{j=1}^{N} \langle (\boldsymbol{G}_j^i)^\top, \boldsymbol{X}(\boldsymbol{\Lambda}^i)^\top \boldsymbol{U}^\top \rangle \langle (\boldsymbol{G}_j^i)^\top, \boldsymbol{V}^* (\boldsymbol{D}^{-1} - \boldsymbol{D}_i^{-1})(\boldsymbol{\Lambda}^i)^\top \boldsymbol{U}^\top \rangle}_{②},$$

where $\boldsymbol{D}_i = (\boldsymbol{V}^{i*})^\top \boldsymbol{V}$. Using Lemma 15, we have:

$$\left\|\boldsymbol{D}_i^{-1}\right\|_2 \le \frac{1}{\sqrt{1 - \text{dist}(\boldsymbol{V}^{i*}, \boldsymbol{V})}} \le \frac{1}{\sqrt{1 - \text{dist}(\boldsymbol{V}^*, \boldsymbol{V}) - \text{dist}(\boldsymbol{V}^{i*}, \boldsymbol{V}^*)}}$$

$$\le \frac{1}{\sqrt{1 - \beta - \text{dist}(\boldsymbol{V}^*, \boldsymbol{V})}} \le 2 \tag{58}$$

Additionally, we obtain:

$$\left\|\boldsymbol{D}^{-1} - \boldsymbol{D}_i^{-1}\right\|_2 \le \left\|\boldsymbol{D}^{-1}\right\|_2 \left\|\boldsymbol{D}_i^{-1}\right\|_2 \left\|\boldsymbol{D}_i - \boldsymbol{D}\right\|_2 \le 4 \left\|(\boldsymbol{V}^{i*} - \boldsymbol{V}^*)^\top \boldsymbol{V}\right\| \le 4\beta. \tag{59}$$

Now, note that we have:

$$\left\|\boldsymbol{U}\boldsymbol{\Lambda}^i \boldsymbol{D}_i^{-1} - \boldsymbol{U}\boldsymbol{U}^\top \boldsymbol{U}^{i*} \boldsymbol{\Lambda}^{i*}\right\|_F = \left\|\left(\boldsymbol{U}\boldsymbol{\Lambda}^i \boldsymbol{D}_i^{-1} - \boldsymbol{U}\boldsymbol{U}^\top \boldsymbol{U}^{i*} \boldsymbol{\Lambda}^{i*}\right) \boldsymbol{D}_i \boldsymbol{D}_i^{-1}\right\|_F$$

$$= \left\|\left(\boldsymbol{U}\boldsymbol{\Lambda}^i - \boldsymbol{U}\boldsymbol{U}^\top \boldsymbol{U}^{i*} \boldsymbol{\Lambda}^{i*} (\boldsymbol{V}^{i*})^\top \boldsymbol{V}\right) \boldsymbol{D}_i^{-1}\right\|_F$$

$$\le \left\|\boldsymbol{U}\boldsymbol{\Lambda}^i - \boldsymbol{U}\boldsymbol{U}^\top \boldsymbol{U}^{i*} \boldsymbol{\Lambda}^{i*} (\boldsymbol{V}^{i*})^\top \boldsymbol{V}\right\|_F \left\|\boldsymbol{D}_i^{-1}\right\|_2$$

$$= \left\|\boldsymbol{\Lambda}^i - \boldsymbol{U}^\top \boldsymbol{U}^{i*} \boldsymbol{\Lambda}^{i*} (\boldsymbol{V}^{i*})^\top \boldsymbol{V}\right\|_F \left\|\boldsymbol{D}_i^{-1}\right\|_2$$

$$\overset{(\xi_1)}{\le} 144 \delta'_{2r} \left\|\boldsymbol{\Lambda}^{i*}\right\|_F \left( \text{dist}(\boldsymbol{U}, \boldsymbol{U}^{i*}) + \text{dist}(\boldsymbol{V}, \boldsymbol{V}^{i*}) \right) \cdot 2$$

$$\le 288 \delta'_{2r} \left\|\boldsymbol{\Lambda}^{i*}\right\|_F \left( \text{dist}(\boldsymbol{U}, \boldsymbol{U}^*) + \text{dist}(\boldsymbol{V}, \boldsymbol{V}^*) + \text{dist}(\boldsymbol{U}^{i*}, \boldsymbol{U}^*) + \text{dist}(\boldsymbol{V}^{i*}, \boldsymbol{V}^*) \right)$$

$$\le 288 \delta'_{2r} \left\|\boldsymbol{\Lambda}^{i*}\right\|_F \left( \text{dist}(\boldsymbol{U}, \boldsymbol{U}^*) + \text{dist}(\boldsymbol{V}, \boldsymbol{V}^*) + 2\beta \right), \tag{60}$$

where $(\xi_1)$ follows from 37 and 58. Continuing from the above inequality, we obtain:

$$\textcircled{1} = \sum_{i=1}^{k}\sum_{j=1}^{N}\left\langle (G_j^i)^\top, X(\Lambda^i)^\top U^\top\right\rangle\left\langle (G_j^i)^\top, V^*\left(U\Lambda^i D_i^{-1} - UU^\top U^{i*}\Lambda^{i*}\right)^\top\right\rangle$$

$$\overset{(\xi_1)}{\le} \sum_{i=1}^{k}\sum_{j=1}^{N}\left\langle X(\Lambda^i)^\top U^\top, V^*\left(U\Lambda^i D_i^{-1} - UU^\top U^{i*}\Lambda^{i*}\right)^\top\right\rangle$$

$$+ 18m\delta_{3r}\max_{i=1}^{k}\left\|X(\Lambda^i)^\top U^\top\right\|_F\left\|V^*\left(U\Lambda^i D_i^{-1} - UU^\top U^{i*}\Lambda^{i*}\right)^\top\right\|_F$$

$$\overset{(\xi_2)}{\le} 2m\max_{i=1}^{k}\left\|X(\Lambda^i)^\top U^\top\right\|_F\left\|V^*\left(U\Lambda^i D_i^{-1} - UU^\top U^{i*}\Lambda^{i*}\right)^\top\right\|_F$$

$$\overset{(\xi_3)}{\le} 2m\left\|\Lambda^i\right\|_2\left\|U\Lambda^i D_i^{-1} - UU^\top U^{i*}\Lambda^{i*}\right\|_F,$$

where $(\xi_1)$ follows since the ensemble $\{G_j^i\}$ satisfies the 3r-GRIP and therefore Lemma 12 applies with the following choice of parameters:

$$\begin{aligned} U_1 &= X, & V_1 &= U, & \Lambda_1^i &= \Lambda^i, \\ U_2 &= V^*(D_i^{-1})^\top, & V_2 &= U, & \Lambda_2^i &= (\Lambda^i)^\top, \\ U_3 &= V^*, & V_3 &= U, & \Lambda_3^i &= (\Lambda^{i*})^\top(U^{i*})^\top U, \end{aligned}$$

$(\xi_2)$ follows from Cauchy-Schwartz and using $\delta_{3r} \le \frac{1}{18}$, and $(\xi_3)$ holds since $U, V^*$ have orthonormal columns and we have used the inequality $\|AB\|_F \le \|A\|_2\|B\|_F$. Finnaly, using 60, we conclude:

$$\textcircled{1} \le 2m\max_{i=1}^{k}\left\|\Lambda^i\right\|_2\left\|U\Lambda^i D^{-1} - UU^\top U^*\Lambda^{i*}\right\|_F$$

$$\le 576m\delta_{2r}'\cdot(\mathrm{dist}(U,U^*) + \mathrm{dist}(V,V^*) + 2\beta)\cdot\max_{i=1}^{k}\left(\left\|\Lambda^i\right\|_2\left\|\Lambda^{i*}\right\|_F\right)$$

$$\overset{(\xi_1)}{\le} 1152m\delta_{2r}'\cdot(\mathrm{dist}(U,U^*) + \mathrm{dist}(V,V^*) + 2\beta)\cdot\max_{i=1}^{k}\left(\left\|\Lambda^{i*}\right\|_2\left\|\Lambda^{i*}\right\|_F\right),$$

where $(\xi_1)$ follows from Lemma 6.

For the second part, applying C-S inequality, gives us:

$$\textcircled{2} = \sum_{i=1}^{k}\sum_{j=1}^{N}\left\langle (G_j^i)^\top, X(\Lambda^i)^\top U^\top\right\rangle\left\langle (G_j^i)^\top, V^*(D^{-1} - D_i^{-1})(\Lambda^i)^\top U^\top\right\rangle$$

$$\le \sqrt{\sum_{i=1}^{k}\sum_{j=1}^{N}\left\langle (G_j^i)^\top, X(\Lambda^i)^\top U^\top\right\rangle^2}\cdot\sqrt{\sum_{i=1}^{k}\sum_{j=1}^{N}\left\langle (G_j^i)^\top, V^*(D^{-1} - D_i^{-1})(\Lambda^i)^\top U^\top\right\rangle^2}$$

$$\overset{(\xi_1)}{\le} \sqrt{(1+\delta_{3r})m\max_{i=1}^{k}\left\|X(\Lambda^i)^\top U^\top\right\|_F^2}\cdot\sqrt{mA_{n,d,r}\max_{i=1}^{k}\left\|V^*(D^{-1} - D_i^{-1})(\Lambda^i)^\top U^\top\right\|_F^2}$$

$$\le \sqrt{2m\max_{i=1}^{k}\|X\|_F^2\|\Lambda^i\|_2^2}\cdot\sqrt{mA_{n,d,r}\max_{i=1}^{k}\left\|D^{-1} - D_i^{-1}\right\|_2^2\|(\Lambda^i)\|_F^2}$$

$$\overset{(\xi_2)}{\le} 8m\beta\sqrt{A_{n,d,r}}\max_{i=1}^{k}\left(\left\|\Lambda^i\right\|_2\left\|\Lambda^i\right\|_F\right)$$

$$\overset{(\xi_3)}{\le} 32m\beta\sqrt{A_{n,d,r}}\max_{i=1}^{k}\left(\left\|\Lambda^{i*}\right\|_2\left\|\Lambda^{i*}\right\|_F\right),$$

where $(\xi_1)$ follows since $\{G_j^i\}$ is sub-isometric with coefficient $A_{n,d,r}$ and satisfies $r$-GRIP with coefficient $\delta_{3r}$, $(\xi_2)$ from 59 and $(\xi_3)$ from Lemma 6.

Combining these two inequalities, we obtain:

$$\boldsymbol{x}^\top \boldsymbol{F}_1 = \text{①} + \text{②} \leq 1152m \cdot \max_{i=1}^{k} \left( \left\| \boldsymbol{\Lambda}^{i*} \right\|_2, \left\| \boldsymbol{\Lambda}^{i*} \right\|_F \right) \cdot \left( \delta'_{2r} \left( \text{dist}(\boldsymbol{U}, \boldsymbol{U}^*) + \text{dist}(\boldsymbol{V}, \boldsymbol{V}^*) \right) + \beta \sqrt{A_{n,d,r}} \right)$$

Since $\boldsymbol{x}$ was arbitrary, this completes the proof.

### D.10. Proof of Lemma 9

Note that we have:

$$\| \boldsymbol{F}_2 \| = \max_{\|\boldsymbol{x}\|=1} \boldsymbol{x}^\top \boldsymbol{F}_2.$$

Let $\boldsymbol{X} \in \mathbb{R}^{d \times r}$ be arbitrary with $\| \boldsymbol{X} \|_F = 1$, and define $\boldsymbol{x} = \text{vec}\,(\boldsymbol{X})$. We have

$$\boldsymbol{x}^\top \boldsymbol{F}_2 =$$
$$= \boldsymbol{x}^\top \left( \sum_{i=1}^{k} \sum_{j=1}^{N} (\boldsymbol{U}\boldsymbol{\Lambda}^i)^\top \otimes \boldsymbol{I}_d \cdot \text{vec}\,((\boldsymbol{G}_j^i)^\top) \, \text{vec}\,((\boldsymbol{G}_j^i)^\top)^\top \cdot \left( \boldsymbol{U}\boldsymbol{U}^\top \boldsymbol{U}^{i*} \boldsymbol{\Lambda}^{i*} \otimes \boldsymbol{I}_d - \boldsymbol{U}^{i*} \boldsymbol{\Lambda}^{i*} \otimes \boldsymbol{I}_d \right) \text{vec}\,(\boldsymbol{V}^*) \right)$$
$$= \sum_{i=1}^{k} \sum_{j=1}^{N} \text{vec}\,(\boldsymbol{X})^\top (\boldsymbol{U}\boldsymbol{\Lambda}^i)^\top \otimes \boldsymbol{I}_d \cdot \text{vec}\,((\boldsymbol{G}_j^i)^\top) \, \text{vec}\,((\boldsymbol{G}_j^i)^\top)^\top \cdot \left( \boldsymbol{U}\boldsymbol{U}^\top \boldsymbol{U}^{i*} \boldsymbol{\Lambda}^{i*} \otimes \boldsymbol{I}_d - \boldsymbol{U}^{i*} \boldsymbol{\Lambda}^{i*} \otimes \boldsymbol{I}_d \right) \text{vec}\,(\boldsymbol{V}^*)$$
$$= \sum_{i=1}^{k} \sum_{j=1}^{N} \text{vec}\,(\boldsymbol{X}(\boldsymbol{\Lambda}^i)^\top \boldsymbol{U}^\top)^\top \cdot \text{vec}\,((\boldsymbol{G}_j^i)^\top) \, \text{vec}\,((\boldsymbol{G}_j^i)^\top)^\top \cdot \text{vec}\,\left( \boldsymbol{V}^* \left( \left( \boldsymbol{U}\boldsymbol{U}^\top \boldsymbol{U}^{i*} - \boldsymbol{U}^{i*} \right) (\boldsymbol{\Lambda}^i)^* \right)^\top \right)$$
$$= \sum_{i=1}^{k} \sum_{j=1}^{N} \left\langle (\boldsymbol{G}_j^i)^\top, \boldsymbol{X}(\boldsymbol{\Lambda}^i)^\top \boldsymbol{U}^\top \right\rangle \cdot \left\langle (\boldsymbol{G}_j^i)^\top, \boldsymbol{V}^* \left( \left( \boldsymbol{U}\boldsymbol{U}^\top \boldsymbol{U}^{i*} - \boldsymbol{U}^{i*} \right) (\boldsymbol{\Lambda}^i)^* \right)^\top \right\rangle$$
$$= \underbrace{\sum_{i=1}^{k} \sum_{j=1}^{N} \left\langle (\boldsymbol{G}_j^i)^\top, \boldsymbol{X}(\boldsymbol{\Lambda}^i)^\top \boldsymbol{U}^\top \right\rangle \cdot \left\langle (\boldsymbol{G}_j^i)^\top, \boldsymbol{V}^* \left( \left( \boldsymbol{U}\boldsymbol{U}^\top \boldsymbol{U}^* - \boldsymbol{U}^* \right) (\boldsymbol{\Lambda}^i)^* \right)^\top \right\rangle}_{\text{①}}$$
$$+ \underbrace{\sum_{i=1}^{k} \sum_{j=1}^{N} \left\langle (\boldsymbol{G}_j^i)^\top, \boldsymbol{X}(\boldsymbol{\Lambda}^i)^\top \boldsymbol{U}^\top \right\rangle \cdot \left\langle (\boldsymbol{G}_j^i)^\top, \boldsymbol{V}^* \left( (\boldsymbol{U}\boldsymbol{U}^\top - \boldsymbol{I})(\boldsymbol{U}^* - \boldsymbol{U}^{i*})(\boldsymbol{\Lambda}^i)^* \right)^\top \right\rangle}_{\text{②}}$$

For the first part, we have:

$$\text{①} \overset{(\xi_1)}{\leq} \sum_{i=1}^{k} \sum_{j=1}^{N} \left\langle \boldsymbol{X}(\boldsymbol{\Lambda}^i)^\top \boldsymbol{U}^\top, \boldsymbol{V}^* \left( \left( \boldsymbol{U}\boldsymbol{U}^\top \boldsymbol{U}^* - \boldsymbol{U}^* \right) (\boldsymbol{\Lambda}^i)^* \right)^\top \right\rangle$$
$$\qquad + 18m\delta_{3r} \max_{i=1}^{k} \left\| \boldsymbol{X}(\boldsymbol{\Lambda}^i)^\top \boldsymbol{U}^\top \right\|_F \cdot \left\| \left( \boldsymbol{U}\boldsymbol{U}^\top \boldsymbol{U}^* - \boldsymbol{U}^* \right) (\boldsymbol{\Lambda}^i)^* (\boldsymbol{V}^*)^\top \right\|_F$$
$$\overset{(\xi_2)}{=} 18m\delta_{3r} \max_{i=1}^{k} \left\| \boldsymbol{X}(\boldsymbol{\Lambda}^i)^\top \boldsymbol{U}^\top \right\|_F \cdot \left\| \left( \boldsymbol{U}\boldsymbol{U}^\top \boldsymbol{U}^* - \boldsymbol{U}^* \right) (\boldsymbol{\Lambda}^i)^* (\boldsymbol{V}^*)^\top \right\|_F$$
$$\overset{(\xi_3)}{=} 18m\delta_{3r} \max_{i=1}^{k} \left\| \boldsymbol{\Lambda}^i \right\|_2 \left\| \boldsymbol{\Lambda}^{i*} \right\|_F \left\| \boldsymbol{U}\boldsymbol{U}^\top \boldsymbol{U}^* - \boldsymbol{U}^* \right\|_2$$
$$\overset{(\xi_4)}{\leq} 36m\delta_{3r} \cdot \text{dist}(\boldsymbol{U}, \boldsymbol{U}^*) \cdot \max_{i=1}^{k} \left( \left\| \boldsymbol{\Lambda}^{i*} \right\|_2, \left\| \boldsymbol{\Lambda}^{i*} \right\|_F \right),$$

where $(\xi_1)$ follows since the ensemble $\{\boldsymbol{G}_j^i\}$ satisfies the 3r-GRIP condition and therefore Lemma 12 applies with the following choice of parameters:

$$
\begin{aligned}
\boldsymbol{U}_1 &= \boldsymbol{X}, & \boldsymbol{V}_1 &= \boldsymbol{U}, & \boldsymbol{\Lambda}_1^i &= (\boldsymbol{\Lambda}^i)^\top, \\
\boldsymbol{U}_2 &= \boldsymbol{V}^{,} & \boldsymbol{V}_2 &= \boldsymbol{U}\boldsymbol{U}^\top\boldsymbol{U}^* - \boldsymbol{U}^*, & \boldsymbol{\Lambda}_2^i &= (\boldsymbol{\Lambda}^{i*})^\top, \\
\boldsymbol{U}_3 &= 0, & \boldsymbol{V}_3 &= 0, & \boldsymbol{\Lambda}_3^i &= 0,
\end{aligned}
$$

$(\xi_2)$ follows since we have:

$$
\left\langle \boldsymbol{X}(\boldsymbol{\Lambda}^i)^\top \boldsymbol{U}^\top, \boldsymbol{V}^* \left( \left(\boldsymbol{U}\boldsymbol{U}^\top\boldsymbol{U}^* - \boldsymbol{U}^*\right)(\boldsymbol{\Lambda}^i)^*\right)^\top \right\rangle = \mathrm{tr}\left( \boldsymbol{X}(\boldsymbol{\Lambda}^i)^\top \underbrace{\boldsymbol{U}^\top\left(\boldsymbol{U}\boldsymbol{U}^\top\boldsymbol{U}^* - \boldsymbol{U}^*\right)}_{=0}(\boldsymbol{\Lambda}^i)^*(\boldsymbol{V}^*)^\top \right)
$$
$$
= \boldsymbol{0},
$$

$(\xi_3)$ holds using $\boldsymbol{U}, \boldsymbol{V}^*$ have orthonormal columns and the inequality $\|\boldsymbol{A}\boldsymbol{B}\|_F \le \|\boldsymbol{A}\|_2 \|\boldsymbol{B}\|_F$ and $(\xi_4)$ holds from Lemma 6.

For the second part, using C-S inequality, we get:

$$
\begin{aligned}
\textcircled{2} &= \sum_{i=1}^k \sum_{j=1}^N \left\langle (\boldsymbol{G}_j^i)^\top, \boldsymbol{X}(\boldsymbol{\Lambda}^i)^\top\boldsymbol{U}^\top \right\rangle \cdot \left\langle (\boldsymbol{G}_j^i)^\top, \boldsymbol{V}^* \left((\boldsymbol{U}\boldsymbol{U}^\top - \boldsymbol{I})(\boldsymbol{U}^* - \boldsymbol{U}^{i*})(\boldsymbol{\Lambda}^i)^*\right)^\top \right\rangle \\
&\le \sqrt{\sum_{i=1}^k \sum_{j=1}^N \left\langle (\boldsymbol{G}_j^i)^\top, \boldsymbol{X}(\boldsymbol{\Lambda}^i)^\top\boldsymbol{U}^\top \right\rangle^2} \cdot \sqrt{\sum_{i=1}^k \sum_{j=1}^N \left\langle (\boldsymbol{G}_j^i)^\top, \boldsymbol{V}^* \left((\boldsymbol{U}\boldsymbol{U}^\top - \boldsymbol{I})(\boldsymbol{U}^* - \boldsymbol{U}^{i*})(\boldsymbol{\Lambda}^i)^*\right)^\top \right\rangle^2} \\
&\overset{(\xi_1)}{\le} \sqrt{(1+\delta_{3r})m \max_{i=1}^k \|\boldsymbol{X}(\boldsymbol{\Lambda}^i)^\top\boldsymbol{U}^\top\|_F^2} \cdot \sqrt{mA_{n,d,r} \max_{i=1}^k \left\|\boldsymbol{V}^* \left((\boldsymbol{U}\boldsymbol{U}^\top - \boldsymbol{I})(\boldsymbol{U}^* - \boldsymbol{U}^{i*})(\boldsymbol{\Lambda}^i)^*\right)^\top\right\|_F^2} \\
&\le \sqrt{2m \max_{i=1}^k \|\boldsymbol{X}\|_F^2 \|\boldsymbol{\Lambda}^i\|_2^2} \cdot \sqrt{mA_{n,d,r} \max_{i=1}^k \|\boldsymbol{U}\boldsymbol{U}^\top - \boldsymbol{I}\|_2^2 \|\boldsymbol{U}^* - \boldsymbol{U}^{i*}\|_2^2 \|\boldsymbol{\Lambda}^{i*}\|_F^2} \\
&\overset{(\xi_2)}{\le} 2m \cdot \beta\sqrt{A_{n,d,r}} \cdot \max_{i=1}^k \left(\|\boldsymbol{\Lambda}^i\|_2 \|\boldsymbol{\Lambda}^{i*}\|_F\right) \\
&\le 4m \cdot \beta\sqrt{A_{n,d,r}} \cdot \max_{i=1}^k \left(\|\boldsymbol{\Lambda}^{i*}\|_2 \|\boldsymbol{\Lambda}^{i*}\|_F\right),
\end{aligned}
$$

where $(\xi_1)$ follows since $\{\boldsymbol{G}_j^i\}$ is sub-isometric with coefficient $A_{n,d,r}$ and satisfies $r$-GRIP with coefficient $\delta_{3r}$, in $(\xi_2)$ we used 53, and $(\xi_3)$ follows from Lemma 6.

Combining these bounds, we conclude:

$$
\boldsymbol{x}^\top \boldsymbol{F}_2 = \textcircled{1} + \textcircled{2} \le 36m \cdot \max_{i=1}^k \left(\|\boldsymbol{\Lambda}^{i*}\|_2 \|\boldsymbol{\Lambda}^{i*}\|_F\right) \cdot \left(\delta_{3r}\mathrm{dist}(\boldsymbol{U}, \boldsymbol{U}^*) + \beta\sqrt{A_{n,d,r}}\right)
$$

Since $\boldsymbol{x}$ was arbitrary, this completes the proof.

### D.11. Proof of Lemma 10

We have:

$$
\begin{aligned}
\sigma_{\min}(\boldsymbol{R}_{t+1}) &= \min_{\|\boldsymbol{z}\|=1} \|\boldsymbol{R}_{t+1}\boldsymbol{z}\|_2 = \min_{\|\boldsymbol{z}\|=1} \|\boldsymbol{V}_{t+1}\boldsymbol{R}_{t+1}\boldsymbol{z}\|_2 , \\
&= \min_{\|\boldsymbol{z}\|=1} \left\|\widehat{\boldsymbol{V}}_{t+1}\boldsymbol{z}\right\|_2 = \min_{\|\boldsymbol{z}\|=1} \left\|\boldsymbol{V}^*((\boldsymbol{D}_t)^{-1})^\top \boldsymbol{z} - \mathrm{mat}(\boldsymbol{H}_t)\boldsymbol{z}\right\|_2 \\
&\geq \min_{\|\boldsymbol{z}\|=1} \left\|\boldsymbol{V}^*((\boldsymbol{D}_t)^{-1})^\top \boldsymbol{z}\right\|_2 - \max_{\|\boldsymbol{z}\|=1} \|\mathrm{mat}(\boldsymbol{H}_t)\boldsymbol{z}\|_2 \\
&= \min_{\|\boldsymbol{z}\|=1} \left\|((\boldsymbol{D}_t)^{-1})^\top \boldsymbol{z}\right\|_2 - \|\mathrm{mat}(\boldsymbol{H}_t)\|_2 \\
&= \sigma_{\min}\left((\boldsymbol{D}_t)^{-1}\right) - \|\mathrm{mat}(\boldsymbol{H}_t)\|_2 \\
&= \frac{1}{\sigma_{\max}(\boldsymbol{D}_t)} - \|\mathrm{mat}(\boldsymbol{H}_t)\|_2 \\
&= \frac{1}{\|(\boldsymbol{V}^*)^\top \boldsymbol{V}_t\|_2} - \|\mathrm{mat}(\boldsymbol{H}_t)\|_2 \\
&\geq \frac{1}{\|\boldsymbol{V}^*\|_2 \|\boldsymbol{V}_t\|_2} - \|\mathrm{mat}(\boldsymbol{H}_t)\|_2 \\
&\geq 1 - \|\mathrm{mat}(\boldsymbol{H}_t)\|_2 .
\end{aligned}
$$

Using 48 completes the proof.

### D.12. Proof of Proposition 1

Let's define the operators $\mathcal{A} : R^{d\times r} \times R^{d\times r} \times R^{k\cdot(r\times r)} \to R^m$ and $\mathcal{B} : R^{d\times r} \times R^{d\times r} \times R^{k\cdot(r\times r)} \to R^m$ as follows:

$$
\mathcal{A}\left(\boldsymbol{U}, \boldsymbol{V}, \{\boldsymbol{\Lambda}\}_{i=1}^k\right) = \begin{pmatrix} \frac{1}{\sqrt{m}}\left\langle \boldsymbol{G}_1^1, \boldsymbol{U}\boldsymbol{\Lambda}^1 \boldsymbol{V}^\top \right\rangle \\ \vdots \\ \frac{1}{\sqrt{m}}\left\langle \boldsymbol{G}_n^1, \boldsymbol{U}\boldsymbol{\Lambda}^1 \boldsymbol{V}^\top \right\rangle \\ \vdots \\ \vdots \\ \frac{1}{\sqrt{m}}\left\langle \boldsymbol{G}_1^k, \boldsymbol{U}\boldsymbol{\Lambda}^k \boldsymbol{V}^\top \right\rangle \\ \vdots \\ \frac{1}{\sqrt{m}}\left\langle \boldsymbol{G}_n^k, \boldsymbol{U}\boldsymbol{\Lambda}^k \boldsymbol{V}^\top \right\rangle \end{pmatrix}, \mathcal{B}\left(\boldsymbol{U}, \boldsymbol{V}, \{\boldsymbol{\Lambda}\}_{i=1}^k\right) = \begin{pmatrix} \frac{1}{\sqrt{m}}\left\|\boldsymbol{U}\boldsymbol{\Lambda}^1 \boldsymbol{V}^\top\right\|_F \\ \vdots \\ \frac{1}{\sqrt{m}}\left\|\boldsymbol{U}\boldsymbol{\Lambda}^1 \boldsymbol{V}^\top\right\|_F \\ \vdots \\ \vdots \\ \frac{1}{\sqrt{m}}\left\|\boldsymbol{U}\boldsymbol{\Lambda}^k \boldsymbol{V}^\top\right\|_F \\ \vdots \\ \frac{1}{\sqrt{m}}\left\|\boldsymbol{U}\boldsymbol{\Lambda}^k \boldsymbol{V}^\top\right\|_F . \end{pmatrix}
$$

We need to show that with high probabillity, for all $\boldsymbol{U}, \boldsymbol{V}$ and $\{\boldsymbol{\Lambda}^i\}_{i=1}^k$, the following inequality holds:

$$
\left|\left\|\mathcal{A}\left(\boldsymbol{U}, \boldsymbol{V}, \{\boldsymbol{\Lambda}\}_{i=1}^k\right)\right\|_2^2 - \left\|\mathcal{B}\left(\boldsymbol{U}, \boldsymbol{V}, \{\boldsymbol{\Lambda}\}_{i=1}^k\right)\right\|_2^2\right| \leq \delta_r \max_{i=1}^k \left\|\boldsymbol{U}\boldsymbol{\Lambda}^i(\boldsymbol{V})^\top\right\|_F^2 . \tag{61}
$$

**Observation 1.** *Note that for $\boldsymbol{U}, \boldsymbol{V}, \{\boldsymbol{\Lambda}^i\}_{i=1}^k$ and invertible matrices $\boldsymbol{R}_1, \boldsymbol{R}_2 \in R^{r\times r}$, under the transformation $\boldsymbol{U} \leftarrow \boldsymbol{U}\boldsymbol{R}_1, \boldsymbol{V} \leftarrow \boldsymbol{V}\boldsymbol{R}_2$ and $\boldsymbol{\Lambda}^i \leftarrow (\boldsymbol{R}_1)^{-1}\boldsymbol{\Lambda}(\boldsymbol{R}_2^\top)^{-1}$, we have that*

$$
\mathcal{A}\left(\boldsymbol{U}, \boldsymbol{V}, \{\boldsymbol{\Lambda}^i\}_{i=1}^k\right), \mathcal{B}\left(\boldsymbol{U}, \boldsymbol{V}, \{\boldsymbol{\Lambda}^i\}_{i=1}^k\right) \text{ and } \boldsymbol{U}\boldsymbol{\Lambda}^i \boldsymbol{V}^\top
$$

*stay the same.*

Therefore, it suffices to prove that 61 holds for all $\boldsymbol{U}, \boldsymbol{V} \in \mathrm{St}(r, d)$. Moreover, since 61 is scale-invariant, we may impose $\left\|\boldsymbol{\Lambda}^i\right\|_F \leq 1$ for every $i = 1, \dots, k$ and establish the bound with the same constant $\delta_r$. We define the set $S$ as follows:

$$
S = \{(\boldsymbol{U}, \boldsymbol{V}, \{\boldsymbol{\Lambda}^i\}_{i=1}^k) : \boldsymbol{U}, \boldsymbol{V} \in \mathrm{St}(r, d), \{\boldsymbol{\Lambda}^i\}_{i=1}^k \in R^{k\cdot(r\times r)} \text{ and } \max_{i=1}^k \left\|\boldsymbol{\Lambda}^i\right\|_F \leq 1\}.
$$

Let $\delta = \bar{\delta}_r$. Let $S_1, S_2$ be a $\delta$-covers for $\text{St}(r,d) \subseteq B(\sqrt{r}, dr)$ and $\mathrm{B}(1, r^2)$ respectively. Have $|S_1| \leq \left(\frac{9\sqrt{r}}{\delta}\right)^{dr}$ and $|S_2| \leq \left(\frac{3}{\delta}\right)^{r^2}$ (Wainwright, 2019). Define $\bar{S}$ as follows:

$$\bar{S} = \{(\bar{U}, \bar{V}, \{\bar{\Lambda}^i\}_{i=1}^k) : \bar{U}, \bar{V} \in S_1, \{\bar{\Lambda}^i\}_{i=1}^k \in (S_2)^k\}.$$

Now, for a fixed $(\bar{U}, \bar{V}, \{\bar{\Lambda}^i\}_{i=1}^k) \in \bar{S}$, by Lemma 11, we have:

$$\mathbf{P}\left[\left|\left\|\mathcal{A}\left(\bar{U}, \bar{V}, \{\bar{\Lambda}^i\}_{i=1}^k\right)\right\|_2^2 - \left\|\mathcal{B}\left(\bar{U}, \bar{V}, \{\bar{\Lambda}^i\}_{i=1}^k\right)\right\|_2^2\right| \leq \delta \max_{i=1}^k \left\|\bar{\Lambda}^i\right\|_F^2\right] \geq 1 - C_1 e^{-cm\delta^2}. \tag{62}$$

Let $A$ be the event such that:

$$\left|\left\|\mathcal{A}\left(\bar{U}, \bar{V}, \{\bar{\Lambda}\}_{i=1}^k\right)\right\|_2^2 - \left\|\mathcal{B}\left(\bar{U}, \bar{V}, \{\bar{\Lambda}^i\}_{i=1}^k\right)\right\|_2^2\right| \leq \delta \text{ for all } (\bar{U}, \bar{V}, \{\bar{\Lambda}^i\}_{i=1}^k) \in \bar{S}$$

Using 62 and applying union bound gives us:

$$\mathbf{P}[A] \geq 1 - C_1|\bar{S}|e^{-cm\delta^2} \geq 1 - C_1\left(\frac{9\sqrt{r}}{\delta}\right)^{dr+kr^2} \cdot e^{-cm\delta^2}$$

$$\geq 1 - C_1 e^{-cm\delta^2 + (dr+kr^2)\log\left(\frac{9\sqrt{r}}{\delta}\right)} \geq 1 - C_1^{-\frac{cm\delta^2}{2}},$$

where the last inequality follows from the fact that $m \geq \frac{2(dr+kr^2)}{c\delta^2}\log\left(\frac{9\sqrt{r}}{\delta}\right)$. We now show that if $A$ holds, then inequality 61 follows. From this point onward, assume that $A$ holds.

We set

$$\gamma := \sup\left\{\left|\left\|\mathcal{A}\left(U, V, \{\Lambda^i\}_{i=1}^k\right)\right\|_2 - \left\|\mathcal{B}\left(U, V, \{\Lambda^i\}_{i=1}^k\right)\right\|_2\right| \Big| (U, V, \{\Lambda^i\}_{i=1}^k) \in S\right\}.$$

Now, for a fixed $(U, V, \{\Lambda^i\}_{i=1}^k) \in S$, let $(\bar{U}, \bar{V}, \{\bar{\Lambda}^i\}_{i=1}^k) \in \bar{S}$ such that we have:

$$\left\|U - \bar{U}\right\|_F \leq \delta, \left\|V - \bar{V}\right\|_F \leq \delta, \text{ and } \left\|\Lambda^i - \bar{\Lambda}^i\right\|_F \leq \delta \text{ for } i = 1, \ldots k.$$

Now using the linearity of $\mathcal{A}$ and applying triangle inequality we obtain:

$$
\begin{aligned}
\left|\left\|\mathcal{A}\left(U, V, \{\Lambda^i\}_{i=1}^k\right)\right\|_2 - \left\|\mathcal{B}\left(U, V, \{\Lambda^i\}_{i=1}^k\right)\right\|_2\right| \leq \quad & \left\|\mathcal{A}\left(U - \bar{U}, V, \{\Lambda^i\}_{i=1}^k\right)\right\|_2 \\
& + \left\|\mathcal{A}\left(\bar{U}, V - \bar{V}, \{\Lambda^i\}_{i=1}^k\right)\right\|_2 \\
& + \left\|\mathcal{A}\left(\bar{U}, \bar{V}, \{\Lambda^i - \bar{\Lambda}^i\}_{i=1}^k\right)\right\|_2 \\
& + \underbrace{\left|\left\|\mathcal{A}\left(\bar{U}, \bar{V}, \{\bar{\Lambda}^i\}_{i=1}^k\right)\right\|_2 - \left\|\mathcal{B}\left(\bar{U}, \bar{V}, \{\bar{\Lambda}^i\}_{i=1}^k\right)\right\|_2\right|}_{\text{②}} \\
& + \underbrace{\left|\left\|\mathcal{B}\left(\bar{U}, \bar{V}, \{\bar{\Lambda}^i\}_{i=1}^k\right)\right\|_2 - \left\|\mathcal{B}\left(U, V, \{\Lambda^i\}_{i=1}^k\right)\right\|_2\right|}_{\text{①}}.
\end{aligned}
$$

Observe that for every $i$ we have:

$$\left\|(U - \bar{U})\Lambda^i(V)^\top\right\|_F \leq \left\|(U - \bar{U})\right\|_F \left\|\Lambda^i(V)^\top\right\|_F \leq \left\|(U - \bar{U})\right\|_F \left\|\Lambda^i\right\|_F \leq \delta,$$

and similarly:

$$\left\|\bar{U}\Lambda^i(V - \bar{V})^\top\right\|_F \leq \delta, \left\|\bar{U}(\Lambda^i - \bar{\Lambda}^i)\bar{V}^\top\right\|_F \leq \delta.$$

If $U = \bar{U}$ we have $\left\|\mathcal{A}\left(U - \bar{U}, V, \{\Lambda^i\}_{i=1}^k\right)\right\|_2 = 0$. Otherwise, by homogeneity we have:

$$\left\|\mathcal{A}\left(U - \bar{U}, V, \{\Lambda^i\}_{i=1}^k\right)\right\|_2 - \left\|\mathcal{B}\left(U - \bar{U}, V, \{\Lambda^i\}_{i=1}^k\right)\right\|_2 =$$

$$\left\|U - \bar{U}\right\|_F \left(\left\|\mathcal{A}\left(\frac{U - \bar{U}}{\left\|U - \bar{U}\right\|_F}, V, \{\Lambda^i\}_{i=1}^k\right)\right\|_2 - \left\|\mathcal{B}\left(\frac{U - \bar{U}}{\left\|U - \bar{U}\right\|_F}, V, \{\Lambda^i\}_{i=1}^k\right)\right\|_2\right)$$

Now, recalling Observation 1 and noting that for every $i = 1, \ldots, k$,

$$\left\| \frac{\boldsymbol{U} - \bar{\boldsymbol{U}}}{\|\boldsymbol{U} - \bar{\boldsymbol{U}}\|_F} \boldsymbol{\Lambda}^i \boldsymbol{V}^\top \right\|_F \leq \|\boldsymbol{\Lambda}^i\|_F \leq 1,$$

we conclude that the second term is bounded by $\gamma$, which yields:

$$\left\| \mathcal{A} \left( \boldsymbol{U} - \bar{\boldsymbol{U}}, \boldsymbol{V}, \{\boldsymbol{\Lambda}^i\}_{i=1}^k \right) \right\|_2 \leq \left\| \mathcal{B} \left( \boldsymbol{U} - \bar{\boldsymbol{U}}, \boldsymbol{V}, \{\boldsymbol{\Lambda}^i\}_{i=1}^k \right) \right\|_2 + \delta\gamma \leq \delta + \delta\gamma,$$

where the final inequality follows from the fact that all entries of $\mathcal{B} \left( \boldsymbol{U} - \bar{\boldsymbol{U}}, \boldsymbol{V}, \{\boldsymbol{\Lambda}^i\}_{i=1}^k \right)$ are bounded by $\frac{\delta}{\sqrt{m}}$. Similarly, we have

$$\left\| \mathcal{A} \left( \bar{\boldsymbol{U}}, \boldsymbol{V} - \bar{\boldsymbol{V}}, \{\boldsymbol{\Lambda}^i\}_{i=1}^k \right) \right\|_2 \leq \delta + \delta\gamma, \tag{63}$$

$$\left\| \mathcal{A} \left( \bar{\boldsymbol{U}}, \bar{\boldsymbol{V}}, \{\boldsymbol{\Lambda}^i - \bar{\boldsymbol{\Lambda}}^i\}_{i=1}^k \right) \right\|_2 \leq \delta + \delta\gamma. \tag{64}$$

63 follows identically and for 64 one needs to multiply and divide by $\max_{i=1}^k \|\boldsymbol{\Lambda}^i - \bar{\boldsymbol{\Lambda}}^i\|_F$. For ① observe that we have:

$$\left| \left\| \mathcal{B} \left( \bar{\boldsymbol{U}}, \bar{\boldsymbol{V}}, \{\bar{\boldsymbol{\Lambda}}^i\}_{i=1}^k \right) \right\|_2 - \left\| \mathcal{B} \left( \boldsymbol{U}, \boldsymbol{V}, \{\boldsymbol{\Lambda}^i\}_{i=1}^k \right) \right\|_2 \right| \leq \left\| \mathcal{B} \left( \bar{\boldsymbol{U}}, \bar{\boldsymbol{V}}, \{\bar{\boldsymbol{\Lambda}}^i\}_{i=1}^k \right) - \mathcal{B} \left( \boldsymbol{U}, \boldsymbol{V}, \{\boldsymbol{\Lambda}^i\}_{i=1}^k \right) \right\|_2.$$

Note that, for every $i = 1, \ldots k$, we have:

$$\left\| \boldsymbol{U}\boldsymbol{\Lambda}^i(\boldsymbol{V})^\top \right\|_F - \left\| \bar{\boldsymbol{U}}\bar{\boldsymbol{\Lambda}}^i(\bar{\boldsymbol{V}})^\top \right\|_F \leq \left\| (\boldsymbol{U} - \bar{\boldsymbol{U}})\boldsymbol{\Lambda}^i\boldsymbol{V}^\top \right\|_F + \left\| \bar{\boldsymbol{U}}(\boldsymbol{\Lambda}^i - \bar{\boldsymbol{\Lambda}}^i)\boldsymbol{V}^\top \right\|_F + \left\| \bar{\boldsymbol{U}}\bar{\boldsymbol{\Lambda}}^i(\boldsymbol{V} - \bar{\boldsymbol{V}})^\top \right\|_F \leq 3\delta,$$

which gives us:

$$① \leq \left\| \mathcal{B} \left( \bar{\boldsymbol{U}}, \bar{\boldsymbol{V}}, \{\bar{\boldsymbol{\Lambda}}^i\}_{i=1}^k \right) - \mathcal{B} \left( \boldsymbol{U}, \boldsymbol{V}, \{\boldsymbol{\Lambda}^i\}_{i=1}^k \right) \right\|_2 \leq 3\delta.$$

For ② we have:

$$\left| \left\| \mathcal{A} \left( \bar{\boldsymbol{U}}, \bar{\boldsymbol{V}}, \{\bar{\boldsymbol{\Lambda}}^i\}_{i=1}^k \right) \right\|_2 - \left\| \mathcal{B} \left( \bar{\boldsymbol{U}}, \bar{\boldsymbol{V}}, \{\bar{\boldsymbol{\Lambda}}^i\}_{i=1}^k \right) \right\|_2 \right| = \frac{\left| \left\| \mathcal{A} \left( \bar{\boldsymbol{U}}, \bar{\boldsymbol{V}}, \{\bar{\boldsymbol{\Lambda}}^i\}_{i=1}^k \right) \right\|_2^2 - \left\| \mathcal{B} \left( \bar{\boldsymbol{U}}, \bar{\boldsymbol{V}}, \{\bar{\boldsymbol{\Lambda}}^i\}_{i=1}^k \right) \right\|_2^2 \right|}{\left\| \mathcal{A} \left( \bar{\boldsymbol{U}}, \bar{\boldsymbol{V}}, \{\bar{\boldsymbol{\Lambda}}^i\}_{i=1}^k \right) \right\|_2 + \left\| \mathcal{B} \left( \bar{\boldsymbol{U}}, \bar{\boldsymbol{V}}, \{\bar{\boldsymbol{\Lambda}}^i\}_{i=1}^k \right) \right\|_2}$$

$$\leq \frac{\delta \max_{i=1}^k \|\bar{\boldsymbol{\Lambda}}^i\|_F^2}{\left\| \mathcal{A} \left( \bar{\boldsymbol{U}}, \bar{\boldsymbol{V}}, \{\bar{\boldsymbol{\Lambda}}^i\}_{i=1}^k \right) \right\|_2 + \left\| \mathcal{B} \left( \bar{\boldsymbol{U}}, \bar{\boldsymbol{V}}, \{\bar{\boldsymbol{\Lambda}}^i\}_{i=1}^k \right) \right\|_2},$$

which can be further bounded

$$\leq \min \left( \frac{\delta}{\left\| \mathcal{A} \left( \bar{\boldsymbol{U}}, \bar{\boldsymbol{V}}, \{\bar{\boldsymbol{\Lambda}}^i\}_{i=1}^k \right) \right\|_2 + \left\| \mathcal{B} \left( \bar{\boldsymbol{U}}, \bar{\boldsymbol{V}}, \{\bar{\boldsymbol{\Lambda}}^i\}_{i=1}^k \right) \right\|_2}, \frac{\delta \max_{i=1}^k \|\bar{\boldsymbol{\Lambda}}^i\|_F^2}{\left\| \mathcal{B} \left( \bar{\boldsymbol{U}}, \bar{\boldsymbol{V}}, \{\bar{\boldsymbol{\Lambda}}^i\}_{i=1}^k \right) \right\|_2} \right)$$

$$\leq \min \left( \frac{\delta}{\left\| \mathcal{A} \left( \bar{\boldsymbol{U}}, \bar{\boldsymbol{V}}, \{\bar{\boldsymbol{\Lambda}}^i\}_{i=1}^k \right) \right\|_2 + \left\| \mathcal{B} \left( \bar{\boldsymbol{U}}, \bar{\boldsymbol{V}}, \{\bar{\boldsymbol{\Lambda}}^i\}_{i=1}^k \right) \right\|_2}, \delta\sqrt{k} \right),$$

where the second inequality holds since

$$\left\| \mathcal{B} \left( \bar{\boldsymbol{U}}, \bar{\boldsymbol{V}}, \{\bar{\boldsymbol{\Lambda}}^i\}_{i=1}^k \right) \right\|_2 = \sqrt{\frac{n}{m} \sum_{i=1}^k \|\bar{\boldsymbol{\Lambda}}^i\|_F^2} \geq \frac{1}{\sqrt{k}} \max_{i=1}^k \|\bar{\boldsymbol{\Lambda}}^i\|_F.$$

Moreover, using triangle inequality, we get:

$$\left| \left\| \mathcal{A} \left( \bar{\boldsymbol{U}}, \bar{\boldsymbol{V}}, \{\bar{\boldsymbol{\Lambda}}^i\}_{i=1}^k \right) \right\|_2 - \left\| \mathcal{B} \left( \bar{\boldsymbol{U}}, \bar{\boldsymbol{V}}, \{\bar{\boldsymbol{\Lambda}}^i\}_{i=1}^k \right) \right\|_2 \right| \leq \left\| \mathcal{A} \left( \bar{\boldsymbol{U}}, \bar{\boldsymbol{V}}, \{\bar{\boldsymbol{\Lambda}}^i\}_{i=1}^k \right) \right\|_2 + \left\| \mathcal{B} \left( \bar{\boldsymbol{U}}, \bar{\boldsymbol{V}}, \{\bar{\boldsymbol{\Lambda}}^i\}_{i=1}^k \right) \right\|_2$$

Combigning both bounds, gives us:

$$\left\| \mathcal{A} \left( \bar{\boldsymbol{U}}, \bar{\boldsymbol{V}}, \{\bar{\boldsymbol{\Lambda}}^i\}_{i=1}^k \right) \right\|_2 - \left\| \mathcal{B} \left( \bar{\boldsymbol{U}}, \bar{\boldsymbol{V}}, \{\bar{\boldsymbol{\Lambda}}^i\}_{i=1}^k \right) \right\|_2 \leq \min \left( \delta\sqrt{k}, \sqrt{\delta} \right).$$

Putting together all bounds we obtain:

$$\left\| \mathcal{A} \left( \boldsymbol{U}, \boldsymbol{V}, \{\boldsymbol{\Lambda}^i\}_{i=1}^k \right) \right\|_2 - \left\| \mathcal{B} \left( \boldsymbol{U}, \boldsymbol{V}, \{\boldsymbol{\Lambda}^i\}_{i=1}^k \right) \right\|_2 \le 3\delta + 3\delta\gamma + \min\left( \delta\sqrt{k}, \sqrt{\delta} \right) + 3\delta \le 7\min\left( \delta\sqrt{k}, \sqrt{\delta} \right) + 3\delta\gamma$$

Taking supremum over $S$ gives:

$$\gamma \le 7\min\left( \delta\sqrt{k}, \sqrt{\delta} \right) + 3\delta\gamma,$$

from which we get:

$$\gamma \le 14\min\left( \delta\sqrt{k}, \sqrt{\delta} \right) = 14c(k,\delta).$$

Hence, for all $(\boldsymbol{U}, \boldsymbol{V}, \{\boldsymbol{\Lambda}\}_{i=1}^k) \in S$ we have:

$$\begin{aligned}
\left| \left\| \mathcal{A} \left( \boldsymbol{U}, \boldsymbol{V}, \{\boldsymbol{\Lambda}^i\}_{i=1}^k \right) \right\|_2^2 - \left\| \mathcal{B} \left( \boldsymbol{U}, \boldsymbol{V}, \{\boldsymbol{\Lambda}^i\}_{i=1}^k \right) \right\|_2^2 \right| &= \left| \left\| \mathcal{A} \left( \boldsymbol{U}, \boldsymbol{V}, \{\boldsymbol{\Lambda}^i\}_{i=1}^k \right) \right\|_2 - \left\| \mathcal{B} \left( \boldsymbol{U}, \boldsymbol{V}, \{\boldsymbol{\Lambda}^i\}_{i=1}^k \right) \right\|_2 \right| \\
&\quad \cdot \left| \left\| \mathcal{A} \left( \boldsymbol{U}, \boldsymbol{V}, \{\boldsymbol{\Lambda}^i\}_{i=1}^k \right) \right\|_2 + \left\| \mathcal{B} \left( \boldsymbol{U}, \boldsymbol{V}, \{\boldsymbol{\Lambda}^i\}_{i=1}^k \right) \right\|_2 \right| \\
&\le 14c(k,\delta) \left( 2 \left\| \mathcal{B} \left( \boldsymbol{U}, \boldsymbol{V}, \{\boldsymbol{\Lambda}^i\}_{i=1}^k \right) \right\|_2 + 14c(k,\delta) \right) \\
&\le 14c(k,\delta) \left( 2 + 14c(k,\delta) \right) \\
&\le 50\min\left( \delta\sqrt{k}, \sqrt{\delta} \right). \\
&\le \delta_r.
\end{aligned}$$

### D.13. Proof of Lemma 12

Let's define

$$\boldsymbol{U} = \left[ \boldsymbol{U}_1, \boldsymbol{U}_2, \boldsymbol{U}_3 \right] \in R^{d\times 3r}, \boldsymbol{V} = \left[ \boldsymbol{V}_1, \boldsymbol{V}_2, \boldsymbol{V}_3 \right] \in R^{d\times 3r}, \boldsymbol{\Lambda}^i = \mathbf{diag}(\boldsymbol{\Lambda}_1^i, \boldsymbol{\Lambda}_2^i, \boldsymbol{\Lambda}_3^i) \in R^{3r\times 3r}.$$

Note that we have:

$$\boldsymbol{X}^i + \boldsymbol{Y}^i = \boldsymbol{U}_1\boldsymbol{\Lambda}_1^i(\boldsymbol{V}_1)^\top + \boldsymbol{U}_2\boldsymbol{\Lambda}_2^i(\boldsymbol{V}_2)^\top + \boldsymbol{U}_3\boldsymbol{\Lambda}_3^i(\boldsymbol{V}_3)^\top = \boldsymbol{U}\boldsymbol{\Lambda}^i(\boldsymbol{V})^\top.$$

Since the ensemble $\{\boldsymbol{G}_j^i\}$ satisfies $\delta_{3r}$-GRIP, than we have:

$$\left| \sum_{i=1}^k \sum_{j=1}^N \left\langle \boldsymbol{G}_j^i, \boldsymbol{X}^i + \boldsymbol{Y}^i \right\rangle^2 - \sum_{i=1}^k \sum_{j=1}^N \left\| \boldsymbol{X}^i + \boldsymbol{Y}^i \right\|_F^2 \right| \le m\delta_{3r} \max_{i=1,\ldots,k} \left\| \boldsymbol{X}_i + \boldsymbol{Y}_i \right\|_F^2.$$

Applying triangle inequality, gives us:

$$\begin{aligned}
2 \left| \sum_{i=1}^k \sum_{j=1}^N \left\langle \boldsymbol{G}_j^i, \boldsymbol{X}^i \right\rangle \left\langle \boldsymbol{G}_j^i, \boldsymbol{Y}^i \right\rangle - \sum_{i=1}^k \sum_{j=1}^N \left\langle \boldsymbol{X}^i, \boldsymbol{Y}^i \right\rangle \right| &\le m\delta_{3r} \max_{i=1,\ldots,k} \left\| \boldsymbol{X}_i + \boldsymbol{Y}_i \right\|_F^2 \\
&\quad + \left| \sum_{i=1}^k \sum_{j=1}^N \left\langle \boldsymbol{G}_j^i, \boldsymbol{X}^i \right\rangle^2 - \sum_{i=1}^k \sum_{j=1}^N \left\| \boldsymbol{X}^i \right\|_F^2 \right| \\
&\quad + \left| \sum_{i=1}^k \sum_{j=1}^N \left\langle \boldsymbol{G}_j^i, \boldsymbol{Y}^i \right\rangle^2 - \sum_{i=1}^k \sum_{j=1}^N \left\| \boldsymbol{Y}^i \right\|_F^2 \right|
\end{aligned}$$

Now, applying $3r-$GRIP for the last two terms in the RHS, gives us the bound:

$$\leq m\delta_{3r} \left( \max_{i=1,\ldots,k} \left\| \boldsymbol{X}^i + \boldsymbol{Y}^i \right\|_F^2 + \max_{i=1,\ldots,k} \left\| \boldsymbol{X}^i \right\|_F^2 + \max_{i=1,\ldots,k} \left\| \boldsymbol{Y}^i \right\|_F^2 \right)$$

$$\leq 3m\delta_{3r} \max_{i=1,\ldots,k} \left( \left\| \boldsymbol{X}^i + \boldsymbol{Y}^i \right\|_F^2 + \left\| \boldsymbol{X}^i \right\|_F^2 + \left\| \boldsymbol{Y}^i \right\|_F^2 \right)$$

$$\leq 6m\delta_{3r} \max_{i=1,\ldots,k} \left( \left\| \boldsymbol{X}^i \right\|_F^2 + \left\| \boldsymbol{Y}^i \right\|_F^2 + \left\| \boldsymbol{X}^i \right\|_F \left\| \boldsymbol{Y}^i \right\|_F \right)$$

where in the second inequality we have used the simple fact that if $\{a_i\}_{i=1}^k, \{b_i\}_{i=1}^k, \{c_i\}_{i=1}^k$ are collections of positive numbers, than the following inequality holds:

$$\max_{i=1,\ldots,k} a_i + \max_{i=1,\ldots,k} b_i + \max_{i=1,\ldots,k} c_i \leq 3 \max_{i=1,\ldots,k} (a_i + b_i + c_i).$$

Now, note that if we replace $(\boldsymbol{\Lambda}_1^i, \boldsymbol{\Lambda}_2^i, \boldsymbol{\Lambda}_3^i)$ with $(\lambda^i \boldsymbol{\Lambda}_1^i, \frac{\boldsymbol{\Lambda}_2^i}{\lambda^i}, \frac{\boldsymbol{\Lambda}_3^i}{\lambda^i})$, for a non-zero real $\lambda^i$, the LHS doesn't change and in the RHS we get $(\boldsymbol{X}^i, \boldsymbol{Y}^i)$ replaced with $\left( \frac{\boldsymbol{X}^i}{\lambda^i}, \lambda^i \boldsymbol{Y}^i \right)$. Therefore, optimizing over the RHS, gives us the final bound:

$$\leq 18m\delta_{3r} \max_{i=1,\ldots,k} \left( \left\| \boldsymbol{X}^i \right\|_F \left\| \boldsymbol{Y}^i \right\|_F \right),$$

which completes the proof.

### D.14. Proof of Proposition 5

We need to show that with high probability, for all $\boldsymbol{\Lambda} \in R^{r \times r}$, the following inequality holds:

$$\left| \sum_{i=1}^n \left\langle \boldsymbol{G}_i, \boldsymbol{U}\boldsymbol{\Lambda}\boldsymbol{V}^\top \right\rangle^2 - \sum_{i=1}^n \left\| \boldsymbol{U}\boldsymbol{\Lambda}\boldsymbol{V}^\top \right\|_F^2 \right| \leq n\delta_r \left\| \boldsymbol{U}\boldsymbol{\Lambda}\boldsymbol{V}^\top \right\|_F^2.$$

Note that, for invertible matrices $\boldsymbol{R}_1, \boldsymbol{R}_2 \in R^{r \times r}$, scalar $\gamma \in R$ under the transformations

$$\boldsymbol{U} \leftarrow \boldsymbol{U}\boldsymbol{R}_1, \boldsymbol{V} \leftarrow \boldsymbol{V}\boldsymbol{R}_2, \boldsymbol{\Lambda} \leftarrow \gamma(\boldsymbol{R}_1)^{-1}\boldsymbol{\Lambda}(\boldsymbol{R}_2^\top)^{-1},$$

the statement of the problem remains the same. Therefore WLOG, $\boldsymbol{U}$ and $\boldsymbol{V}$ have orthonormal columns, and the problem becomes showing that with high probability, for every $\boldsymbol{\Lambda} \in R^{r \times r}$ such that $\|\boldsymbol{\Lambda}\|_F = 1$, the following inequality holds:

$$\left| \frac{1}{n} \sum_{i=1}^n \left\langle \boldsymbol{G}_i, \boldsymbol{U}\boldsymbol{\Lambda}\boldsymbol{V}^\top \right\rangle^2 - 1 \right| \leq \delta_r,$$

Define a linear map $\mathcal{A}$ as follows:

$$\mathcal{A}(\boldsymbol{\Lambda}) = \begin{pmatrix} \frac{1}{\sqrt{n}} \left\langle \boldsymbol{G}_1, \boldsymbol{U}\boldsymbol{\Lambda}(\boldsymbol{V})^\top \right\rangle \\ \vdots \\ \frac{1}{\sqrt{n}} \left\langle \boldsymbol{G}_n, \boldsymbol{U}\boldsymbol{\Lambda}(\boldsymbol{V})^\top \right\rangle \end{pmatrix}$$

Let $\delta = \frac{\delta_r}{9}$. Let $B_r = \{\boldsymbol{\Lambda} \in R^{r \times r} : \|\boldsymbol{\Lambda}\|_F = 1\}$ and $\bar{S}_r$ be a $\delta$ cover for $B_r$ in $\|\cdot\|_F$. By classical covering results we have:

$$|\bar{S}_r| \leq \left( \frac{3}{\delta} \right)^{r^2}.$$

Now, by Lemma 11, we have that for a fixed $\boldsymbol{\Lambda} \in \bar{B}_r$, the following holds:

$$\mathbf{P}\left[ \left| \|\mathcal{A}(\boldsymbol{\Lambda})\|_2^2 - 1 \right| \geq \delta \right] \leq C_1 e^{-cn\delta^2}.$$

Applying union bound, gives us:

$$\mathbf{P}\left[\left|\|\mathcal{A}(\mathbf{\Lambda})\|_2^2 - 1\right| \le \delta \text{ for every } \mathbf{\Lambda} \in \bar{S}_r\right] \ge 1 - |\bar{S}_r| C_1 e^{-cn\delta^2} \ge 1 - C_1\left(\frac{3}{\delta}\right)^{r^2} e^{-cn\delta^2}$$

$$\ge 1 - C_1 e^{-cn\delta^2 + r^2 \log\left(\frac{3}{\delta}\right)} \ge 1 - C_1 e^{-cn\delta^2\left(1 - \frac{r^2 \log\left(\frac{3}{\delta}\right)}{cn\delta^2}\right)}$$

$$\ge 1 - C_1 e^{-\frac{cn\delta^2}{2}},$$

where the last inequality follows from the fact that $n \ge \frac{2r^2 \log\left(\frac{3}{\delta}\right)}{c\delta^2}$.

Now, denote:

$$\gamma = \sup\{\|\mathcal{A}(\mathbf{\Lambda})\|_2 : \mathbf{\Lambda} \in B_r\}.$$

Let us fix $\mathbf{\Lambda} \in B_r$. We know that there is $\bar{\mathbf{\Lambda}} \in \bar{B}_r$, such that $\|\mathbf{\Lambda} - \bar{\mathbf{\Lambda}}\|_F \le \delta$. Applying the triangle inequality and using the fact that

$$\|\mathcal{A}\left(\mathbf{\Lambda} - \bar{\mathbf{\Lambda}}\right)\|_2 = \|\mathbf{\Lambda} - \bar{\mathbf{\Lambda}}\|_F \left\|\mathcal{A}\left(\frac{\mathbf{\Lambda} - \bar{\mathbf{\Lambda}}}{\|\mathbf{\Lambda} - \bar{\mathbf{\Lambda}}\|_F}\right)\right\|_2 \le \|\mathbf{\Lambda} - \bar{\mathbf{\Lambda}}\|_F \gamma,$$

gives us:

$$\|\mathcal{A}\left(\mathbf{\Lambda}\right)\|_2 = \|\mathcal{A}\left(\mathbf{\Lambda} - \bar{\mathbf{\Lambda}} + \bar{\mathbf{\Lambda}}\right)\|_2 \le \|\mathcal{A}\left(\bar{\mathbf{\Lambda}}\right)\|_2 + \|\mathcal{A}\left(\mathbf{\Lambda} - \bar{\mathbf{\Lambda}}\right)\|_2 \le \|\mathcal{A}\left(\mathbf{\Lambda} - \bar{\mathbf{\Lambda}}\right)\|_2 + 1 + \delta$$

$$\le \gamma \|\mathbf{\Lambda} - \bar{\mathbf{\Lambda}}\|_F + 1 + \delta \le \gamma\delta + 1 + \delta.$$

Taking supremum over $B_r$ in the LHS, we conclude:

$$\gamma \le \gamma\delta + 1 + \delta,$$

from which it follows $\gamma = \frac{1+\delta}{1-\delta} \le (1+\delta)^2 \le 1 + 3\delta$. Similarly for every $\mathbf{\Lambda} \in B_r$, we have:

$$\|\mathcal{A}\left(\mathbf{\Lambda}\right)\|_2 \ge \|\mathcal{A}\left(\bar{\mathbf{\Lambda}}\right)\|_2 - \|\mathcal{A}\left(\mathbf{\Lambda} - \bar{\mathbf{\Lambda}}\right)\|_2 \ge \sqrt{1 - \delta} - \|\mathbf{\Lambda} - \bar{\mathbf{\Lambda}}\|_F \gamma$$

$$\ge 1 - \delta - \gamma\delta \ge 1 - 3\delta.$$

Combigning both bounds we conclude the following:

$$\mathbf{P}\left[\left|\|\mathcal{A}\left(\mathbf{\Lambda}\right)\|_2^2 - 1\right| \le 9\delta \text{ for all } \mathbf{\Lambda} \in R^{r \times r}\right] \ge 1 - C_1 e^{-\frac{cn\delta^2}{2}}.$$

Taking $\delta = \frac{\delta_r}{9}$ completes the proof.

## E. Appendix

### E.1. Used tasks

Here we list the tasks used in the experiments. We write explicitly some definitions of the tasks, but for most of them we give the task code and refer the reader to https://github.com/allenai/natural-instructions.

The tasks used with $T_1^1$ in **??**, from most similar to least similar are:

- task370(given a list remove numbers that are divisible by 3), task205(given a list remove numbers that are even), task367 (given a list remove numbers that are not integer)

- task370,task205,task097(given a list remove duplicates)

- task370,task205,task488(given a list extract alphabetical elements)

- task370,task205,task506(positions of all alphabetical numbers in a list)

- task205,task093,task206

- task370,task637,task1214

- task370,task378,task586

- task064,task504,task096

- task162,task378,task586

- task162,task1203,task586

- task1210,task1203,task586

The tasks used with $T_1^2$ in **??**, from most similar to least similar are:

- task852(given a list of lists, multiply all odd elements in each list),task371(given a list of lists, multiply all numbers in each list),task207(given a list of lists, find the maximum in each list)

- task852,task371,task205

- task852,task371,task637

- task852,task373,task098

- task371,task095,task904

- task373,task1446,task1214

- task373,task199,task1214

- task1308,task199,task1214

The tasks used in subsection A.3, from most similar to least similar are:

- task1206,task1211,task1202,task1215

- task367,task372,task369,task205

- task851,task852,task368,task207

- task064,task100,task091,task099

- task373,task374,task368,task125

- task064,task078,task091,task099

- task605,task497,task636,task637

- task123,task205,task098,task097

- task851,task497,task369,task206

- task366,task851,task374,task123

- `task497,task636,task205,task208`

- `task267,task063,task509,task125`

- `task267,task600,task499,task370`

- `task1446,task160,task523,task499`

- `task064,task162,task494,task1197`

- `task1542,task157,task378,task097`

- `task078,task523,task371,task162`

- `task850,task122,task1203,task617`

The tasks used in **??** are `task095,task097,task098,task123,task205,task207,task366,task367, task368,task369,task370,task372,task373,task374,task497,task605,task636,task637,task851, task852`.

