# OpenReview forum: "Collaborative and Efficient Fine-tuning: Leveraging Task Similarity"
_ICML.cc/2026/Conference — ICML 2026 regular_

### Official Review · Reviewer_cdhp · 2026-03-07

**Soundness:** 2
**Presentation:** 3
**Significance:** 2
**Originality:** 2
**Overall Recommendation:** 3
**Confidence:** 4

**Summary:**

The authors propose CoLoRA which trains shared and personalized adapters and enables scarce data clients to benefit from other clients' data. The proposed adapter structure is motivated by a notion of task similarity, measured as the overlap between the column (or row) subspaces of fine-tuned adapters of different tasks. CoLoRA is theoretically analyzed for linear regression tasks where sample complexity and reconstruction error is provided. The experimental results show that when tasks are similar, CoLoRA improves over evaluated baselines.

**Compliance With Llm Reviewing Policy:**

Affirmed.

**Final Justification:**

The authors addressed my concern regarding the lack of extensive evaluation. The new results strengthen the empirical evaluation.

My main concern was the paper’s positioning in terms of what it claims to solve. It was quite unclear what the paper meant by efficiency in the context of collaborative learning. The authors acknowledged that their original framing (both in the paper and in the first rebuttal reply) was imprecise. In the new light that communication cost is the main bottleneck and not the number of parameters trained, I believe the paper’s first few sections need significant refinement and reassessment before acceptance.

For these reasons, and in the interest of scientific clarity, I cannot recommend acceptance in its current form. Hence, I will keep my score. That said, I am confident that a revised version with improved framing and clearer articulation of the core contributions would offer a much stronger and more compelling paper.

**Key Questions For Authors:**

1. Why is the proposed task similarity notion limited to language tasks? It seems to be applicable to even vision tasks as it just depends on the parameters of the fine-tuned LoRA adapters.

**Limitations:**

Please see my weaknesses section above.

**Strengths And Weaknesses:**

## Strengths

1. This paper is well written and presents the preliminaries and theoretical analysis in a clear and well-structured manner.
2. The proposed notion of task similarity and its extensive theoretical analysis with linear regression is valuable and has potential for various applications.
3. The authors provides sufficient evidence that the proposed notion does capture task similarity, e.g. by comparing similarity with known semantically similar tasks (Fig. 1).
4. The connection to matrix-sensing and clear description of federated CoAltMin (Algorithm 2) is also appreciated.

## Weaknesses

**Motivation:** One of the original motivations of model personalization in FL is to avoid training a single global model when the data across clients is too heterogeneous for a single global model to reliably solve all local objectives simultaneously [1]. However, when the clients are similar i.e. hold less heterogeneous data, a single global model should still suffice. The motivation of authors to personalize models when tasks are similar and further thrive under higher similarity thus contradicts the common understanding behind model personalization in FL.

**Positioning**
1) On one hand, the paper positions itself to solve the *data scarcity* problem by leveraging more clients and thus obtaining a larger data pool. However, this is a standard motivation for collaboration. Thus if the novelty is to run collaborative training only over similar clients, how is the server going to identify such similar clients without running the training in first place? This is exactly what clustering based approaches propose to do, such as the ones cited by the authors [3] or others [1]. Therefore, the positioning of the paper is unclear.
2) On the other hand, the paper also positions itself to solve *efficiency* of collaborative fine-tuning. In this context, the notion of task scalability is not well defined. What is the main bottleneck? Is it communication or purely the number of parameters? If is it the latter, why is it a problem if the final deployed model at each client is still just going to be one model (whether personalized or not)?

**Lack of extensive evaluations**
1) The client pool size of each experiment is just four. The experiments should be validated with at least a few tens of clients in each pool.
2) Evaluations are limited to only one base model.
3) The authors must demonstrate the effectiveness of CoLoRA against stronger FL LoRA baselines, such as Ravan [2] on larger client pools and bigger base models. There should be convincing evidence that model personalization is a necessity even when tasks are similar.

### References
[1] Liu, J., Lai, F., Dai, Y., Akella, A., Madhyastha, H. V., & Chowdhury, M. (2023, October). Auxo: Efficient federated learning via scalable client clustering. In Proceedings of the 2023 ACM Symposium on Cloud Computing (pp. 125-141).

[2] Raje, A., Askin, B., Jhunjhunwala, D., & Joshi, G. Ravan: Multi-Head Low-Rank Adaptation for Federated Fine-Tuning. In The Thirty-ninth Annual Conference on Neural Information Processing Systems. (NeurIPS 2025)

[3] Gabrielsson, R. B., Zhu, J., Bhardwaj, O., Choshen, L., Greenewald, K., Yurochkin, M., & Solomon, J. (2025, October). Compress then Serve: Serving Thousands of LoRA Adapters with Little Overhead. In International Conference on Machine Learning (pp. 18062-18095). PMLR.

---

> ### Author Rebuttal · Authors · 2026-03-30
>
> We thank the reviewer for their detailed comments and insights. Please see our response below.
>
> **New larger-scale experiments.** As suggested by the reviewer, we conducted large-scale experiments with 20 users and further evaluated our method on a 3B-parameter model. As shown in the table below, CoLoRA consistently outperforms the baselines.
>
> ### Experiment with 20 clients, rougeL scores
>
> | Method | T1        | T2        | T3        | T4        | T5        | T6        | T7        | T8        | T9        | T10       | T11       | T12       | T13       | T14       | T15       | T16       | T17       | T18       | T19       | T20       |
> |--------|-----------|-----------|-----------|-----------|-----------|-----------|-----------|-----------|-----------|-----------|-----------|-----------|-----------|-----------|-----------|-----------|-----------|-----------|-----------|-----------|
> | CoLoRA | **1.000** | 0.960     | **0.991** | 0.961     | **0.996** | **0.938** | **0.864** | **0.897** | **0.957** | **0.850** | **0.965** | 0.175     | 0.932     | 0.990     | **1.000** | **0.985** | **0.988** | **0.986** | **0.451** | **0.529** |
> | RoLoRA | 0.960     | 0.919     | **0.991** | 0.944     | 0.890     | 0.860     | 0.739     | 0.833     | 0.820     | 0.711     | 0.780     | 0.173     | 0.933     | 0.972     | 0.997     | 0.886     | 0.964     | 0.946     | 0.260     | 0.201     |
> | ALoRA  | **1.000** | 0.960     | 0.987     | 0.971     | 0.868     | 0.883     | 0.828     | 0.796     | 0.888     | 0.653     | 0.933     | 0.213     | **0.936** | 0.991     | **1.000** | 0.971     | 0.983     | 0.985     | 0.424     | 0.432     |
> | Local  | 0.920     | 0.937     | 0.962     | **0.972** | 0.786     | 0.848     | 0.787     | 0.790     | 0.853     | 0.664     | 0.876     | 0.253     | 0.917     | **0.996** | **1.000** | 0.924     | 0.972     | 0.974     | 0.376     | 0.331     |
> | FedDPA | **1.000** | 0.960     | 0.769     | 0.920     | 0.761     | 0.814     | 0.704     | 0.846     | 0.817     | 0.547     | 0.861     | **0.334** | 0.888     | **0.996** | **1.000** | 0.970     | 0.936     | 0.741     | 0.224     | 0.291     |
>
>
> | Method | Average |
> | ------ | ------- |
> | CoLoRA | **0.871**   |
> | RoLoRA | 0.790   |
> | ALoRA  | 0.835   |
> | Local  | 0.807   |
> | FedDPA | 0.769   |
>
> ### Larger model, rougeL scores
> **model: Qwen2.5-3B-Instruct, rank $r=4$**
>
> | Subspace similarity | 0.131 | 0.169 | 0.193 | 0.212 | 0.234 | 0.265 |
> |---------------------|-------|-------|-------|-------|-------|-------|
> | CoLoRA              | **0.815** | 0.967 | 0.959 | **0.961** | **0.760** | **0.920** |
> | RoLoRA              | 0.791 | 0.967 | 0.953 | 0.915 | 0.732 | 0.910 |
> | ALoRA               | 0.806 | 0.964 | **0.959** | 0.927 | 0.685 | 0.910 |
> | Local               | 0.760 | 0.955 | 0.946 | 0.922 | 0.661 | 0.880 |
> | FedDPA              | 0.753 | **0.972** | 0.919 | 0.717 | 0.699 | 0.860 |
>
> **Positioning.** We clarify that CoLoRA does not require pre-clustering of similar clients; the shared $A, B$ are averaged across all clients while personalized $\Lambda_i$ automatically absorbs heterogeneity (Figures 3, 4). CoLoRA is complementary to clustering: one could cluster first, then run CoLoRA within clusters. Regarding scalability, although each user has to store $O(dr + r^2)$ parameters, the bottleneck is *training* many more parameters (and not storing them). Local LoRA trains $O(kdr)$ parameters, which scale linearly with the number of users $k$. However, CoLoRA trains $O(dr + kr^2)$ parameters with $r \ll d$.
>
> **Need for personalization.** We thank the reviewer for this point. We respectfully argue there is no contradiction — the tension arises from conflating *similar* with *identical*. A single global model suffices when tasks are identical ($\xi = 1$), however, we study the more realistic regime where tasks are related yet distinct ($\xi < 1$), where personalization becomes crucial. In fact, model personalization has been an active area of research in federated and distributed training.
> The shared $A, B$ capture common structure while the personalized $\Lambda_i$ contain task-specific differences, that is, CoLoRA provides collaboration and personalization simultaneously.
>
> **Task similarity applicable to vision.** Our column subspace similarity metric (Definition 1) operates on LoRA adapter parameters and is indeed applicable to any modality, including vision. We referred to it as a similarity notion for "language tasks" because existing alternatives like Task2Vec are feasible for small vision models but prohibitively expensive for large language models, making our lightweight adapter-based metric particularly valuable in the language setting. We will revise the text to clarify this generality.
>
> **New baselines.** We agree with the reviewer to include further baselines; however, the cited method [2] is not open source, and we did not have enough time to implement it from scratch. We will examine it for the final revised paper.

---

> > ### Author Rebuttal · Reviewer_cdhp · 2026-04-04
> >
> > I thank the authors for their responses. I appreciate the new results with higher number of clients and a different base model. However, my concerns on positioning of the paper and evaluating stronger baselines are still unresolved.
> >
> > The training happens independently on each client where each trains $O(rd)$ parameters when using LoRA. Calling the collective number of parameters trained across all clients as a _bottleneck_ is thus misleading. As I understand, the bottleneck is either local i.e. per client or there is no bottleneck. I would appreciate the authors explanation on this point and I am happy to engage in a further discussion.
> >
> > I acknowledge the authors' clarification on the need for personalization by distinguishing between identical vs similar clients. However, I do not understand how the federated server is going to identify client groups for the proposed scenario where they have similar but not identical distributions and benefit from collaborative training to solve data scarcity. The premise of CoLoRA is leveraging task similarity. Therefore either this discovery is a part of the algorithm design or done separately via other methods such as clustering. The authors say clustering is orthogonal and can be applied, but I am not sure how the method is supposed to work without identify similar clients in the first place.
> >
> > I am still not convinced that model personalization is a necessity when the tasks are (highly) similar. Evaluation on stronger baselines is needed to sufficiently prove this necessity. The code for recommended baseline is open source and available in their supplementary material as a zip file.

---

> > > ### Author Response · Authors · 2026-04-06
> > >
> > > Thank you for the detailed feedback. Please see our response below.
> > >
> > > **Identical vs. similar tasks:** We would first like to note that in many applications, such as federated fine-tuning, task similarity is **known by construction**. For instance, in fine-tuning a next word prediction model for smartphone users, the task is already similar for the users, i.e., next word prediction. The heterogeneity only lies in the local data distribution (e.g., personal writing styles), not in the underlying task itself. In such applications, one does not need to estimate or discover task similarity -- it is an inherent property of the application itself. This is one promising application where CoLoRA is naturally deployed.
> > > That said, we agree that a *feedback mechanism* would expand CoLoRA's practicality to more heterogeneous environments. One natural approach is to monitor local task similarities via the alignment of local and global column subspaces of $B$ adapters during fine-tuning. In case the alignment becomes poor, the federated server could attenuate the contributions of particular users or switch to exclusively local fine-tuning entirely. We will discuss the feedback mechanism in the revision and provide an ablation experiment, showcasing CoLoRA's adaptation to more heterogeneous tasks.
> > >
> > > **Evaluation on more baselines:** As suggested by the reviewer, we experimented with an additional baseline, i.e., **RAVAN**, and provide the results below. We used 4 heads with rank 80. We note that RAVAN does not generate personalized models, which constitutes one major intuition behind the superior performance of personalized methods such as CoLoRA.
> > >
> > > ### Experiment with 20 users/tasks, rougeL scores
> > > model: **Qwen2.5-1.5B-Instruct**, rank $ r=4 $
> > >
> > > | Method |   T1   |   T2   |   T3   |   T4   |   T5   |   T6   |   T7   |   T8   |   T9   |  T10   |  T11   |  T12   |  T13   |  T14   |  T15   |  T16   |  T17   |  T18   |  T19   |  T20   |
> > > |--------|--------|--------|--------|--------|--------|--------|--------|--------|--------|--------|--------|--------|--------|--------|--------|--------|--------|--------|--------|--------|
> > > | **CoLoRA** | **1.000** | 0.960 | **0.991** | 0.961 | **0.996** | **0.938** | **0.864** | **0.897** | **0.957** | **0.850** | **0.965** | 0.175 | 0.932 | 0.990 | **1.000** | **0.985** | **0.988** | **0.986** | **0.451** | **0.529** |
> > > | RoLoRA | 0.960 | 0.919 | **0.991** | 0.944 | 0.890 | 0.860 | 0.739 | 0.833 | 0.820 | 0.711 | 0.780 | 0.173 | 0.933 | 0.972 | 0.997 | 0.886 | 0.964 | 0.946 | 0.260 | 0.201 |
> > > | ALoRA  | **1.000** | 0.960 | 0.987 | 0.971 | 0.868 | 0.883 | 0.828 | 0.796 | 0.888 | 0.653 | 0.933 | 0.213 | **0.936** | 0.991 | **1.000** | 0.971 | 0.983 | 0.985 | 0.424 | 0.432 |
> > > | Local  | 0.920 | 0.937 | 0.962 | 0.972 | 0.786 | 0.848 | 0.787 | 0.790 | 0.853 | 0.664 | 0.876 | 0.253 | 0.917 | **0.996** | **1.000** | 0.924 | 0.972 | 0.974 | 0.376 | 0.331 |
> > > | FedDPA | **1.000** | 0.960 | 0.769 | 0.920 | 0.761 | 0.814 | 0.704 | 0.846 | 0.817 | 0.547 | 0.861 | **0.334** | 0.888 | **0.996** | **1.000** | 0.970 | 0.936 | 0.741 | 0.224 | 0.291 |
> > > | **RAVAN**  | 0.920 | 0.935 | 0.935 | **0.974** | 0.572 | 0.811 | 0.727 | 0.714 | 0.686 | 0.587 | 0.716 | 0.166 | 0.895 | 0.966 | 0.999 | 0.952 | 0.977 | 0.938 | 0.295 | 0.213 |
> > >
> > >
> > > | Method | Average |
> > > | ------ | ------- |
> > > | CoLoRA | **0.871**   |
> > > | RoLoRA | 0.790   |
> > > | ALoRA  | 0.835   |
> > > | Local  | 0.807   |
> > > | FedDPA | 0.769   |
> > > | RAVAN  | 0.749   |
> > >
> > >
> > >
> > > | Method  | 0.131 | 0.169 | 0.193 | 0.212 | 0.234 | 0.265 |
> > > |---------|-------|-------|-------|-------|-------|-------|
> > > | CoLoRA  | 0.833 | **0.944** | **0.948** | **0.961** | **0.727** | **0.970** |
> > > | RoLoRA  | 0.829 | 0.866 | 0.935 | 0.932 | 0.593 | 0.930 |
> > > | ALoRA   | **0.839** | 0.915 | 0.940 | 0.942 | 0.653 | 0.940 |
> > > | Local   | 0.806 | 0.905 | 0.937 | 0.939 | 0.635 | 0.910 |
> > > | FedDPA  | 0.755 | 0.831 | 0.906 | 0.691 | 0.457 | 0.950 |
> > > | RAVAN   | 0.798 | 0.895 | 0.813 | 0.862 | 0.487 | 0.930 |
> > >
> > > **Bottleneck:** We thank the reviewer for this precise observation. The reviewer is correct that the per-client LoRA training cost is $O(dr)$ with or without personalization, and the collective parameter count across all clients does not constitute a computational bottleneck. We acknowledge that our original phrasing was imprecise.
> > > Our efficiency claim for CoLoRA is primarily about communication cost, not local computation. In standard federated (full) fine-tuning, each client must communicate the full model update of size $O(d^2)$ to the server per communication round. As a LoRA-type method, CoLoRA reduces this to $O(dr)$ per round by restricting updates to the low-rank factors. Moreover, periodic averaging further reduces the total number of communication rounds required. Together, these two mechanisms yield a significant reduction in overall communication load compared to naive federated fine-tuning. We will revise the language around collective parameter counts as the bottleneck in the revision, as pointed out.

---

### Official Review · Reviewer_PQU2 · 2026-03-11

**Soundness:** 3
**Presentation:** 3
**Significance:** 2
**Originality:** 2
**Overall Recommendation:** 4
**Confidence:** 5

**Summary:**

This paper proposes CoLoRA, which decomposes each user's LoRA adapter as BΛᵢA with shared B, A and task specific r×r matrices Λᵢ, reducing parameters for collaborative fine-tuning. Experiments on Natural Instructions with Qwen2.5-1.5B show CoLoRA outperforms Local LoRA and federated baselines when tasks are sufficiently similar. However, some decisions in the experiment setup deviate from best practices, undermining the overall soundness.

**Compliance With Llm Reviewing Policy:**

Affirmed.

**Key Questions For Authors:**

- My soundness concern mainly come from 1. the evaluation metric, 2. lora hyper-params. If you can share results with exact match accuracy, breaking down by task. Rerun some of the exps with higher lora rank. I'm open to change my opinion.

**Limitations:**

The paper has some technical weaknesses, but they are not inherently unaddressable in the scope of this work, I would prefer resolve them in review and rebuttal rather than shun them away as limitations.

**Strengths And Weaknesses:**

Strength:
- There is some theory stuff, looks cool.

Weakness:
- Multiple inconsistency with standard practices: Lora rank=4 is very rarely used, and there are results showing rank matters for lora to function properly (LoRA Learns Less and Forgets Less). Rouge-L is mostly used for translation and summarization, where the ground truth is just one example among many good answers, for tasks in program execution, exact match makes more sense.
- Overestimated originality: The training algorithm CoLoRA is less distinct from (Serving thousands of LoRA adapters with little overhead) than how the authors saw it. That paper actually come with three variants, JD-Full JD-Diag and JD-Cluster, where JD-Full also use r x r matrices for task-specific information while having globally shared A and B matrix. In addition, the CoAltMin algorithm to solve U and V is closely resembles the algorithm proposed for JD-Full.
- Mild practical issues: The similarity estimation uses 600 examples per task, while the actual training uses 50. This difference crease confusion about when to use this algorithm. If I only have 50 examples, can I still find most similar tasks? And if I do have as many as 600 examples, is it still meaningful to transfer from other tasks?

---

> ### Author Rebuttal · Authors · 2026-03-30
>
> We thank the reviewer for their feedback. Please see our response in two main fronts below.
>
>  **Novelty.** First, we would like to begin by clarifying our novelty compared to the cited work [1] by the reviewer. We respectfully emphasize that the two works address fundamentally different problems. CoLoRA is a *training* method: it learns the shared factors $A, B$ and personalized $\Lambda_i$ end-to-end from data, with the goal of improving fine-tuning performance under data scarcity. In contrast, the cited work compresses *already-trained* independent LoRA adapters into the $B\Lambda_i A$ form — it does not propose a training procedure or address collaborative learning. Regarding CoAltMin, while alternating minimization over shared and personalized factors is a natural algorithmic choice that appears in both settings, our theoretical contribution lies in the convergence and sample complexity guarantees (Theorem 1) under task similarity, which has no counterpart in the cited work.
>
> **LoRA rank.** We chose $r = 4$ primarily because CoLoRA's parameter advantage is most pronounced at small ranks — the personalized overhead scales as $O(kr^2)$, so small $r$ is precisely the regime where CoLoRA offers the greatest efficiency gains over local LoRA ($O(kdr)$). That said, we conducted new experiments with larger rank $r=16$ and provide the results below.
>
> **Exact match.** We agree that exact match is a more natural metric for program execution tasks, where outputs are deterministic. We ran new experiments and report *exact match* as an additional evaluation metric as requested by the reviewer (please see below).
>
> **Practical note.** The 600 samples were used solely for the *offline analysis* of the similarity metric (Figure 1, Section 2.3) — not as a prerequisite for running CoLoRA. In practice, CoLoRA does not require explicit similarity estimation: if domain knowledge suggests tasks are related, users can apply CoLoRA directly. Our experiments confirm expected degradation when similarity is low, so applying it speculatively carries little risk.
>
> **Larger Model.** We also conducted large-scale experiments on a larger model Qwen2.5-3B-Instruct (**3B** parameters) (see our response to Reviewer **cdhp**).
>
> ### Larger rank, exact matching score
>
> **model: Qwen2.5-1.5B-Instruct, rank $r=16$**
>
> | Subspace similarity | 0.131 | 0.169 | 0.193 | 0.212 | 0.234 | 0.265 |
> |---------------------|-------|-------|-------|-------|-------|-------|
> | CoLoRA              |**0.630**|**0.850**|**0.870**|**0.750**|**0.720**| 0.940 |
> | RoLoRA              | 0.600   | 0.720   | 0.790   | 0.620   | 0.470   | 0.870 |
> | ALoRA               |**0.630**| 0.740   | 0.730   | 0.690   | 0.480   |**0.960**|
> | Local               |0.620    | 0.720   | 0.670   | 0.650   | 0.380   | 0.920 |
> | FedDPA              | 0.580   | 0.670   | 0.610   | 0.030   | 0.320   | 0.880 |
>
> ### 20 client experiment, exact matching score
>
> | Method | T1        | T2        | T3        | T4        | T5        | T6        | T7        | T8        | T9        | T10       | T11       | T12       | T13       | T14       | T15       | T16       | T17       | T18       | T19       | T20       |
> |--------|-----------|-----------|-----------|-----------|-----------|-----------|-----------|-----------|-----------|-----------|-----------|-----------|-----------|-----------|-----------|-----------|-----------|-----------|-----------|-----------|
> | CoLoRA | **1.000** | **1.000** | **0.920** | 0.720     | **0.960** | **0.680** | **0.600** | **0.920** | **0.640** | **0.760** | **0.720** | 0.040     | **0.560** | 0.920     | **1.000** | **0.810** | **0.840** | **0.880** | **0.040** | 0.000 |
> | RoLoRA | 0.920     | 0.760     | **0.920** | 0.600     | 0.520     | 0.560     | 0.240     | 0.640     | 0.240     | 0.320     | 0.200     | 0.040     | **0.560** | 0.800     | 0.920     | 0.571     | 0.640     | 0.560     | 0.000     | 0.000 |
> | ALoRA  | **1.000** | 0.960     | 0.880     | **0.760** | 0.480     | 0.480     | 0.440     | 0.560     | 0.400     | 0.160     | 0.440     | 0.080     | **0.560** | **0.960** | **1.000** | 0.667     | **0.840** | 0.840     | **0.040** | 0.000 |
> | Local  | 0.920     | 0.880     | 0.760     | **0.760** | 0.240     | 0.400     | 0.280     | 0.560     | 0.320     | 0.240     | 0.320     | 0.120     | 0.520     | **0.960** | **1.000** | 0.667     | **0.840** | 0.720     | 0.000     | 0.000|
> | FedDPA | **1.000** | 0.960     | 0.200     | 0.480     | 0.280     | 0.360     | 0.200     | 0.800     | 0.200     | 0.320     | 0.320     | **0.160** | 0.320     | **0.960** | **1.000** | 0.762     | 0.360     | 0.160     | 0.000     | 0.000 |
>
> | Method | Average   |
> | ------ | --------- |
> | CoLoRA | **0.700** |
> | RoLoRA | 0.501     |
> | ALoRA  | 0.577     |
> | Local  | 0.525     |
> | FedDPA | 0.442     |
>
> [1]  Brüel-Gabrielsson et. al., Compress then Serve: Serving Thousands of LoRA Adapters with Little Overhead, ICML 2025.

---

> > ### Author Rebuttal · Reviewer_PQU2 · 2026-03-31
> >
> > Thanks for the clarification, as well as additional experiments.
> > The updated results are convincing, and the extend of new experiments are impressive. Thank you for your effort!
> > My concerns are resolved, I would be happy to change my assessment.

---

> > > ### Author Response · Authors · 2026-04-03
> > >
> > > We sincerely thank the reviewer for the positive assessment and for acknowledging our additional experiments. We are glad that the clarifications and new results were convincing. Your practical notes and suggestions have made our paper stronger.

---

### Official Review · Reviewer_NaBK · 2026-03-11

**Soundness:** 3
**Presentation:** 3
**Significance:** 2
**Originality:** 2
**Overall Recommendation:** 5
**Confidence:** 3

**Summary:**

The paper proposes a method for aggregating LoRA adapters from different clients in a federated learning setting, exploiting data similarity for efficient personalized fine-tuning of foundation models. The authors introduce two algorithms. The first, CoAltMin, finds shared orthogonal matrices $U$ and $V$ and client-specific diagonal matrices $\Lambda_i$ that minimize the sum of reconstruction errors $\| M_i - U \Lambda_i V^T\|^2$. This is solved via alternating optimization: $\Lambda_i$ is updated first, then $U$ and $V$ are updated using the current $\Lambda_i$. The second algorithm, CoLoRa-Alt is extends this idea to the federated LoRA setting, where the objective becomes: $min_{A,B, \Lambda_i}  \sum_{i=}^k \sum_{D_i} \ell(f_{W_o + A\lambda_iB}(x,y)$. The authors provide theoretical convergence guarantees for CoAltMin to the optimoal $U^*$ and $V^*$. Experiments are conducted on the NAtural Instruction dataset.

**Compliance With Llm Reviewing Policy:**

Affirmed.

**Final Justification:**

The paper makes a theoretical contribution introducing  CoAltMin and CoLoRA-Alt algorithm and giving sample complexity guarantees for CoAltMin. The writing is clear and the paper well organized. The rebuttal addressed most concerns adequately, reinforcing my original assessment. About the formal guarantees for Algorithm 2; a corollary bounding local drift would close this gap and should be add in the paper revision. Also, Algorithm 2 (being the method actually evaluated) should be added in the main body.

**Key Questions For Authors:**

Q1: Relationship to one-shot QR decomposition. How does the proposed method relate to performing a single QR decomposition on the concatenated $U$ and $V$  matrices, as done in TSV-M (Gargiulo et al.[1])? Intuitively, CoLoRA-Alt should outperform this since $\Lambda_i$ is also optimized, but a direct comparison would make the contribution clearer.

Q2. Ablation studies. Were ablations performed over the key hyperparameters $n, N, T$ in Algorithm 1?

Q3. Was a centralized LoRA baseline (with full data access) included as an oracle upper bound? This would contextualize how much is lost due to federation.

Q4. Why not study the linear regression setting with a LoRA-style parameterization to obtain a theoretical result closer to the actual algorithm used in practice?

**Limitations:**

yes

**Strengths And Weaknesses:**

### Strenghts:
S1. The paper provides theoretical convergence guarantees for Algorithm 1 (CoAltMin).

S2.  The paper is well written and clear.

### Weaknesses:
W1. Algorithm 2 is relegated to the appendix despite being the experimentally evaluated method, indeed all experiments on natural instructions dataset use CoLoRA-Alt (Algorithm 2), yet it appears in the appendix. Meanwhile, the convergence guarantees are for Algorithm 1 only, and no discussion is provided on whether or how these guarantees extend to Algorithm 2.
The role of task similarity is unclear in Algorithm 2. How inter-client data similarity is leveraged in CoLoRA-Alt is not explained, even though it is a central motivation of the paper.
The update of $\Lambda_i$ in Algorithm 2 is underspecified, in particular the TrainLocal step in Algorithm 2 does not clarify how $\Lambda_i$ is updated.

W2. Computational cost is not analyzed. CoLoRA-Alt involves multiple rounds of local training (first with $A$ fixed, then with $B$ fixed), effectively doubling the local training cost per round compared to standard LoRA. This overhead is never analyzed or discussed. Relatedly, it is unclear how the baselines are trained: are they given the same number of gradient steps, or the same wall-clock budget? This makes fair comparison difficult.
Moreover. evaluations on well-established benchmarks such as GLUE or commonsense reasoning datasets would substantially strengthen the empirical claims and facilitate comparison with the broader literature.

W3. Figures 3 and 4 lack statistical significance testing. The relationship between task similarity and ROUGE performance should be supported by appropriate statistical tests.

---

> ### Author Rebuttal · Authors · 2026-03-30
>
> We appreciate the reviewer's feedback and questions. Please see our response below for each point.
>
> **Relationship between Algorithms 1 and 2.** The two algorithms are conceptually identical — both alternate between updating shared matrices $U,V$ and personalized matrices $\Lambda_i$. Algorithm 1 assumes centralized data access for updating $U,V$; Algorithm 2 replaces this with local training plus averaging, making it federated-friendly by avoiding raw data sharing (addressing privacy and communication concerns). Since both minimize the same objective, we expect similar optimization behavior. The only additional theoretical challenge for Algorithm 2 is controlling local drift from local updates, which is standard in federated learning analysis.
>
> **Role of task similarity in Algorithm 2.** We would like to clarify that when tasks are similar, their optimal shared factors are close, so averaging $A$ (or $B$) across users produces a good common representation. Each client then benefits from this when updating its private $\Lambda_i$. When similarity is low, the averaged factors are less useful, matching the trend in our experiments. We will make this mechanism explicit in the revision.
>
> **Computational cost comparison.** The reviewer correctly notes that CoLoRA-Alt alternates between fixing $A$ and fixing $B$, which doubles the local training phases per round. However, we emphasize that the per-phase cost is comparable to standard LoRA -- in each phase, the number of trainable parameters is $ O (dr + kr^2)$ rather than $O(kdr)$ for independent LoRA. So, the overhead is in communication rounds, and not in parameter count. That said, we agree this tradeoff deserves explicit discussion and will add a computational cost analysis (wall-clock time, FLOPs per round, communication cost) in the revision.
>
> **Benchmarks and tasks.** All methods are compared under the same budget: identical numbers of trainable parameters, local training steps, and communication frequency, ensuring matched computational and communication costs. We will clarify this more explicitly in the revision. Regarding our task choice, we emphasize that the central premise of CoLoRA is to boost individual task performance by leveraging *task similarity*. Testing this claim requires a setting in which the similarity between the tasks can be varied controllably. Natural Instructions provide exactly such a setting to manifest our motivating premise.
>
> **Statistical tests for experiments.** We agree that reporting statistical significance would strengthen the empirical claims. In a revision, we will add confidence intervals (or standard deviations) across multiple random seeds for each experimental configuration.
>
> **Connection to Gargiulo et al.** Thank you for pointing out Gargiulo et al. [1]. There is indeed an interesting connection, as both their work and ours relate task similarity to the spectral structure of task-specific perturbations. However, the two approaches address fundamentally different problems. Gargiulo et al. focus on model merging, where independently trained adapters are combined via a single decomposition (e.g., QR), whereas our setting is collaborative learning, where shared and task-specific components are learned jointly during training. We will acknowledge the contributions of Gargiulo et al.[1] in the updated manuscript and will discuss the connection.
>
> **Oracle upper bounds.** Thank you for raising this point. We have computed the corresponding upper bounds (see below) and will include them in the revision.
>
> | Subspace similarity | 0.114 | 0.118 | 0.126 | 0.131 | 0.134 | 0.147 | 0.162 | 0.169 | 0.182 | 0.184 | 0.193 | 0.207 | 0.211 | 0.212 | 0.222 | 0.229 | 0.234 | 0.265 |
> |---------------------|-------|-------|-------|-------|-------|-------|-------|-------|-------|-------|-------|-------|-------|-------|-------|-------|-------|-------|
> | Oracle              | 0.512 | 0.763 | 0.696 | 0.908 | 0.680 | 0.580 | 0.827 | 0.982 | 0.950 | 0.936 | 0.977 | 0.981 | 0.976 | 0.993 | 0.886 | 0.893 | 0.925 | 0.987 |
>
> **Linear regression under LoRA.** Let us clarify that our theoretical formulation *is* already LoRA-style: $U,V$ correspond to shared factors $B,A$ and $\Lambda_i$s are personalized components. The QR orthonormalization in Algorithm 1 is a standard technical step to resolve scaling ambiguity — it does not change the recovered product $U \Lambda_i V^T$.
>
> **Larger-scale experiments and ablation studies.** To further support and corroborate our experimental results, we have conducted new experiments that could be of interest to the reviewer. In particular, we conducted large-scale experiments on a larger model Qwen2.5-3B-Instruct (**3B** parameters) and 20 users (see our response to Reviewer **cdhp**), and experiments with rank $r=16$ (see our response to Reviewer **PQU2**). We will include further ablations in a revision.
>
> [1] Gargiulo et al., Task Singular Vectors: Reducing Task Interference in Model Merging, CVPR, 2024.

---

> > ### Author Rebuttal · Reviewer_NaBK · 2026-04-04
> >
> > Thank you, I appreciate the response, most of my concern have been addressed. About the answer to W1 I still have some doubts.
> > The claim that the two algorithms are "conceptually identical" and that the analysis extends is not clear to me.
> > Algorithm 1 uses exact joint minimization; Algorithm 2 uses local SGD + averaging. These are not equivalent, does the proof still work  under approximate local updates?

---

> > > ### Author Response · Authors · 2026-04-05
> > >
> > > We thank the reviewer for acknowledging the rebuttal. Regarding Algorithms 1 and 2, the key point is that the exact minimization step in Algorithm 1 can be relaxed to multi-step SGD in Algorithm 2 for practical implementation. Let us justify this with a simple example. Denote each user's local loss with $f_i(u,v)$ and assume that it is $\mu$-strongly convex in $u,v$. Exact minimization at step $t$ in Algorithm 1 corresponds to solving $ \min_v \sum_{i=1}^{k} f_i(u_t,v) $. We can replace this exact minimization with sufficient steps of SGD on $ \sum_{i=1}^{k} f_i(u_t,v) $ with step-size $ \eta $, which induces an additional error term $ O(\eta \sigma^2 / \mu) $, with $ \sigma^2 $ being SGD noise. With proper choice of ste-size $ \eta $, we would still be able to recover the original ground truth with a slower rate. Now, we can do a second approximation. Instead of running (centralized) SGD on the aggregate loss $ \sum_{i=1}^{k} f_i(u_t,v) $, we repeatedly perform the following: update each $ v $ parameter locally only on $ f_i(u_t,v) $ (via $ S $ local SGD updates) and average the local updated model to get a new $ v $. After sufficient rounds of updates and with a proper step-size, the averaged model follows the model updated without local updates. The discrepancy between these two procedures is bounded by the local model drift, which scales as $ O( \eta^2 S^2 G^2) $ for $ S $ local steps with gradient bound $ G $ (see, e.g., Karimireddy et al., 2020). This is, in fact, the premise of FedAvg-type algorithms.
> > >
> > >  In short, with the following two approximation steps, we can connect the solutions found by Algorithms 1 and 2:
> > > $$
> > >  \text{Exactly solve } \min_v \sum_{i=1}^{k} f_i(u_t,v) \text{ at each step $t$ (Alg 1)}
> > > $$
> > > $$
> > > \downarrow +O(\eta \sigma^2 / \mu)
> > > $$
> > > $$
> > >  \text{Update $v$ by running multiple SGD steps on } \sum_{i=1}^{k} f_i(u_t,v)
> > > $$
> > > $$
> > > \downarrow +O( \eta^2 S^2 G^2)
> > > $$
> > > $$
> > >  \text{In each round, run $ S $ local SGD steps for $v$ on each } f_i(u_t,v) \text{ and average them periodically (Alg 2)}
> > > $$
> > >
> > >  We finally note that this exact vs. approximate gap is a standard and well-accepted pattern in the federated learning literature. For instance, Collins et al. (2021) analyze FedRep under exact local minimization of the client head $w_i^* = \arg⁡  \min⁡_w f_i(w;B) $, but implement it in practice with multiple epochs of local SGD. Their theoretical analysis relies on the sufficient descent achieved by exact minimization, and they empirically validate that increasing the number of local update steps (approaching exact minimization) monotonically improves convergence. Extending theoretical guarantees from algorithm 1 to algorithm 2 is pretty standard and doable; it requires bounding local model drifts at each communication round. In fact, a follow-up work closed the same gap for FedRep using the same logic (Collins et al, 2022).
> > >
> > >  A formal analysis extending the guarantees from Algorithm 1 to Algorithm 2 is feasible using standard local SGD drift bounds (Karimireddy et al., 2020), but would largely replicate known techniques. We chose to focus our theoretical contribution on the role of common and personalized components of the adapter. We are happy to include a formal remark and proof sketch in the appendix if the reviewer considers this valuable.
> > >
> > >
> > > Collins et al. (2021), Exploiting shared representations for personalized federated learning, ICML.
> > >
> > > Collins et al. (2022), FedAvg with Fine Tuning: Local Updates Lead to Representation Learning, NeurIPS.
> > >
> > > Karimireddy et al. (2020), SCAFFOLD: Stochastic Controlled Averaging for Federated Learning, ICML.

---

### Official Review · Reviewer_a9Yp · 2026-03-12

**Soundness:** 3
**Presentation:** 3
**Significance:** 2
**Originality:** 4
**Overall Recommendation:** 3
**Confidence:** 3

**Summary:**

This paper introduces CoLoRA, a method that exploits task similarity to collaboratively and efficiently fine-tune personalized foundation models.
They propose shared adapter capturing task similarities across all tasks and personalized adapters tailored to user-specific tasks.
There is theoretically deduction of the proposed collaborative alternative minimization method, CoAltMin, and provide guarantees for ground truth recovery.
The experiments are done on Qwen2.5-1.5B-Instruct, and compared with various PEFT Lora methods.

**Compliance With Llm Reviewing Policy:**

Affirmed.

**Final Justification:**

Part of my concerns are solved, but the assumption should be further clarified and more experiments should be done in the revision.

**Key Questions For Authors:**

1. The motivation of that paper is when tasks are already similar and data is very scarce. But such motivaiton is not well-supported, I am not sure whether CoLoRA is truly necessary, rather than just useful for a narrow setting?

2. The similarity notion is based mainly on column subspaces, motivated by the claim that A stays near initialization. How robust is this assumption across models, ranks, datasets? Did you observe cases where row-space information matters?

3. In low-similarity regimes, local lora appears better. Do you have a mechanism to automatically decide when a client should collaborate, with whom, or whether it should remain local? Also, since local lora outperforms CoLoRA at low similarity, isn’t the main challenge really task selection or clustering, rather than collaborative fine-tuning itself?

Minor question:
1. The paper positions itself as a solution for federated and distributed settings. However, communicating the  A and  B matrices iteratively can leak information about the local datasets. whether the personalized Λ_imatrices are sufficient to protect sensitive client data?

**Limitations:**

See Weaknesses.

**Strengths And Weaknesses:**

--- Strengths

1. It proposes a simple, parameter-efficient way to let multiple similar tasks share useful adaptation structure while still keeping task-specific personalization.
2. The paper is well-structured, and the claims are well-supported.
3. The paper is evaluated with tasks in federated setting, and compared with different LoRA baselines.

--- Weaknesses

1. In the paper, the similarity measure requires first training LoRA adapters independently and then comparing subspaces. I think it may weaken the claim to deploy the method in practical federated setting, since it is better to know similarity before deciding whom to collaborate with.

2. The experiments are mostly on natural instructions, especially program-execution tasks. They should not represent/cover the general and common settings for real users.

3. In the experiments, the authors manually construct groups of clients with known, varying levels of similarity to prove their point. They do not present an end-to-end system where a random pool of clients is automatically clustered into similar groups. However, this should be a strong baseline/ablation setting.

---

> ### Author Rebuttal · Authors · 2026-03-30
>
> We thank the reviewer for the feedback. Please see our response below.
>
> **Task diversity in experiments:** The central premise of CoLoRA is to boost individual task performance by leveraging *task similarity*. Testing this claim requires a setting in which the similarity between the tasks can be varied controllably. Natural Instructions and particularly Program Execution provide exactly such a setting to manifest our motivating premise. Given that there is no universally robust task similarity metric for language tasks, including tasks from significantly diverse pools would undermine a meaningful correlation between performance and similarity. We also note that although Program Execution is the primary task pool, we do include tasks from other pools (Sentiment Analysis, Common Sense Classification) as stated in Section 5. Nevertheless, we agree with the reviewer and believe expanding our method to more areas would better demonstrate CoLoRA's benefit over the baselines. We will make sure to include such new results in the revision.
>
> **Row subspace:** We would like to clarify that our theoretical analysis accounts for *both* row and column subspaces (see Section 4, Definition 2). As also shown in Theorem 1, the convergence guarantee depends on this joint similarity notion, and CoLoRA recovers both row and column ground-truth matrices $U^* $ and $V^* $.
> On the experiments front, the column subspace-based task similarity is chosen motivated by the evidence in [1], where adapter $A$ remains close to its initialization. This stems from the standard LoRA initialization, where $A$ and $B$ are initialized from a Gaussian distribution and zero, respectively. As a result, the effective update is driven through $B$, which limits the movement of $A$. Consequently, row subspace similarities remain nearly identical across tasks. One could take both row and column subspaces into account as a similarity metric; however, the identical row subspace similarity (due to LoRA) does not carry a strong signal.
>
> **Practicality of CoLoRA:** We believe that training on similar tasks is not a narrow scenario. It shows up in many settings, such as federated fine-tuning. In federated and distributed learning, individual clients (smartphones, hospitals, enterprise users) typically possess only small, private datasets. At the same time, these clients often perform *related* tasks. e.g., a next word predictor for writing an email. Our framework suggests fine-tuning a model using all users' data and personalizing it according to each user's writing style.
> Data scarcity is also a common challenge in distributed and federated learning since edge devices have typically far less storage capacity compared to data centers [2]. This makes efficient fine-tuning particularly challenging in edge applications such as smartphones, autonomous vehicles, and IoT devices. This also addresses another point raised by the reviewer regarding when to collaborate. Federated fine-tuning is a perfect example for scenarios where CoLoRA is expected to outperform local methods.
> We would like to add that assuming a known range of similarity between the tasks is quite well-established in the theoretical analysis of federated learning methods. For instance, distributed losses have closed gradients (as a proxy for task similarity).
>
> **Knowing similarities before training:** The reviewer's concern about knowing the task similarity is indeed totally valid. Language task similarity by itself is a quite challenging and under-explored area. In fact, we put significant effort into devising a robust and efficient task similarity notion for language tasks; however, existing notions used for small neural net models and simple image classification tasks do not apply to large language tasks. This is, in fact, an interesting direction to pursue.
>
> **Privacy leakage:** This is, in fact, an active area of research in federated learning. Although the crux of FL is to keep user data local, it still communicates rich gradient signals between the users and the coordinator and private data could be inferred from these signals. This same concern applies to our CoLoRA methods as well. However, there have been several fixes developed for this. The two main tools are *encryption* and *differential privacy*. We can further encrypt $A$ and $B$ signals during the training, or inject structured noise before communicating them.
>
> **Larger-scale experiments:** We would also like to highlight additional experiments we have conducted that could be of interest to the reviewer. In particular, we conducted large-scale experiments on a larger model Qwen2.5-3B-Instruct (**3B** parameters) and 20 users (see our response to Reviewer **cdhp**) and with rank $r=16$ (see our response to Reviewer **PQU2**).
>
> [1] Ban, H. and Ji, K. Rethinking parameter sharing for llm fine-tuning with multiple loras, 2025.
>
> [2] Chen et al., An empirical survey of data augmentation for limited data learning in NLP,  2023.

---

> > ### Author Rebuttal · Reviewer_a9Yp · 2026-04-03
> >
> > Thank you for your response.
> > 1. Although the author have explained the task similarity as a well-defined assumption from the federated setting. However, I think it is not so responsible to delevop/analyze methods on a totally ideal assumption, especially federated learning itself focus on the practical performace.
> >
> > Even though task similarity can be pre-computed then apply the method and the author also discussed the performance across similarities, to ensure the practicability, there should be at least a 'feedback' mechanism that enable the method to be 'aware' of possible similarities. By doing this, so maybe some hyper-parameters can be changed or even turn into the local lora (since in low-similarity regimes, local lora appears better). I think by this way, this work can mitigate the gap to practicability.
> >
> > 2. For the evaluation on more tasks, I expect the revision from authors in revision, and I think it really worth to be discussed to make the paper more complete. Especially, I recommend to add some cases that you said as the 'narrow scenario', which can further support your motivation.

---

> > > ### Author Response · Authors · 2026-04-03
> > >
> > > 1. We appreciate the constructive suggestion. We would first like to note that in many applications, such as federated fine-tuning, task similarity is **known by construction**. For instance, in fine-tuning a next word prediction model for smartphone users, the task is already similar for the users, i.e., next word prediction. The heterogeneity only lies in the local data distribution (e.g., personal writing styles), not in the underlying task itself. In such applications, one does not need to estimate or discover task similarity -- it is an inherent property of the application itself. This is one promising application where CoLoRA is naturally deployed.
> > > That said, we agree that a *feedback mechanism* would expand CoLoRA's practicality to more heterogeneous environments,  as suggested by the reviewer. One natural approach is to monitor local task similarities via the alignment of local and global column subspaces of $B$ adapters during fine-tuning. In case the alignment becomes poor, one could attenuate the contribution of particular users or switch to exclusively local fine-tuning entirely. We will discuss the feedback mechanism in the revision and provide an ablation experiment, showcasing CoLoRA's adaptation to more heterogeneous tasks.
> > >
> > > 2. We agree with the reviewer about expanding the experiment setting. As we reported larger-scale experiments in the rebuttal (Qwen2.5-3B-Instruct with **3B** parameters, **20** tasks, and rank $r=16$), we will conduct additional experiments aligned with federated fine-tuning applications and include in the revision (e.g., personalized next-word prediction or email writing across users with related but distinct styles).

---

### Decision · Program_Chairs · 2026-04-30

**Decision:**

Accept (regular)

**Comment:**

The reviewers' main concerns focus on the presentation, originality, and assumptions of the proposed method, as well as the results of the experimental settings.

After the rebuttal, most of the concerns have been addressed. As noted by one reviewer, the authors should position this paper more effectively by more precisely emphasizing the objective of the proposed method.

This is a borderline paper with merit. Thus, I recommend a weak acceptance.

Note: one problematic reference is found, Reference: Standley, T., Zamir, A. R., Chen, D., Guibas, L., Malik, J., and Savarese, S. Identifying beneficial task relations for multi-task learning in deep neural networks. arXiv preprint arXiv:1702.08303, 2017. --- Issue: authors mismatch with axXiv.